# 👜ENTP: Enhancing Low-Quality SFT Data via Neural-Symbolic Text Purge-Mix

## Abstract

Supervised Fine-Tuning (SFT) adapts pre-trained Large Language Models (LLMs) to domain-specific instructions by training on a carefully curated subset of high-quality instruction–response pairs, typically drawn from a larger dataset that often contains many low-quality or noisy samples. Despite its effectiveness, this *quality-first* paradigms often suffer from two caveats. On the one hand, *quality filters are inherently imperfect*, many samples that pass through these filters are not truly high-quality. On the other hand, discarding the vast majority of low-quality or frequently occurring examples *may lose potentially valuable signal*. As much of the readily available instruction-following data online has already been utilized, further improvements now depend on leveraging, rather than discarding, the examples that were previously filtered out. To address these two issues, we introduce **ENTP**, which stands for **E**nhancing low-quality SFT data via **N**eural-symbolic **T**ext **P**urge-Mix. Similar to the ENTP personality type from MBTI, **ENTP** is creative in enhancing the low-quality data via purging (noisy information removal) and mixing (with extracted information from all available data and model knowledge). Specifically, the symbolic component identifies and isolates low-quality raw corpora using statistical priors, while the connectionist component extracts latent representations to guide the reconstruction of missing or corrupted information. This synergy generates hybrid instruction-response pairs that augment informational value while preserving corpus diversity. Our experiments demonstrate that fine-tuning LLMs on data augmented by **ENTP**, which are derived solely from low-quality sets, consistently outperforms **13** established data-selection methods across 5 standard instruction-following benchmarks. Notably, it can even surpass fine-tuning on the full original dataset ($\approx$300K examples). Our findings demonstrate that ostensibly low-quality data is a critical resource; leveraging it through intelligent purification and synthesis is key to efficient and effective instruction alignment.

## 1 Introduction

LLMs have demonstrated exceptional performance in a plenty of downstream tasks, ranging from natural language understanding to generative AI applications (Zhang et al., 2024b; Cheng et al., 2024; Tayebi Arasteh et al., 2024; He et al., 2024; Wei et al., 2025b; Biswas & Talukdar, 2024). A pivotal technique that has contributed to enhancing the effectiveness of LLMs is *Supervised Fine-Tuning* (SFT), also known as *Instruction Tuning*. SFT involves further training a pre-trained LLM on a curated dataset comprising instruction-response pairs, aligning the model's responses more closely with human preference or expectations (Wei et al., 2025a; Gupta et al., 2025; Yu et al., 2025). This process bridges the gap between the model's inherent next-word prediction capabilities and the nuanced understanding required for specific tasks.

However, some studies have demonstrated that, during the SFT phase, the quality of data becomes more crucial than the quantity (Zhou et al., 2023). This highlights the importance of high-quality data selection for SFT, which can greatly reduce training costs and improve

Figure 1: Traditional "quality-first" paradigm (a) v.s. our proposed paradigm (b). Part(a) represents the traditional data-selection paradigm, where only the high-quality data is selected (encircled by a green circle); Part (b) illustrates our proposed paradigm, which exploits information from neglected low-quality corpora to generate more expressive synthetic corpora. Part (c): the legend includes Non-regular circles (corpora with varying degrees of imperfections), Regular circles (larger diameters correspond to more information). Each symbol is color-coded to represent a distinct corpus.

efficiency. Earlier studies have designed rule-based systems in conjunction with empirical metrics, such as perplexity and completion length, to perform data filtering (Gao, 2021). Recently, using LLMs as data selectors has become a mainstream paradigm for high-quality data selection (Liu et al., 2024a; Wei et al., 2024; Pang et al., 2025; Zhao et al., 2023). Detailed discussion is given in Appendix A. Although significant attention has been directed toward the extraction of high-quality raw data, a persistent yet underappreciated limitation has emerged. Most publicly accessible data on the internet have already been incorporated into existing datasets, resulting in a scarcity of untapped high-quality raw data. Besides, the high-quality native data within widely used public datasets have largely been exhausted, and such data typically constitute less than 10% of the total dataset volume (Pang et al., 2025; Xia et al., 2024; Li et al., 2024b). Consequently, due to insufficient new data sources, continued reliance on this small subset of high-quality native data for SFT places inherent constraints on further scaling the capabilities of LLMs, revealing a fundamental limitation of the high-quality data selection paradigm. This observation is also supported by experiments from several other studies (Wang et al., 2024).

**Purge + Mix of the Low Quality Data**  In Figure 1, to alleviate the shortage of the high-quality raw data, we propose a novel paradigm (part (b)) that leverages low quality raw corpora, in contrast to the typical paradigm of extracting high quality subsets from raw corpora (part (a)). Specifically, our approach builds on the traditional data selection paradigm, initially partitioning data into high- and low-quality sets via calibrated LLM scores. Subsequently, all low-quality corpora are input into the purification stage, where key representations, such as important terms or potential matching patterns from instruction-response pairs, are extracted. In the following corpora fusion stage, all input representations are integrated into a new synthetic corpus. This new synthetic corpus retains most of the key features from the sourced corpora while also providing additional complementary information, thereby significantly enhancing the expressive capability of each data instance. The final step involves combining the high-quality corpus with the synthetic corpus to form the blended dataset. Our key contributions are summarized as follows:

- **A Novel Paradigm of Corpora Fusion:** We propose **ENTP**, which first extracts predefined knowledge from the input corpus using a set of explicit symbolic rules, and subsequently leverages LLMs to enrich this knowledge with supplementary information, yielding a merged corpus that exhibits substantial informational depth and encapsulates knowledge across multiple dimensions.

- **Empirical Observations:** Extensive experiment results reveal two key findings: (1) Low-quality data makes a non-trivial contribution and should not be overlooked, aligning with the scaling-law conclusion; (2) Fine-tuning 3 representative LLMs on the synthetic dataset surpasses 13 baselines across 5 commonly used benchmarks. These baselines encompass 4 LLM-free approaches, 6 LLM-based methods, as well as native low-quality/high-quality datasets and full-data configurations.

## 2 PRELIMINARY

In this section, we introduce the essential technology underpinning **ENTP**: *Score Transition Matrix*, which estimates the transition probabilities between observed and unseen ground true labels to correct noisy labels. Besides, **ENTP** also builds upon another well-established preliminary, *Average Silhouette Score*, evaluating clustering quality by balancing cohesion and separation. Its technical details are given in the Appendix B.1.

**Score Transition Matrix** Recent studies have demonstrated that LLM-based data-quality assessment suffers from knowledge inconsistency, whereby the identical data may receive different and occasionally vastly divergent scores depending on the LLM employed (Zheng et al., 2024; Pang et al., 2025). To detect and correct potential errors in the raw LLM-generated scores, **ENTP** employs the **Score Transition Matrix** (Zhu et al., 2021), modeling misclassification probabilities under the clusterability condition. This enables error adjustment without ground-truth annotations.

Following the same setup as Pang et al. (2025), our sourced corpora set $D$, composed of $N$ corpus-score pairs, is defined as $D := \{\mathbf{x}_n, \tilde{y}_n\}_{n=1}^N$, where $\mathbf{x}_n$ stands for the embedding vector of the $n^{th}$ corpus generated by the embedding model[1], and $\tilde{y}_n$ represents the corresponding raw LLM-rated score. Meanwhile, $y_n$ denotes the unseen ground-truth score. In our setting, both $\tilde{y}_n$ and $y_n$ are assumed to lie within the same discretized $K$-class classification space $Y$. We have $K = 6$, where all LLM-rated scores span from 0 to 5. The score transition matrix $\mathbf{T}(\mathbf{x})$ is defined as a $K \times K$ square matrix indexed by the feature-space embedding $\mathbf{x}$. Its entry, $\mathbf{T}_{i,j}(\mathbf{x})$, denotes the probability that an unseen ground-true label $i$ is flipped to an observed label $j$. Applying this theory to our problem setting, $\mathbf{T}_{i,j}(\mathbf{x}_n)$ is defined as follows:

$$\mathbf{T}_{i,j}(\mathbf{x}_n) = \mathbb{P}(\tilde{y}_n = j | y_n = i, \mathbf{x}_n), n \in [N], i, j \in [K].$$

*Remark.* The sets $[N] = \{1, 2, ..., N\}$ and $[K] = \{0, 1, ..., K-1\}$ are as above. In the ideal case where $\tilde{y}_n = y_n$ for all $n \in [N]$, $\mathbf{T}(\mathbf{x})$ becomes the identity matrix $\mathbf{I}$, signifying zero misclassification error. Consequently, the deviation of $\mathbf{T}(\mathbf{x})$ from $\mathbf{I}$ quantifies the error rate in the raw LLM-generated scores.

## 3 🧑‍🎓 ENTP: ENHANCING LOW-QUALITY SFT DATA VIA NEURAL-SYMBOLIC TEXT PURGE-MIX

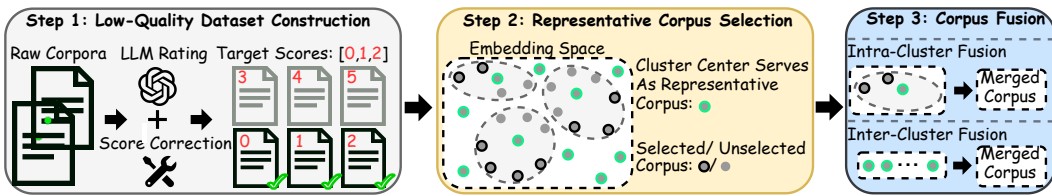

Figure 2: Overview of **ENTP**. **Step (1)** separates the corpora into two subsets based on corrected LLM scores: high-quality (scores 3–5) and low-quality (scores 0–2); **Step (2)** clusters the raw low-quality corpora by inter-corpus similarity and then selects the representative corpora for each cluster; **Step (3)** integrates connectionist and symbolism to fuse corpora through an iterative multi-step process, offering *Intra-Cluster Fusion*, combining representative corpora within the same cluster, and *Inter-Cluster Fusion*, merging those from different clusters; yielding hybrid datasets that preserve diversity while enriching informational value.

We introduce **ENTP**, which consists of: *Low-Quality Dataset Construction, One-Hop Cluster-Based Representative Selection, and Neural-Symbolic Two-to-One Corpora Fusion*, a workflow is given in Figure 2.

---

[1]Hugging Face Embedding Model Used In **ENTP**: BAAI/bge-large-en-v1.5

### 3.1 Step 1: Low-Quality Dataset Construction

We begin by prompting one of the most intelligent LLMs, gpt-4o-mini[2], to assign quality scores to each sample tuple (Instruction, Input, Response). These scores reflect multiple dimensions of interest, such as rarity, complexity, and informativeness. We adopt the prompt template from DS[2] (Pang et al., 2025), where the complete prompt is included in the Appendix C.1 for reference.

**LLM-Rating Score Correction** Because the LLM-generated ratings often suffer from inaccuracy and inconsistency, we integrate a rating correction step inspired by Zhu et al. (2021):

**Theorem 1. (K-NN Score Clusterability)** *Sourced Corpora D satisfies K-NN Score Clusterability if $\forall n$, the embedding vector $\mathbf{x}_n$ and its k-Nearest Neighbors $\mathbf{x}_{n_1}, ..., \mathbf{x}_{n_k}$ belong to the same ground-truth class.*

*Remark.* Although $\mathbf{T}$ cannot be computed directly due to inaccessibility to the ground-truth scores, an effective estimation method is provided by Zhu et al. (2021).

**Score Transition Matrix & Consensus Vectors** Our objective can be cast as a $K$-class classification task, where, given the raw corpus and the corresponding LLM-generated score, the goal is to determine which ground-truth score should be assigned. The probability distribution of the ground truth score is defined as $\mathbf{p} := [\mathbb{P}(\tilde{y}_n), n \in [N], i \in [K]]^T$, and the score transition matrix is given by $\mathbf{T}_g = \mathbf{T} \cdot \mathbf{H}_g, \forall g \in [K]$, where $\mathbf{H}_g := [\mathbf{e}_{g+1}, ..., \mathbf{e}_K, \mathbf{e}_1, ..., \mathbf{e}_g]$ is a cyclic permutation matrix. $\mathbf{e}_g$ denotes a $K \times 1$ column vector with a 1 in the $g$-th position and 0s elsewhere. The matrix $\mathbf{H}_g$ cyclically shifts each column of $\mathbf{T}$ to the left by $g$ positions. We define $(i + g)_K := [(i + g - 1) \mod K] + 1$ as the index resulting from a cyclic shift by $g$ positions within a range of size $K$. Therefore, the corresponding first-, second-, and the third-order consensus vectors are defined as follows:

$$\mathbf{q}^{[1]} := [\mathbb{P}(\tilde{y}_1 = i), \ i \in [K]]^T = \mathbf{T}^T \mathbf{p} \ ,$$

$$\mathbf{q}_z^{[2]} := [\mathbb{P}(\tilde{y}_1 = i, \ \tilde{y}_2 = (i + z)_K), \ i \in [K]]^T = (\mathbf{T} \odot \mathbf{T}_z)^T \mathbf{p} \ ,$$

$$\mathbf{q}_{z,g}^{[2]} := [\mathbb{P}(\tilde{y}_1 = i, \ \tilde{y}_2 = (i + z)_K, \ \tilde{y}_3 = (i + g)_K), \ i \in [K]]^T = (\mathbf{T} \odot \mathbf{T}_z \odot \mathbf{T}_g)^T \mathbf{p} \ , \quad (1)$$

where $\tilde{y}_{1-3}$ denote the LLM-rated scores for three embedding vectors $\mathbf{x}_1, \mathbf{x}_2, \mathbf{x}_3$. $\mathbf{x}_2$ and $\mathbf{x}_3$ are top-2 nearest neighbors of $\mathbf{x}_1$ in the embedding space. These consensus vectors capture the probability that neighboring embedding features are assigned identical scores, effectively embedding the score transition dynamics into the measure of score agreement. More importantly, all consensus vectors can be estimated by analyzing the distribution of raw LLM-rated scores. Eq. 1 provides the theoretical foundation for inferring $\mathbf{T}$ and $\mathbf{p}$ from the estimated consensus vectors (Zhu et al., 2021). Liu et al. (2023) and Zhu et al. (2021) further established that, under the third-order consensus vector setting, the problem can be effectively solved to recover accurate estimates of $\mathbf{T}$ and $\mathbf{p}$. With $\mathbf{T}$ and $\mathbf{p}$ estimated, we can straightforwardly apply Bayes' theorem (Joyce, 2003) to infer the most likely ground-truth score conditioned on the observed LLM-rated score and the input corpus.
Drawing inspiration from the LLM-based data selection (Pang et al., 2025) and heuristic noisy data identification (Zhu et al., 2021), this step integrates these two paradigms, starting with LLM Rating and Score Correction, to derive corrected overall scores, denoted as $s^o$, ranging from 0 to 5 for all raw samples. As depicted in Figure 2 (Step 1), the raw dataset is partitioned into two subsets based on these scores: the low-quality set defined as $\mathbf{S}_{lq} = \{s^o | s^o \in [0, 1, 2]\}$, and the high-quality set defined as $\mathbf{S}_{hq} = \{s^o | s^o \in [3, 4, 5]\}$.

### 3.2 Step 2: One-Hop Cluster-Based Representative Selection

After isolating the low-quality subset, this step selects a representative subset of observations that best captures its defining characteristics. As shown in Figure 2 (Step 2), we randomly select a corpus as the cluster centroid, compute cosine similarities with all other corpora, and assign to the same cluster any corpus with a similarity of at least 0.9. This

---

[2]gpt-4o-mini-2024-07-18

forms a *One-Hop Cluster*, where each member is within one similarity-threshold "hop" of the centroid. Such clusters capture latent similarities and preserve inter-data diversity, enhancing representation without over-homogenization.

Next, we apply k-means clustering (MacQueen, 1967) to partition each initial cluster into sub-clusters, determining the optimal number of sub-clusters (k) by evaluating clustering quality over a predefined range of k values using the Silhouette method (Rousseeuw, 1987), where its detailed calculation is provided in the Appendix C.2. For any initial cluster comprising at least two sub-clusters of three or more corpora, two representative corpora are chosen from each sub-cluster. The first representative, $r_0$, is selected based on the highest cosine similarity to the averaged embedding vector of the sub-cluster. To promote diversity, we adopt a *Maximal Marginal Relevance*-inspired scoring function (MMR Score) parameterized by $\alpha$ (Carbonell & Goldstein, 1998). Once $r_0$ is determined, we compute the MMR score for each remaining corpus within the sub-cluster. The entire procedure can be computed as follows:

$$\text{Given: } r_{\text{avg}} = \frac{1}{N} \sum_{r_a \in \mathbf{R}} r_a(.), \ N = |\mathbf{R}|; \ r_0 = \arg\max_{r_a \in \mathbf{R}} \text{Sim}(r_a, r_{\text{avg}}); \ \mathbf{S} = \{r_0\}$$

$$\Rightarrow \text{MMR Score}(r_i) = \arg\max_{r_i \in \mathbf{R} \setminus \mathbf{S}} \Big[ \alpha \, \text{Sim}(r_i, r_{\text{avg}}) \ - \ (1 - \alpha) \max_{r_j \in \mathbf{S}} \text{Sim}(r_i, r_j) \Big],$$

where $\mathbf{R}$ and $\mathbf{S}$ denote the candidate set comprising all corpora from the same sub-cluster and the selected set of representative corpora (with first element $r_0$), respectively; $N$ is the cardinality of the set $\mathbf{R}$ (i.e., the number of its elements); and $r_{\text{avg}}$ stands for the component-wise mean (centroid) vector computed over all vectors in the candidate set $\mathbf{R}$. To compute the MMR score for selecting additional representatives, $r_i \in R \setminus S$ signifies that $r_i$ lies in the set-difference of $R$ and $S$ — that is, it is a member of the candidate set $R$ but has not yet been selected into $S$. $\text{Sim}(\cdot, \cdot)$ represents the cosine-similarity operation, and $\alpha \in [0, 1]$ is the weighting parameter that controls the trade-off between relevance and diversity — smaller values of $\alpha$ place greater emphasis on selecting corpora that lie farther from the cluster center.

In the alternative scenario, when a resulting sub-cluster contains fewer than three vectors, all vectors in that sub-cluster are selected as representative vectors. We provide Algorithm 1 in the Appendix C.2.

### 3.3 STEP 3: NEURAL-SYMBOLIC TWO-TO-ONE CORPUS FUSION

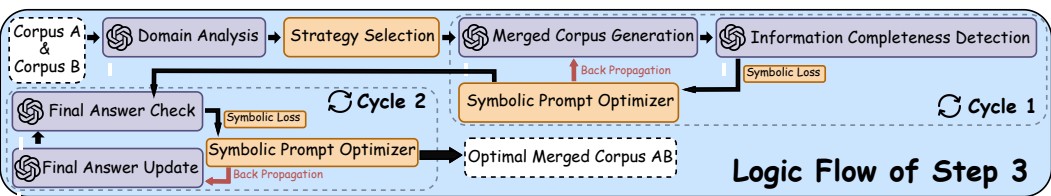

Figure 3: Logic flow of Step 3: all purple blocks represent the connectionist components, corresponding to different LLM-invoking operators, while all orange blocks stand for the symbolic components, involving the utilization of symbolic rules. Step 3 effectively combines the generalization capability of connectionism with the explicit symbolic rule, thereby achieving the purification, and fusion of the low-quality corpus.

As shown in Figure 3, the black arrow denotes the forward pass, where each module's output feeds into the next in sequence. The red arrow signifies back-propagation: the prompt template modified in the later Symbolic Prompt Optimizer is propagated back to the earlier Merged Corpus Generation/ Final Answer Check operator to update the corresponding content of the merged corpus. The complete process is comprised of a preparation step and two sequential cycles. Full details are provided in the Appendix C.3.

**Preparation Step**   We first provide the LLM with the prompt template $\mathcal{P}_{\text{DA}}$ to perform domain analysis of the input corpus pair, after which their relationship is classified as same-, related-, or unrelated-domain. To generate the merged corpus, we provide the LLM with

nine relation-dependent strategies inspired by writing-studies literature (Nelson & King, 2023; Knobel, 2017; Bazerman, 2003), rather than letting it autonomously search for an optimal fusion paradigm. Relying solely on the LLM's internal priors greatly increases reasoning time and cost, whereas supplying external, stable prior knowledge narrows the search space, reduces computation, and accelerates convergence toward human-preferred outcomes. For each relation category, three natural-language fusion strategies incorporating prior knowledge are provided, yielding $\mathbf{S} = \{\mathcal{S}_{\text{same}}, \mathcal{S}_{\text{rel}}, \mathcal{S}_{\text{unrel}}\}$. Hence, $\mathcal{F}(\mathcal{P}_{\text{DA}}(\mathcal{C}_{\text{A}}, \mathcal{C}_{\text{B}})) = \mathcal{S}$, $\mathcal{S} \in \mathbf{S}$, where $\mathcal{F}(.)$ denotes the LLM operator, and $C_{\text{A}}, C_{\text{B}}$ stand for the input corpus pair.

**Cycle 1** After obtaining the fusion strategies from the preparation step, we generate the initial merged corpus via the Merged Corpus Generation (MCG) operator, which serves as the starting point of the first cycle. This corpus is then passed to the Information Completeness Detection (ICD) operator, with prompt template $\mathcal{P}_{\text{ICD}}$, to compute the symbolic loss $\mathcal{L}_{\text{Sym}}$. The symbolic loss, essentially a JSON object, specifies which information in the current merged corpus should be removed or retained. It is subsequently provided to the Symbolic Prompt Optimizer (SPO) to update the prompt template $\mathcal{P}_{\text{MCG}}$ for the MCG operator in the next iteration, marking the end point of the entire iteration. This cycle is dedicated to generating the optimal "### User" session[3]. Thus, the complete first cycle is formulated as:

$$\mathcal{F}(\mathcal{P}_{\text{MCG}}^{\text{i}}(\mathcal{C}_{\text{A}}, \mathcal{C}_{\text{B}}, \mathcal{S})) = \mathcal{C}_{\text{AB}}^{\text{i}} \Rightarrow \mathcal{F}(\mathcal{P}_{\text{ICD}}(\mathcal{C}_{\text{AB}}^{\text{i}})) = \mathcal{L}_{\text{Sym}} \Rightarrow \text{SPO}(\mathcal{P}_{\text{MCG}}^{\text{i}}, \mathcal{L}_{\text{Sym}}) = \mathcal{P}_{\text{MCG}}^{\text{i+1}} \ .$$
$$\Rightarrow \text{The } (\text{i}+1)^{\text{th}} \text{ iteration}: \ \text{Starting From} \ \ \mathcal{F}(\mathcal{P}_{\text{MCG}}^{\text{i+1}})$$

*Remark.* $\mathcal{C}_{\text{AB}}^{\text{i}}$ indicates the $\text{i}^{\text{th}}$ generated merged corpus from the source corpora; $\mathcal{P}_{\text{MCG}}^{\text{i+1}}$ represents the updated prompt template for the MCG operator used in the next iteration.

Consequently, the task of determining the optimal merged corpus in this cycle can be framed as an optimization problem, where the objective is to identify the optimal prompt template $\mathcal{P}_{\text{MCG}}^{*}$ that minimizes the symbolic loss of the finalized merged corpus $\mathcal{C}_{\text{AB}}$. This can be mathematically expressed as follows:

$$\mathcal{P}_{\text{MCG}}^{*} = \underset{\mathcal{P}_{\text{MCG}}}{\arg\min} \ \mathcal{L}_{\text{Sym}} = \underset{\mathcal{P}_{\text{MCG}}}{\arg\min} \ \mathcal{F}(\mathcal{P}_{\text{ICD}}(\mathcal{F}(\mathcal{P}_{\text{MCG}}))) \ \ \Rightarrow \mathcal{C}_{\text{AB}} = \mathcal{F}(\mathcal{P}_{\text{MCG}}^{*}) \ .$$

**Cycle 2** Once this optimal "### User" session is determined, its corresponding "### Assistant" session[3] is fed into the second cycle, beginning with the evaluation of the final answer driven by the Final Answer Check (FAC) operator using prompt template $\mathcal{P}_{\text{FAC}}$. The symbolic loss corresponding to the current answer content is then produced by the FAC operator and input to the SPO operator. As in Cycle 1, the prompt template for the Final Answer Update (FAU) operator, $\mathcal{P}_{\text{FAU}}$, is updated to revise the current answer content, marking the end of this iteration and preparing for the next. Therefore, the entire Cycle 2 is formulated as:

$$\mathcal{F}(\mathcal{P}_{\text{FAC}}(\mathcal{C}_{\text{c1}}^{\text{i}})) = \mathcal{L}_{\text{Sym}} \Rightarrow \text{SPO}(\mathcal{P}_{\text{FAU}}^{\text{i-1}}, \mathcal{L}_{\text{Sym}}) = \mathcal{P}_{\text{FAU}}^{\text{i}} \Rightarrow \mathcal{F}(\mathcal{P}_{\text{FAU}}^{\text{i}}(\mathcal{C}_{\text{c1}}^{\text{i}})) = \mathcal{C}_{\text{c1}}^{\text{i+1}} \ .$$
$$\Rightarrow \text{The } (\text{i}+1)^{\text{th}} \text{ iteration}$$

*Remark.* $\mathcal{C}_{\text{c1}}^{\text{i}}$ denotes the optimal merged corpus generated from the Cycle 1, equivalent to $\mathcal{C}_{\text{AB}}$ when $\text{i} = 1$. Additionally, $\mathcal{P}_{\text{FAU}}^{0}$ denotes the initial prompt template for the FAU operator. Similar to the Cycle 1, determining the optimal answer is posed as finding the prompt template $\mathcal{P}_{\text{FAU}}^{*}$ that minimizes the symbolic loss of the finalized answer from $\mathcal{C}_{\text{c1}}$, expressed as:

$$\mathcal{P}_{\text{FAU}}^{*} = \underset{\mathcal{P}_{\text{FAU}}^{\text{i}}}{\arg\min} \ \mathcal{L}_{\text{Sym}} = \underset{\mathcal{P}_{\text{FAU}}^{\text{i}}}{\arg\min} \ \mathcal{F}(\mathcal{P}_{\text{FAC}}(\mathcal{F}(\mathcal{P}_{\text{FAU}}^{\text{i-1}}))) \ \ \Rightarrow \mathcal{C}_{\text{c1}} = \mathcal{F}(\mathcal{P}_{\text{FAU}}^{*}) \ .$$

This implies that the optimal $\mathcal{P}_{\text{FAU}}^{*}$ corresponds to the prompt template from the previous iteration, as this cycle updates the answer for iteration $\text{i}+1$ using the loss from iteration

---

[3]**ENTP**-generated merged corpus consists of paired "### User" session (containing all the necessary context and the relevant question) and "### Assistant" session (containing the corresponding answer).

i. Overall, $\mathcal{C}_{c1}$ encompasses both the optimal final answer derived from the current cycle and the corresponding optimal question from the preceding cycle, collectively representing a valid merged corpus generated by **ENTP**. Full Algorithm 2 is given in Appendix C.3, and a comprehensive workflow is depicted in Figure 5 (Appendix C.3.2).

## 4 EXPERIMENTS

Table 1: Sourced Corpora Components

| Datasets | Stanford Alpaca | Flan V2 | Open-Assistant 1 | WizardLM | Dolly | Overall |
|---|---|---|---|---|---|---|
| **Data Size** | 52K | 100K | 33K | 100K | 15K | 300K |

### 4.1 EXPERIMENTAL SETUP

**Source Corpora**  We select different proportions of five instruct-following datasets as the source corpora in **ENTP**, including Stanford Alpaca (Taori et al., 2023), Flan_v2 (Longpre et al., 2023), Open Assistant 1 (Köpf et al., 2023), and WizardLM (Xu et al., 2024b), Dolly (Conover et al., 2023). Complete statistics of our sourced corpora are provided in Table 1. Additional details of data pool are listed in Appendix D.1.1.

**Evaluation Dataset & Metrics**  In order to demonstrate the validity of our merged corpora, we adopt five tasks from the OpenLLM Leaderboard as benchmarks for evaluation: MMLU (Hendrycks et al., 2020), TruthfulQA (Lin et al., 2021), GSM8K (Cobbe et al., 2021), BBH (Suzgun et al., 2022), and TyDiQA (Clark et al., 2020). For MMLU, TruthfulQA, GSM8K, and BBH, we use Exact Match (EM) as the evaluation metric. For TyDiQA, we report the 1-shot F1 score. Comprehensive details about our evaluation benchmarks are presented in Appendix D.1.2.

**Base Models & Rating Model**  We adopt three widely recognized open-source LLMs as our base models: Qwen2.5-7B (Team, 2024), LLaMA-3.1-8B (Grattafiori et al., 2024), and Mistral-7B-v0.3 (Jiang et al., 2023). These models are fine-tuned on datasets derived from various baseline methods, with the aim of evaluating the performance of **ENTP**. In addition, we use gpt-4o-mini[2] as our rating model.

**Baselines**  The full set given by **ENTP** consists of 54888 samples drawn from the LQ Set (123786 samples): 15488 samples are obtained via intra-cluster fusion, and 39400 via inter-cluster fusion. We adopt 13 representative data-selection methods for comparison, applying each to the LQ Set for fair comparison, including: ① *Vanilla Base Model*; ② *LQ Set & HQ Set* represent the low/high-quality set (samples with the curated score in range [0, 2]/[3, 5]), obtained from *Full Set*; ③ *Full Set* ($\approx$ 300K samples); ④ *Completion Length* utilizes the length of the whole corpus as an indicator to assess to sample quality; ⑤ $\text{KNN}_i$ is defined as the Average Euclidean Distance from each raw embedding vector to its $i$ nearest neighbors; ⑥ *Perplexity*; ⑦ *Random Selection*; ⑧ *AlpaGasus (Random)* (Chen et al., 2023) employs gpt-4o-mini[2] to score each sample and retains only the highest-rated samples for fine-tuning; and: ⑨ *IFD* (Li et al., 2024b); ⑩ *Superfiltering* (Li et al., 2024a); ⑪ *DEITA* (Liu et al., 2024b); ⑫ *RDS+ & RDS+ (best)* (Ivison et al., 2025); ⑬ $\text{DS}^2$ (Pang et al., 2025); ⑭ *LESS* (Xia et al., 2024); ⑮ *MathFusion* (Pei et al., 2025); ⑯ *Evol-Instruct* (Xu et al., 2024b); ⑰ *Self-Instruct* (Wang et al., 2023b); ⑱ *1-to-1 Rewriting/ Enhancement*; ⑲ *Direct Corpora Fusion Without Step 2&3*; ⑳ *Direct Corpora Fusion Without Step 3*. Comprehensive details of all baselines are provided in the Appendix D.1.3.

**Implementation Details.**  In the one-hop clustering stage, **ENTP** sets the cosine similarity threshold to 0.9. For representative corpus selection, we set $\alpha = 0.2$ to encourage diversity. The gpt-4o-mini[2] model is used as the API-accessed LLM in **ENTP**, with temperature set to 0.4 during the DA operator to encourage broader exploration, and 0.2 for all other modules to ensure consistency. In the two-to-one corpus fusion step, we propose two configurations: intra-cluster fusion, where multiple corpora from the same cluster are progressively merged until a single representative corpus is obtained; and inter-cluster fusion, where two corpora from different clusters are merged in a single pass. Besides, we limit regeneration attempts to 3.

Table 2: **Performance comparison on the OpenLLM leaderboard.** The default data size is 54888. The fine-tuning base model is Mistral-7B-v0.3. Best and second-best results on average are highlighted in **bold red** and **bold black**, respectively. Performance changes of **ENTP** w.r.t. the LQ Set across all benchmarks are also reported.

| Dataset | MMLU (EM ↑) | TruthfulQA (EM ↑) | GSM8K (EM ↑) | BBH (EM ↑) | TydiQA (1-Shot F1 ↑) | Average ↑ |
|---|---|---|---|---|---|---|
| | | Base Model: Mistral-7B-v0.3 | | | | |
| Vanilla Base Model | 59.7 | 30.4 | 38.0 | 47.6 | 54.8 | 46.1 |
| LQ Set (123786) | 47.5 | 43.7 | 43.5 | 52.7 | 41.1 | 45.7 |
| HQ Set (131247) | 58.4 | 39.2 | 46.0 | 55.5 | 52.5 | 50.3 |
| Full Set (300932) | 60.0 | 43.5 | 43.5 | 52.5 | 53.4 | **50.6** |
| Completion Length | 25.4 | 43.5 | 43.0 | 55.7 | 45.8 | 42.7 |
| $KNN_{10}$ | 53.7 | 43.7 | 40.0 | 54.4 | 39.5 | 46.3 |
| Perplexity | 53.8 | 41.8 | 34.5 | 54.8 | 41.9 | 45.4 |
| Random Selection | 52.8 | 42.0 | 41.5 | 56.7 | 48.4 | 48.3 |
| AlpaGasus (Random) | 53.8 | 48.5 | 44.5 | 55.6 | 41.5 | 48.8 |
| IFD | 40.3 | 43.8 | 44.0 | 49.5 | 33.6 | 42.2 |
| Superfiltering | 51.8 | 40.7 | 45.0 | 52.6 | 37.8 | 45.6 |
| DEITA | 44.5 | 39.9 | 43.5 | 50.2 | 46.1 | 44.8 |
| DEITA (Our Curated Score) | 52.2 | 36.6 | 44.0 | 54.3 | 51.7 | 47.8 |
| RDS+ | 47.9 | 41.1 | 43.0 | 52.9 | 41.8 | 45.3 |
| RDS+ (Best) | 51.0 | 43.4 | 46.0 | 54.9 | 44.6 | 48.0 |
| $DS^2$ | 48.7 | 44.1 | 47.5 | 55.1 | 46.9 | 48.5 |
| LESS | 54.1 | 46.2 | 44.0 | 53.8 | 50.5 | 49.7 |
| MathFusion | 50.8 | 59.6 | 44.5 | 52.8 | 41.4 | 49.8 |
| Evol-Instruct | 54.0 | 57.5 | 33.5 | 53.1 | 42.8 | 48.2 |
| Self-Instruct | 53.1 | 43.8 | 45.0 | 55.2 | 50.9 | 49.6 |
| 1-to-1 Rewriting/ Enhancement | 47.3 | 42.4 | 41.5 | 49.4 | 49.9 | 46.1 |
| Direct Corpora Fusion Without Step2&3 | 40.4 | 41.4 | 37.5 | 48.8 | 50.7 | 43.8 |
| Direct Corpora Fusion Without Step3 | 45.9 | 42.8 | 40.0 | 50.0 | 50.3 | 45.8 |
| ENTP | 58.6 (+11.1) | 43.0 (-0.7) | 44.0 (+0.5) | 53.8 (+1.1) | 58.3 (+17.2) | **51.5 (+5.8)** |

Table 3: **Performance comparison on the OpenLLM leaderboard.** The default data size is 54888. The fine-tuning base model is Llama-3.1-8B. Best and second-best results on average are highlighted in **bold red** and **bold black**, respectively. Performance changes of **ENTP** with respect to the LQ Set across all benchmarks are also reported.

| Dataset | MMLU (EM ↑) | TruthfulQA (EM ↑) | GSM8K (EM ↑) | BBH (EM ↑) | TydiQA (1-Shot F1 ↑) | Average ↑ |
|---|---|---|---|---|---|---|
| | | Base Model: Llama-3.1-8B | | | | |
| Vanilla Base Model | 64.1 | 32.9 | 58.0 | 55.3 | 22.1 | 46.5 |
| LQ Set (123786) | 52.7 | 44.3 | 57.0 | 61.0 | 43.9 | 51.8 |
| HQ Set (131247) | 62.3 | 41.8 | 57.5 | 59.3 | 58.9 | 56.0 |
| Full Set (300932) | 63.5 | 42.0 | 61.0 | 59.1 | 62.8 | **57.7** |
| Completion Length | 59.5 | 45.8 | 60.0 | 58.6 | 61.2 | 57.0 |
| $KNN_{10}$ | 59.3 | 37.6 | 52.0 | 59.5 | 55.7 | 52.8 |
| Perplexity | 60.5 | 37.5 | 51.0 | 57.8 | 55.0 | 52.4 |
| Random Selection | 60.2 | 38.0 | 57.0 | 57.7 | 60.4 | 54.7 |
| AlpaGasus (Random) | 53.0 | 44.1 | 56.0 | 59.4 | 49.0 | 52.3 |
| IFD | 45.9 | 46.7 | 57.0 | 56.6 | 42.8 | 49.8 |
| Superfiltering | 58.0 | 43.8 | 61.5 | 56.5 | 51.0 | 54.2 |
| DEITA | 57.6 | 43.3 | 58.5 | 59.0 | 60.8 | 55.8 |
| DEITA (Our Curated Score) | 60.0 | 46.8 | 58.0 | 58.1 | 61.3 | 56.8 |
| RDS+ | 57.1 | 43.6 | 52.5 | 58.6 | 42.0 | 50.8 |
| RDS+ (Best) | 57.1 | 46.6 | 59.5 | 60.8 | 53.0 | 55.4 |
| $DS^2$ | 59.9 | 44.8 | 55.5 | 58.2 | 60.8 | 55.8 |
| LESS | 59.9 | 40.5 | 56.0 | 61.4 | 66.2 | 56.8 |
| Self-Instruct | 52.1 | 18.3 | 56.5 | 57.9 | 57.3 | 48.4 |
| 1-to-1 Rewriting/ Enhancement | 58.3 | 42.5 | 58.5 | 57.9 | 46.1 | 52.7 |
| Direct Corpora Fusion Without Step2&3 | 54.1 | 40.7 | 60.0 | 56.9 | 43.1 | 51.0 |
| Direct Corpora Fusion Without Step3 | 54.5 | 48.8 | 56.0 | 58.3 | 39.6 | 51.4 |
| ENTP | 61.7 (+9.0) | 47.8 (+3.5) | 54.5 (-2.5) | 60.7 (-0.3) | 61.3 (+17.4) | **57.2 (+5.4)** |

## 4.2 EMPIRICAL OBSERVATIONS

All observations reported in this section stem from experiments conducted with the Mistral-7B-v0.3 and Llama-3.1-7B models. Additional findings based on various base models are provided in the Appendix D.2.

**Observation 1: A structural bottleneck in the classical data-selection paradigm progressively emerges.** In Table 2-3, all data-selection baselines, LLM-free or LLM-based, and regardless of whether they leverage a validation split from the test set, exhibit average performance that oscillates around the results obtained with the LQ Set: (1) For Mistral-7B-v0.3, average performance fluctuates near 45.7, with values spanning from 42.7 (Completion Length) to 48.8 (AlpaGasus (Random)); (2) For Llama-3.1-8B, performance centers near 51.8, ranging from 49.8 (IFD) to 57.0 (Completion Length). The majority of baselines produce results that differ only marginally. Hence, our experiments pinpoint a

structural bottleneck in this paradigm: once the most informative subset is extracted from the source pool, further gains become unattainable.

**Observation 2: LQ Set does contain the valuable sample which can contribute to the average performance.** As shown in Table 3, two score-aware baselines, Completion Length and DEITA using our curated scores, achieve average scores of 57.0% and 56.8% respectively. Both outperform the HQ Set configuration (56%), which advocates discarding the whole LQ Set. Hence, relying solely on a small portion of native high-quality data, while discarding the majority of native low-quality data, risks losing valuable information that may enhance model performance.

**Observation 3: Our proposed paradigm offers a viable alternative to the classical data-selection paradigm.** As shown in Tables 2-3, regardless of what base model equipped with, **ENTP** consistently outperform all baselines on average that follow the traditional data-selection paradigm, which extracts an optimal subset from the LQ Set. Specifically, when equipping with the Mistral-7B-v0.3 model, on average **ENTP** achieves superior performance over all baselines, including the Full Set configuration (see Table 2). When switching to the Llama-3.1-8B model, **ENTP** achieves the second-highest average performance among all baselines, trailing only the Full Set setting (see Table 3). In comparison with the source dataset (LQ Set), the main improvements of **ENTP** are reflected on two benchmarks: on MMLU and TyDiQA. With Mistral-7B-v0.3, **ENTP** achieves gains of 11.1% on MMLU and 17.2% on TyDiQA; with Llama-3.1-7B, the improvements are 9.0% (MMLU) and 17.4% (TyDiQA). In terms of overall performance, **ENTP** improves by 5.8% when using Mistral-7B-v0.3, and by 5.4% when using Llama-3.1-8B; in both cases, it outperforms all optimal subsets drawn from the LQ Set. Therefore, all empirical results demonstrate that **ENTP** could overcome the bottleneck inherent in the paradigm of relying solely on native, high-quality data.

## 4.3 Ablation Study

Table 4: **Performance comparison among the LQ Set, HQ Set, Full Set, and various proportions of the ENTP -generated dataset.** The fine-tuning base models are Mistral-7B-v0.3 and Llama-3.1-8B. Best and second-best results on average are highlighted in **bold red** and **bold black**, respectively. The average performance changes of **ENTP**, relative to the LQ Set, are also reported.

| Dataset | MMLU (EM ↑) | GSM8K (EM ↑) | BBH (EM ↑) | TydiQA (1-Shot F1 ↑) | Average ↑ |
|---|---|---|---|---|---|
| Base Model: Mistral-7B-v0.3 | | | | | |
| LQ Set (123786) | 47.5 | 43.5 | 52.7 | 41.1 | 46.3 |
| HQ Set (131247) | 58.4 | 46.0 | 55.5 | 52.5 | 53.1 |
| Full Set (300932) | 60.0 | 43.5 | 52.5 | 53.4 | 52.4 |
| **ENTP**-20% | 59.3 | 41.5 | 54.6 | 55.4 | 52.7 (+6.4) |
| **ENTP**-40% | 58.9 | 42.0 | 50.5 | 56.8 | 52.1 (+5.8) |
| **ENTP**-60% | 59.1 | 45.5 | 52.3 | 56.2 | 53.3 (+7.0) |
| **ENTP**-80% | 58.5 | 44.0 | 53.8 | 57.4 | **53.4 (+7.1)** |
| **ENTP**-100% | 58.6 | 44.0 | 53.8 | 58.3 | **53.7 (+7.4)** |
| Base Model: Llama-3.1-8B | | | | | |
| LQ Set (123786) | 52.7 | 57.0 | 61.0 | 44.7 | 53.9 |
| HQ Set (131247) | 62.3 | 57.5 | 59.3 | 58.9 | 59.5 |
| Full Set (300932) | 63.5 | 61.0 | 59.1 | 62.8 | **61.6** |
| **ENTP**-20% | 63.9 | 57.5 | 61.5 | 52.5 | 58.9 (+5.0) |
| **ENTP**-40% | 62.3 | 56.5 | 58.0 | 56.5 | 58.3 (+4.4) |
| **ENTP**-60% | 62.3 | 57.5 | 60.1 | 57.2 | 59.3 (+5.4) |
| **ENTP**-80% | 62.0 | 56.0 | 61.5 | 57.8 | 59.3 (+5.4) |
| **ENTP**-100% | 61.7 | 54.5 | 60.7 | 61.3 | **59.6 (+5.7)** |

**Ablation Setup** To gain a more comprehensive understanding of how **ENTP**-generated dataset affects the performance of LLMs, we employ the LQ Set, HQ Set and Full Set as control groups. For the experimental groups, we evaluate five configurations of the full **ENTP** -generated dataset, ranging from 20% to 100%, denoted as **ENTP**-x%, where x% indicates the random selection of x% of the merged corpus obtained via intra-cluster and inter-cluster fusion.

**Empirical Scaling Law Holds For ENTP-Generated Data: Full Dataset Ourper-forms All Subsets**  We experiment with subsets of varying volumes (20%–100%) of the full **ENTP** -generated dataset to systematically assess scaling behavior. As shown in Table 4, regardless of the options of the base model, as the dataset size increases, average performance also exhibits an upward trend, consistent with empirical scaling laws. Moreover, across configurations ranging from 20% to 100% of our merged corpus, each **ENTP**-based setting outperforms the source LQ Set, demonstrating the effectiveness of **ENTP**. More ablation studies using different base model with various experimental setup are given in the Appendix E.

## 5 Conclusion

We introduced **ENTP**, re-examining the long-held "quality-first" dogma in supervised fine-tuning. Rather than discarding the vast pool of low-score or head-frequency instruction data, **ENTP** purges the genuinely noisy elements, mixes the remaining signal with model-generated knowledge, and delivers a topic-focused corpus that is both compact and information-rich. Empirically, LLMs fine-tuned on **ENTP**-created corpora consistently outperformed models trained on the full 300K dataset or on conventional "high-quality" subsets across five instruction-following benchmarks. In addtion, our empirical results yield two key insights: (1) **Hidden value in low-quality data.** Even ostensibly poor examples contain complementary information that, when properly distilled, improves downstream performance—corroborating scaling-law observations that "more diverse data" can be as valuable as "better data." (2) **Neural-symbolic fusion is effective for corpus construction.** Symbolic rules provide reliable noise filters, while connectionist models enrich and complete missing content, jointly producing a superior training signal.

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

## Appendix

**Use of Large Language Models**   In our **ENTP**, the LLM is designed as an atomic capability and primarily contributes to the data quality evaluation and data fusion stages.

**Data and Code Availability**   We will release the implementation code of **ENTP**, along with all merged corpora sets used in our experiments, including those generated via intra-cluster and inter-cluster fusion, upon acceptance of the paper.

The rest of Appendix is organized as follows:

- Section A: We give detailed discussions about related work.
- Section B: We provide comprehensive technical details of an additional preliminary component employed by **ENTP**.
- Section C: We provide comprehensive details about **ENTP**.
- Section D: We include omitted experiment details, designs, additional experiment results, and GPU Runtime & API Cost & Validation Set Requirement Analysis.
- Section E: We include a supplementary ablation study.
- Section F: We provide the curated overall score distributions for both the original low-quality corpora and the **ENTP**-generated merged corpora.
- Section G: We provide a concrete end-to-end example.
- Section H: We present several **ENTP**-generated samples.
- Section I: We provide potential future directions.

## A    Related Work

Data selection paradigms can be broadly classified into two categories: those that rely solely on empirical metrics (without LLM involvement) and those that incorporate LLMs.

**Data Selection Without LLM**   Sabbineni et al. (2023) introduced entropy and Error L2-Norm (EL2N) scores to evaluate the "usefulness" or "difficulty" of data examples, demonstrating that score-based selection can reduce semantic error rates and domain classification errors compared to random selection. Xie et al. (2023) extended the classic importance resampling method to high-dimensional settings, proposing the Data Selection with Importance Resampling (DSIR) framework. DSIR estimates importance weights in a reduced feature space and selects data accordingly, achieving significant improvements in downstream tasks such as GLUE. Wainer and Cawley (2017) conducted an extensive empirical evaluation of 15 resampling procedures for Support Vector Machine (SVM) hyperparameter selection, concluding that a 2-fold procedure is appropriate for datasets with 1000 or more data points, while a 3-fold procedure is suitable for smaller datasets.

**LLM-based Data Selection**   Li et al. (2024b) introduced the Instruction-Following Difficulty (IFD) metric, enabling LLMs to autonomously identify challenging instruction-response pairs by measuring discrepancies between expected and actual responses, thereby enhancing model performance with a reduced dataset. Lu et al. (2023) developed the IN-STAG framework, leveraging fine-grained tagging of instruction semantics to select diverse and complex examples, which improved instruction-following capabilities. Additionally, Liu et al. (2024b) employed a comprehensive analysis combining diversity, quality, and complexity metrics to systematically select high-performing data subsets, demonstrating significant improvements in model robustness.

As previously noted, this paradigm overlooks the potential contributions of low-quality data, leading methods that adhere to it to inevitably encounter bottlenecks due to the scarcity of high-quality raw data. In contrast, **ENTP** maximizes the potential of each low-quality corpus, transforming them into rare and expressive synthetic corpora.

## B  More Preliminary

### B.1  Average Silhouette Score

Clustering quality hinges on both how tightly points group within their own clusters (cohesion) and how well they separate from other clusters (separation). The Silhouette Score uniquely captures both dimensions in a single metric, enabling an immediate, interpretable gauge of cluster validity (Rousseeuw, 1987). Mathematically, for each data point $i$ assigned to cluster $C_I$, the cohesion $a(i)$ and separation $b(i)$ are defined as follows:

$$a(i) = \frac{1}{|C_I| - 1} \sum_{j \in C_I, j \neq i} d(i, j),$$

$$b(i) = \min_{J \neq I} \frac{1}{|C_J|} \sum_{j \in C_J} d(i, j),$$

where $d(\cdot, \cdot)$ represents the Euclidean distance; $a(i)$ is computed as the average distance between point $i$ and all other members of its own cluster; $b(i)$ denotes the minimum of the average distances from $i$ to the members of any other cluster $C_J$. Based on these quantities, the silhouette coefficient $s(i)$ for each point $i$ is then defined as:

$$s(i) = \frac{b(i) - a(i)}{\max\{a(i), b(i)\}}, \quad -1 \leq s(i) \leq 1,$$

where $\max\{a(i), b(i)\}$, normalizing denominator, ensures $s(i)$ lies in the range of [-1,1]. When $s(i) \approx +1$, the intra-cluster distance $a(i)$ is much smaller than the nearest inter-cluster distance $b(i)$, indicating that point $i$ lies deep inside its own cluster and is well separated from all others. When $s(i) \approx 0$, the cohesion and separation distances are approximately equal $(a(i) \approx b(i))$, suggesting that $i$ sits near the boundary between two clusters and could plausibly belong to either. Conversely, when $s(i) \approx -1$, the intra-cluster distance exceeds the nearest inter-cluster distance $(a(i) > b(i))$, which implies that $i$ is likely misassigned and would be better placed in its neighboring cluster. Finally, the overall clustering quality is captured by the Average Silhouette Score $\tilde{s}$, defined as the mean of all individual silhouette coefficients $s(i)$:

$$\tilde{s} = \frac{1}{N} \sum_{i=1}^{N} s(i),$$

where $N$ is the total number of examples. A higher $\tilde{s}$ (closer to +1) indicates that clusters are both cohesive, whereas values near 0 or negative signal ambiguous or poor clustering assignments.

## C  ENTP: Enhancing Low-Quality SFT Data via Neural-Symbolic Text Purge-Mix

### C.1  Step 1: Low-Quality Dataset Construction

For the initial LLM rating step, we follow the same setup as Pang et al. (2025), where the LLM is prompted to rate each corpus based on four dimensions, 'Rarity', 'Complexity', 'Informativeness', and 'Overall Rating', with each dimension scored on a scale from 1 to 10. The corresponding detailed prompt template is shown in Figure 6. All initial scores in the range of 1–4 are mapped to 4, those in 9–10 are mapped to 9, and the remaining scores in between are uniformly downscaled to a 0–5 range (Pang et al., 2025). Subsequently, we adopt the K-NN Score Clusterability theory (Zhu et al., 2021) to refine the overall scores generated by the LLM.

### C.2  Step 2: One-Hop Cluster-Based Representative Selection

The complete implementation details are presented in Algorithm 1.

**Algorithm 1** One-Hop Cluster-Based Representative Selection

**Input:** Low-Quality Corpora Set, $\mathbf{S_{lq}} = \{s^o | s^o \in [0, 2]\}$

**Output:** Representative Corpora, $\mathbf{R} = \{C_r^0, ..., C_r^m\}$, where $C_r^i$ stands for the selected representative corpora set for the $i^{th}$ cluster

1: Prepare empty list, $r_{sel}$, $cluster_{one-hop} = [\,]$; Shuffle $\mathbf{S_{lq}}$

2: **for** each $s_i^o \in \mathbf{S}_{lq}$ **do**            ▷ **Step 1:** One-Hop Cluster Formation

3:      **if** $s_i^o$ not in $r_{sel}$ : **then**

4:          $CandidateList_i \leftarrow$ CosineSimilarityOver0.9($s_i^o$, $\{s_j^o; s_i^o \in \mathbf{S_{lq}}, i \neq j\}$)

5:          $cluster_{one-hop} \leftarrow cluster_{one-hop}$.append($CandidateList_i$)

6:          $r_{sel} \leftarrow r_{sel}$.extend($CandidateList_i$)

7:      **end if**

8: **end for**

9: Initialize $\alpha = 0.2$, $num\_reps = 2$

10: **for** each $cluster \in cluster_{one-hop}$ **do**      ▷ **Step 2:** Representative Corpora Selection

11:      Collect Centroid Corpus, $r_0 \leftarrow cluster$          ▷ First Representative Corpus

12:      $cluster$.remove($r_0$)

13:      **if** len($cluster$) $>= 3$ **then**

14:          $\mathbf{K} = [2, \max(10, \text{len}(cluster))]$

15:          $best\_k \leftarrow$ FindBestK($\mathbf{K}, cluster$)     ▷ Finding Out k-value With The Maximum ASS

16:          $sub\_cluster\_labels \leftarrow$ KMeans($best\_k$)

17:          **if** len($sub\_cluster\_labels$) $>= 2$ and ClusterSize($sub\_cluster\_labels$) $>= 3$ **then**

18:             # **Collect The 2$^{nd}$ & 3$^{th}$ Representative Corpus**

19:             Collect $r_0$ & $r_1 \leftarrow$ mmr_selection($sub\_cluster\_labels, num\_reps, \alpha$)

20:          **else**

21:             All $c \in cluster$ Are Updated To The Representative Corpora Set

22:          **end if**

23:      **else**

24:          All $c \in cluster$ Are Updated To The Representative Corpora Set

25:      **end if**

26: **end for**

27: **return** Full Representative Corpora Set, $\mathbf{R}$

### C.3 Step 3: Neural-Symbolic Two-To-One Corpora Fusion

#### C.3.1 Stepwise Component Analysis

As illustrated in the internal logic flow in Figure 3, Step 3 primarily comprises the connectionist and symbolism components. All connectionist components are essentially LLM-invoking operators, each responsible for a distinct task and equipped with its own carefully designed prompt template, defined as follows:

- **Domain Analysis** (DA): Based on the prompt template defined in Figure 10, the LLM extracts salient domain knowledge and the potential matching pattern from the given pair of input corpora.

- **Merged Corpus Generation** (MCG): With the initial prompt template defined in Figure 13, and given the raw corpus pair and three predefined fusion strategies, the LLM produces three corpus fusion variants that fully leverage the prior knowledge embedded in these strategies. Subsequently, the initial prompt template will be updated in response to the corresponding symbolic loss, with all candidate prompt templates defined in Figures 17-25.

- **Information Completeness Detection** (ICD): Utilizing the prompt template in Figure 14, the LLM evaluates the completeness of the merged corpus and its coverage of all elements necessary for the intended use across three aspects: (1) *Key-Term Coverage* For each key term extracted from the raw corpora, the LLM determines whether the merged corpus retains the term, either explicitly or through related information, or omits it, and subsequently outputs two lists: one of retained terms and one of missing terms; (2) *Question Quality* Since each source corpus primarily consists of one or more question–answer pairs, we analyze the question component of the merged corpus by instructing the LLM to: ① verify the presence of a well-formed question; ② classify it as open- or closed-ended; ③ determine whether external knowledge is needed to answer it; ④ identify included contextual details; and ⑤ highlight any missing contextual information; (3) *Answer Quality* Similarly, for the answer component of the merged corpus, we engage the LLM to: ① verify the presence of a direct answer to the question; and ② determine whether regeneration is necessary, providing justification if so.

- **Final Answer Check** (FAC): Referring to the prompt template defined in Figure 12, unlike the answer quality check in **ICD**, the LLM in this operator focuses not only on verifying the presence of a direct answer but also on identifying any unnecessary, irrelevant or redundant information that needs to be removed.

- **Final Answer Update** (FAU): The prompt templates designed for this operator are responsible for pruning the answer section labeled "### Assistant" without modifying any information in the "### User" section. They primarily address cases of omitted direct answers (see Figure 15) as well as the removal of unnecessary, irrelevant, or redundant information (see Figure 16).

Furthermore, the symbolism components are defined as follows:

- **Strategy Selection** (SS): As illustrated in Figure 11, we define nine fusion strategies, three for each of the three relationship types ("same-domain", "related-domain", and "unrelated-domain"), derived from the literature-writing study (Nelson & King, 2023; Knobel, 2017; Bazerman, 2003), leveraging prior knowledge to guide the LLM in merging two corpora on a case-by-case basis.

- **Symbolic Loss** (denoted as $\mathcal{L}_{\text{Sym}}$): As shown in Figure 26 and Figure 27, symbolic loss is represented as a structured, schema-compliant JSON-like object. All root nodes are explicitly defined as attribute nodes, such as "context_contain" (from **ICD**), "context_missing" (from **ICD**), and "direct_answer" (from **FAC**); the branch nodes capture the corresponding information, such as the context contained in the current corpus, the necessary context that is missing, and the direct answer itself.

- **SPO** (Symbolic Prompt Optimizer): A logic controller that enforces the regeneration budget and checks whether all root node conditions from **ICD** and **FAC** are satisfied

(i.e., $\mathcal{L}_{\text{Sym}} = 0$). It quantifies symbolic loss by counting unsatisfied root node conditions and updates the candidate prompt templates for the **MCG** or **FAC** operator via backpropagation to address the identified symbolic loss.

### C.3.2 STEPWISE WORKFLOW

The completed and detailed stepwise workflow is depicted in Figure 5, structured into two sequential sub-processes: **Cycle 1**, followed by **Cycle 2**. More specifically, we first input two raw corpora into the DA operator. Once the relationship is determined ("related-domain" in our example), the corresponding strategy set is then allocated to participate in the MCG operator. With the generation of three merged corpora from distinct fusion strategies, they are then input to the ICD operator to obtain the corresponding symbolic loss. This step also serves as the entry point of the **Cycle 1**. Subsequently, all symbolic losses are input to the SPO operator, which prepares candidate prompt templates to address the corresponding losses. Thereafter, these templates are used to update the MCG operator's prompt template via backpropagation for the next iteration. Once all checking conditions from the ICD operator are satisfied (i.e., $\mathcal{L}_{\text{Sym}} = 0$) or the maximum number of regenerations is reached, the merged corpus with the minimum symbolic loss is then selected as the optimal corpus from **Cycle 1**, denoted as $\mathcal{C}_{\text{C1}}$. In the next step, this optimal corpus $\mathcal{C}_{\text{C1}}$ is provided to the FAC operator to derive the symbolic loss for its answer section labeled "### Assistant". Similarly, this symbolic loss is then passed to the SPO operator to obtain the candidate prompt template, marking the commencement of **Cycle 2**. Immediately afterward, the candidate prompt template is back-propagated to the FAU operator to modify the corpus $\mathcal{C}_{\text{C1}}$ in preparation for the next iteration. Immediately thereafter, upon satisfying all checking conditions specified by the FAC operator (i.e., $\mathcal{L}_{\text{Sym}} = 0$) or reaching the maximum number of regenerations, the final "### Assistant" content (containing only the answer) with the minimum symbolic loss from **Cycle 2** is combined with the retained optimal "### User" section from **Cycle 1**, yielding the optimal merged corpus, $\mathcal{C}_{\text{AB}}$.

### C.3.3 DISCUSSION OF LLM INFERENCE SPACE EXPLORATION

Compared to one of the prevalent paradigms for LLM inference space search (Zhang et al., 2024a; Xu et al., 2024a), which primarily relies on Monte Carlo Tree Search (MCTS) encompassing four core steps, selection, expansion, simulation, and backpropagation, the backpropagation phase in MCTS updates nodes sequentially from the simulation node back up to the root node. This paradigm is generally applied in scenarios where no specialized prior knowledge is available, and the process must rely solely on the LLM's inherent prior knowledge. However, in our case, the core question is:

*What should the combination of corpus A and corpus B actually be?*

Following the traditional MCTS paradigm, where the fusion process relies entirely on the LLM's prior knowledge, regardless of the relationship between corpus A and corpus B, the LLM would simply concatenate the two corpora to form the merged corpus AB. From a human cognitive perspective, such a merged corpus lacks a clear theme or focus. Even worse, the response generated from this merged corpus may be unrelated to significant thematic content, resulting in a corpus that is entirely uninterpretable and essentially meaningless. Therefore, instead of relying solely on the LLM's prior knowledge, we incorporate prior knowledge from the literature-writing domain, which not only provides clear guidelines but also significantly narrows the LLM's reasoning search space, thereby reducing its computational cost and enabling faster convergence to the most probable optimal solution. As illustrated in Figure 5, our iterative procedure of Cycle 1 and Cycle 2 progressively achieves global optimality through sequential local optimizations.

### C.3.4 FULL IMPLEMENTATION DETAILS CAN BE FOUND IN ALGORITHM 2.

---

**Algorithm 2** Neural-Symbolic Two-To-One Corpora Fusion

---

**Input:** Raw Corpus A and B, $\{\mathcal{C}_A, \mathcal{C}_B\}$; Carefully Designed Prompt Set, $\{\mathcal{P}_{DA}, \mathcal{P}_{MCG}, \mathcal{P}_{ICD}, \mathcal{P}_{FAC}, \mathcal{P}_{FAU}\}$
**Output:** Optimal Merged Corpora Generated By Different Fusion Strategies, $\{\mathcal{C}_{AB}, ...\}$

---

1: **# Prompt LLM: Perform Domain Analysis (DA) Task**
2: symbolic report$_{DA}$ $\leftarrow \mathcal{F}(\mathcal{P}_{DA}(\mathcal{C}_A, \mathcal{C}_B))$
3: Strategy Set$_{AB}$ $\leftarrow$ FusionStrategySelection(symbolic report$_{DA}$)  $\triangleright$ **Symbolic Logic Controller**
4: **# Prompt LLM: Perform Merged Corpus Generation (MCG) Task**
5: Merged Corpus List $\leftarrow \mathcal{F}(\mathcal{P}_{MCG}(\mathcal{C}_A, \mathcal{C}_B, \text{Strategy Set}_{AB}))$
6: Final Optimal Merged Corpus List, $\mathbf{L}_{optimal} \leftarrow []$
7: **for** each corpus $\in$ Merged Corpus List **do**
8:     temp corpus, $\mathcal{C}_{temp} \leftarrow$ corpus
9:     temp strategy, $s_{temp} \leftarrow$ corresponding strategy
10:     **# Prompt LLM: Perform Information Completeness Detection (ICD) Task**
11:     symbolic loss, $\mathcal{L}_{Sym} \leftarrow \mathcal{F}(\mathcal{P}_{ICD}(\mathcal{C}_{temp}))$
12:     $num\_retry \leftarrow 2$
13:     buffer list for storing all temporary merged corpus, $\mathbf{C} \leftarrow [\mathcal{C}_{temp}]$
14:     buffer list for storing all symbolic loss of the corresponding temporary merged corpus, $\mathbf{L} \leftarrow [\mathcal{L}_{Sym}]$
15:     **while** $\mathcal{L}_{Sym} \neq 0$ and $num\_retry < 4$ **do**  $\triangleright$ **Cycle 1**
16:         **# Update Prompt Template for MCG Task**
17:         $\mathcal{P}_{MCG} \leftarrow$ SymbolicPromptOptimizer($\mathcal{P}_{MCG}, \mathcal{L}_{Sym}$)  $\triangleright$ **Back Propagation**
18:         **# Update Merged Corpus, $\mathcal{C}_{temp}$**
19:         $\mathcal{C}_{temp} \leftarrow \mathcal{F}(\mathcal{P}_{MCG}(\mathcal{C}_A, \mathcal{C}_B, \int_{temp}))$
20:         **# Collect The Latest Merged Corpus**
21:         $\mathbf{C}$.append($\mathcal{C}_{temp}$)
22:         **# Update Symbolic Loss, $\mathcal{L}_{Sym}$**
23:         $\mathcal{L}_{Sym} \leftarrow \mathcal{F}(\mathcal{P}_{ICD}(\mathcal{C}_{temp}))$
24:         $\mathbf{L}$.append($\mathcal{L}_{Sym}$)
25:         $num\_retry$ += 1
26:     **end while**
27:     **if** $\mathcal{L}_{Sym} == 0$ **then**
28:         optimal merged corpus from Cycle 1, $\mathcal{C}_{c1} \leftarrow \mathcal{C}_{temp}$
29:     **else if** $\mathbf{L}$.count(min($\mathbf{L}$)) == 1 and $num\_retry > 3$ **then**
30:         optimal merged corpus from Cycle 1, $\mathcal{C}_{c1} \leftarrow \mathbf{C}[\mathbf{L}.\text{index}(\min(\mathbf{L}))]$
31:     **else if** $\mathbf{L}$.count(min($\mathbf{L}$)) > 1 and $num\_retry > 3$ **then**
32:         optimal merged corpus from Cycle 1, $\mathcal{C}_{c1} \leftarrow \mathbf{C}[\text{random.choice}([i \text{ for } i, v \text{ in enumerate}(\mathbf{L}) \text{ if } v == \min(\mathbf{L})])]$
33:     **end if**
34:     symbolic loss, $\mathcal{L}_{Sym} \leftarrow \mathcal{F}(\mathcal{P}_{FAC}(\mathcal{C}_{c1}))$
35:     $num\_retry \leftarrow 2$
36:     buffer list for storing all temporary merged corpus, $\mathbf{C} \leftarrow [\mathcal{C}_{c1}]$
37:     buffer list for storing all symbolic loss of the corresponding temporary merged corpus, $\mathbf{L} \leftarrow [\mathcal{L}_{Sym}]$
38:     **while** $\mathcal{L}_{Sym} \neq 0$ and $num\_retry < 4$ **do**  $\triangleright$ **Cycle 2**
39:         **# Update Prompt Template for FAU Task**
40:         $\mathcal{P}_{FAU} \leftarrow$ SymbolicPromptOptimizer($\mathcal{P}_{FAU}, \mathcal{L}_{Sym}$)  $\triangleright$ **Back Propagation**
41:         **# Update Merged Corpus, $\mathcal{C}_{c1}$**
42:         $\mathcal{C}_{c1} \leftarrow \mathcal{F}(\mathcal{P}_{FAU}(\mathcal{C}_{c1}))$
43:         **# Collect The Latest Merged Corpus**
44:         $\mathbf{C}$.append($\mathcal{C}_{c1}$)
45:         **# Update Symbolic Loss, $\mathcal{L}_{Sym}$**
46:         $\mathcal{L}_{Sym} \leftarrow \mathcal{F}(\mathcal{P}_{FAU}(\mathcal{C}_{C1}))$
47:         $\mathbf{L}$.append($\mathcal{L}_{Sym}$)
48:         $num\_retry$ += 1
49:     **end while**
50:     **if** $\mathcal{L}_{Sym} == 0$ **then**
51:         optimal merged corpus from Cycle 2, $\mathcal{C}_{AB} \leftarrow \mathcal{C}_{c1}$
52:     **else if** $\mathbf{L}$.count(min($\mathbf{L}$)) == 1 and $num\_retry > 3$ **then**
53:         optimal merged corpus from Cycle 2, $\mathcal{C}_{AB} \leftarrow \mathbf{C}[\mathbf{L}.\text{index}(\min(\mathbf{L}))]$
54:     **else if** $\mathbf{L}$.count(min($\mathbf{L}$)) > 1 and $num\_retry > 3$ **then**
55:         optimal merged corpus from Cycle 2, $\mathcal{C}_{AB} \leftarrow \mathbf{C}[\text{random.choice}([i \text{ for } i, v \text{ in enumerate}(\mathbf{L}) \text{ if } v == \min(\mathbf{L})])]$
56:     **end if**
57:     **# Collect The Finalized Merged Corpus**
58:     $\mathbf{L}_{optimal}$.append($\mathcal{C}_{AB}$)
59: **end for**
60: **return** Eligible Merged Corpus List, $\mathbf{L}_{optimal}$

---

# D   MORE EXPERIMENTS

## D.1   EXPERIMENTAL SETUP

Table 5: Comprehensive overview of the source corpora used in this work. We report three additional descriptive dimensions, the average number of conversation turns ($\bar{N}_{\text{rounds}}$), the average prompt length ($\bar{L}_{\text{prompt}}$), and the average response length ($\bar{L}_{\text{response}}$), to provide a more nuanced understanding of the composition of our source corpora.

| Datasets | Derived From | Data size | $\bar{N}_{\text{rounds}}$ | $\bar{L}_{\text{prompt}}$ | $\bar{L}_{\text{response}}$ |
|---|---|---|---|---|---|
| **Stanford Alpaca** | Generated w/ Davinci-003 | 52K | 1.0 | 23.5 | 56.4 |
| **Flan V2** | Human Annotation | 100K | 1.0 | 304.1 | 27.7 |
| **Open-Assistant 1** | Human Annotation | 33K | 1.6 | 32.3 | 189.1 |
| **WizardLM** | ChatGPT Annotation | 100K | 1.0 | 122.3 | 352.5 |
| **Dolly** | Human Annotation | 15K | 1.0 | 99.5 | 79.3 |

### D.1.1   SOURCE CORPORA

For the source corpora used in this work, we follow the same setup as $DS^2$ (Pang et al., 2025), where the corpora consist of five instruction-following datasets originating either from human annotations or generated by powerful LLMs. A comprehensive overview of our source corpora is provide in Table 5. Notably, all of the component datasets differ across format, annotation quality, prompt length, and target task, underscoring the rich diversity of our source data pool.

### D.1.2   EVALUATION SETUP

In this paper, we conduct experiments on five evaluation tasks: MMLU (Hendrycks et al., 2020), TruthfulQA (Lin et al., 2021), BBH (Suzgun et al., 2022), GSM8K (Cobbe et al., 2021), and TyDiQA (Clark et al., 2020). The hyperparameter settings primarily follow those used in recent work by Wang et al. (2023a). Besides, to ensure consistency with the format of our generated merged corpora, we apply our marker format (**### User\n{input}\n### Assistant\n{output}**) to each input-output pair, embedding it into the corresponding official prompt template in the evaluation set. For reproducibility, we provide a brief summary of the key details:

- **MMLU (Hendrycks et al., 2020)**: Following the original MMLU setup, all evaluations are conducted in the zero-shot setting.

- **TruthfulQA (Lin et al., 2021)**: We follow the default QA prompt template with 6 in-context examples to generate answers for 818 TruthfulQA questions. In alignment with the setup in Wang et al. (2023a), we use two LLaMA-2-7B-based models to evaluate the truthfulness[4] and informativeness[5] of the generated responses. These judge models assess the truthful and informative rates separately. Similarly, we report the Informative-Truthful Rate as our final metric, calculated as the product of the informativeness and truthfulness rates (Lin et al., 2021).

- **BBH (Suzgun et al., 2022)**: Using the official prompts, we perform generation under a 3-shot setting without CoT reasoning. Additionally, 40 examples from each BBH sub-task are selected for evaluation.

- **GSM8K (Cobbe et al., 2021)**: We evaluate the fine-tuned models on a randomly selected subset of 200 samples from the original test set (1319 samples). Specifically, we adopt an 8-shot in-context learning setup to simulate the chain-of-thought (CoT) reasoning setting.

- **TydiQA (Clark et al., 2020)**: We use this dataset to evaluate model performance on multilingual question answering across nine languages. For each language, 100 examples

---

[4]Hugging Face Model: allenai/truthfulqa-truth-judge-llama2-7B

[5]Hugging Face Model: allenai/truthfulqa-info-judge-llama2-7B

are selected. To help the models adapt to the answer format, one in-context example is provided during evaluation. The average F1 score across all languages is reported.

### D.1.3 DETAILED BASELINE METHOD DESCRIPTIONS

Our **ENTP**-generated synthetic dataset comprises 54888 samples drawn from the LQ set (totaling 123786 samples); of these, 15488 are generated via intra-cluster fusion and 39400 via inter-cluster fusion. To assess the efficacy of **ENTP**, we compare it against 13 representative data-selection baselines, each applied uniformly to the LQ Set to ensure a fair comparison:

1. *Vanilla Base Model* denotes the original base model without any fine-tuning;

2. *LQ Set & HQ Set* represent the low-quality set (123786 samples) and high-quality set (131247 samples), both extracted from *Full Set*. We first employ the LLM-rating step from $DS^2$ (Pang et al., 2025) to assign overall-quality scores to every sample in the source data pool. Subsequently, we apply the clusterability-based method (Zhu et al., 2021) to correct the potential scoring bias. Samples with curated scores in the range [0, 2] form the LQ Set, while those with scores in [3, 5] compose the HQ Set;

3. *Full Set* comprises 300932 samples as our source data pool;

4. *Completion Length* utilizes the length of the whole corpus as an indicator to assess to sample quality. Intuitively, longer completions tend to reflect richer, higher-quality dialogues, providing more context, depth, and informativeness;

5. $KNN_i$ is defined as the Average Euclidean Distance (AED) from each raw embedding vector to its $i$ nearest neighbors within the embedding space. We obtain all embeddings using the same model[1] and then rank samples by their AED in ascending order. Samples with smaller distances are considered more centrally located and thus more representative in the embedding space;

6. *Perplexity*, computed using a pre-trained language model in a zero-shot fashion, is employed as the evaluation metric. We compute perplexity for each sample using LLaMA-3.1-8B-Instruct model. Samples are then selected in descending order of perplexity. A larger perplexity score indicates greater model uncertainty, suggesting the sample is more difficult or rare;

7. *Random Selection*, all samples are randomly selected;

8. *AlpaGasus (Random)* (Chen et al., 2023) employs ChatGPT to score each sample and retains only the highest-rated samples for fine-tuning. For a fair comparison, we use gpt-4o-mini[2] as the scoring model. Since the number of samples receiving the top score (55530) exceeds our required dataset size (54888), we randomly sample the final set from among those highest-scoring samples;

9. *IFD* (Li et al., 2024b), Instruction-Following Difficulty, quantifies how much an instruction aids a model's generation by comparing the model's loss (or perplexity) with and without instruction context. A higher IFD score indicates that the model is less familiar with a given sample, implying this sample is relatively rare;

10. *Superfiltering* (Li et al., 2024a) utilizes a small and weaker model, GPT-2 (Radford et al., 2019)[6], for the data selection;

11. *DEITA* (Liu et al., 2024b) jointly uses two pre-trained scoring model to rate data samples based on complexity[7] and quality[8]. However, all the single-turn samples are rated as 3. In order to further demonstrate the effectiveness of this method, we also employ our curated scores as an alternative, which is reported as *DEITA (Our Curated Score)*;

12. *RDS+* (Ivison et al., 2025), representation-based data selection, utilizes a weighted mean pooling of a pre-trained model's final hidden states for computing the cosine similarity between the raw dataset and the validation set. Accordingly, this method extracts an

---

[6]Hugging Face Model: openai-community/gpt2
[7]Hugging Face Model: hkust-nlp/deita-complexity-scorer
[8]Hugging Face Model: hkust-nlp/deita-quality-scorer

optimal subset from the source pool for each test benchmark individually. Nevertheless, this test-specific subset does not necessarily yield superior performance on that specific benchmark, in fact, a subset curated using a different validation set may outperform it. Consequently, to showcase the upper performance bound of this method, we also report the best result achieved for each test benchmark, denoted as *RDS+ (best)*;

13. DS$^2$ (Pang et al., 2025) leverages LLM-generated quality scores, corrected via a score transition matrix, and further integrates cosine similarity-based long-tail scoring to select samples that are both high-rated and rare;

14. *LESS* (Xia et al., 2024) requires a validation set for each evaluation benchmark. It first constructs a gradient datastore for the validation set and then computes the influence score for every sample in the entire low-quality set. For a fair comparison, we collect the top 54888 samples ranked by LESS for each validation set. Moreover, to present the optimal performance of LESS, we report results only on the corpus subsets selected exclusively for each corresponding task;

15. *MathFusion* (Pei et al., 2025) provides three fusion strategies, including conditional fusion, parallel fusion, and sequential fusion. We apply all three strategies to the low-quality corpora via random pairing using gpt-4o-mini$^2$. We then evenly select 18296 merged corpora per strategy, resulting in a total of 54888 merged corpora;

16. *Evol-Instruct* (Xu et al., 2024b) offers five *In-depth Evolving* prompt templates and one *In-breadth Evolving* prompt template. Following the official configuration, we set the number of evolution iteration to $M = 4$. After completing all four evolution rounds, we randomly sample 54888 evolved corpora as the final selection;

17. *Self-Instruct* (Wang et al., 2023b) provides two types of prompt templates: one for classification corpora and one for non-classification corpora. For a fair comparison, we use gpt-4o-mini$^2$ to generate the augmented corpora. Similarly, we randomly sample 54888 augmented corpora as the final selection;

18. *1-to-1 Rewriting/ Enhancement* serves as a simple baseline in which we use gpt-4o-mini$^2$ to directly rewrite or enhance for each low-quality corpus. After obtaining all the augmented corpora, we randomly sample 54888 of them as the final selection;

19. *Direct Corpora Fusion Without Step 2&3* represents the baseline in which we use gpt-4o-mini$^2$ to directly fuse two corpora via randomly pairing samples from the low-quality set, bypassing both our clustering step (Step 2) and neural-symbolic fusion step (Step 3);

20. *Direct Corpora Fusion Without Step 3* represents the baseline in which we use gpt-4o-mini$^2$ to directly fuse two corpora via randomly pairing samples from the representative low-quality set, bypassing our neural-symbolic fusion step (Step 3) only.

### D.1.4 TRAINING SETUP

In our experiments, we fine-tune three LLMs, including Mistral-7B-v0.3 (Jiang et al., 2023), LLaMA-3.1-8B (Grattafiori et al., 2024), and Qwen2.5-7B (Team, 2024) using eight NVIDIA H20 or A800 GPUs. Following the experimental setup of Wang et al. (2023a), we apply LoRA (Hu et al., 2022) with a rank of 64 and a scaling factor of 16 to all experiments. The training configuration includes a batch size of 128, a learning rate of 1e-4, 5 training epochs, a dropout rate of 0.1, and a warm-up ratio of 0.03. The maximum input length is set to 2048 tokens for all models by default.

### D.2 MORE EMPIRICAL OBSERVATIONS

**Further evidence supporting the effectiveness of ENTP.** As shown in Table 6, even when paired with the Qwen2.5-7B model, our **ENTP** consistently outperforms all 13 baselines, including the Full Set setting, on average, further demonstrating its ability to overcome the limitations of relying solely on raw high-quality data.

Table 6: **Performance comparison on the OpenLLM leaderboard.** The default data size is 54888. The fine-tuning base model is Qwen2.5-7B. Best and second-best results on average are highlighted in **bold red** and **bold black**, respectively. Performance changes of **ENTP** with respect to the LQ Set across all benchmarks are also reported.

| Dataset | MMLU (EM ↑) | GSM8K (EM ↑) | BBH (EM ↑) | TydiQA (1-Shot F1 ↑) | Average ↑ |
|---|---|---|---|---|---|
| Base Model: Qwen2.5-7B | | | | | |
| Vanilla Base Model | 71.8 | 83.5 | 58.1 | 25.3 | 59.7 |
| LQ Set (123786) | 69.3 | 77.5 | 58.6 | 55.4 | 65.2 |
| HQ Set (131247) | 72.2 | 79.0 | 60.4 | 60.2 | 68.0 |
| Full Set (300932) | 72.0 | 78.0 | 59.8 | 65.2 | **68.8** |
| Completion Length | 67.2 | 75.5 | 57.9 | 62.7 | 65.8 |
| KNN$_{10}$ | 70.4 | 77.5 | 57.7 | 63.2 | 67.2 |
| Perplexity | 70.1 | 76.0 | 52.9 | 63.7 | 65.7 |
| Random Selection | 69.3 | 75.5 | 57.3 | 65.4 | 66.9 |
| AlpaGasus (Random) | 65.8 | 74.0 | 58.1 | 57.6 | 63.9 |
| IFD | 63.9 | 68.5 | 53.2 | 52.1 | 59.4 |
| Superfiltering | 68.3 | 76.0 | 55.0 | 59.4 | 64.7 |
| DEITA | 68.6 | 76.0 | 59.4 | 59.5 | 65.9 |
| DEITA (Our Curated Score) | 68.0 | 73.0 | 57.6 | 62.6 | 65.3 |
| RDS+ | 69.1 | 78.5 | 55.4 | 55.1 | 64.5 |
| RDS+ (Best) | 69.1 | 78.5 | 57.9 | 55.1 | 65.2 |
| DS$^2$ | 67.2 | 79.5 | 58.1 | 61.6 | 66.6 |
| **ENTP** | 69.2 (-0.1) | 79.5 (+2.0) | 59.1 (+0.5) | 69.3 (+13.9) | **69.3 (+4.1)** |

### D.2.1 GPU Runtime & API Cost & Validation Set Requirement Comparison

Regarding the cost analysis, Table 7 presents a comparison of GPU runtime, API cost, and validation set requirements across several baselines. In addition, we report the average API cost per resultant corpus.

Table 7: Comparison of GPU Runtime, API Cost, and Validation Set Requirement Across Baselines

| | LESS (2024) | MathFusion (2025) | Evol-Instruct (2024b) | **ENTP** |
|---|---|---|---|---|
| Average API Cost (in USD) | 0 | 0.004 | 0.003 | 0.005 |
| GPU Runtime (in GPU-hours) | 152.5 | 17 | 17 | 17 |
| Validation Set | Required | Not Required | Not Required | Nor Required |

## E MORE ABLATION STUDY

### E.1 SUPPLEMENTARY ABLATION SETUP

We introduce the Vanilla Base Model, LQ Set, HQ Set and Full Set as control groups. More experimental-group configuration for different research objectives are provided as follows:

- To gain deeper insight into the effects of **ENTP**'s two fusion mechanisms, Intra-Cluster and Inter-Cluster fusion, on LLM performance across four downstream tasks (MMLU (Hendrycks et al., 2020), BBH (Suzgun et al., 2022), GSM8K (Cobbe et al., 2021), and TyDiQA (Clark et al., 2020)), we independently sample varying proportions from each fusion-generated merged corpora to create experimental groups. These settings are denoted as "Intra-x%" and "Inter-x%", where x% indicates the proportion of data selected from the corresponding fusion-produced dataset;

- To further investigate the impact of thees two fusion mechanisms on the HQ Set, we configured three experimenetal groups: (1) *HQ+Intra-x%* (adding x% samples from Intra-Cluster Fusion), (2) *HQ+Inter-x%* (adding x% from Inter-Cluster Fusion), and (3)

*HQ+**ENTP**-x%* (adding x% of both fusion types). In all three cases, the entire HQ Set is included. We then evaluate these configurations across five downstream tasks: MMLU (Hendrycks et al., 2020), Truthfulqa (Lin et al., 2021), BBH (Suzgun et al., 2022), GSM8K (Cobbe et al., 2021), and TyDiQA (Clark et al., 2020)

### E.2 More Empirical Observations

**Additional Observation 1:** **Empirical scaling laws consistently hold across all ENTP -generated datasets.** As shown in Table 8-13, across all configurations, whether using only the Intra-Cluster Fusion, only the Inter-Cluster Fusion, both fusion types, and irrespective of combining with the full HQ Set, increasing the volume of **ENTP** -generated data consistently improves average model performance, aligning with established empirical neural scaling laws.

**Additional Observation 2:** **Low-Quality does contribute to the average performance.** According to the results from Table 13, across all three testing-model settings, the Full Set, which comprises the entire HQ Set and the LQ set, yields higher average performance than the HQ Set alone: 50.6 (+0.3), 57.7 (+1.7), and 63.3 (+1.3), respectively. These experimental results also align with the predictions of the scaling laws (Kaplan et al., 2020), indicating that the prevailing data-selection paradigm's claim, that a small subset of data can outperform the full dataset, has significant limitations. Similarly, Pang et al. (2025) demonstrated that fine-tuning LLMs on a curated subset can outperform using the full dataset. However, their curated subset does not entirely consist of the highest-scoring data points. This indicates that: (1) so-called low-quality data still contains substantial informational value; and (2) relying solely on native high-quality data may be insufficient for significantly enhancing LLM performance on downstream tasks. Therefore, it is inadvisable to discard low-quality data outright.

**Additional Observation 3:** **Using just portions of the ENTP -generated data, whether from Intra-Cluster or Inter-Cluster Fusion, consistently outperforms the HQ Set alone, and in some cases, even surpasses the Full Set configuration on average.** As shown in Table 8, when Qwen2.5-7B or Mistral-7B-v0.3 serves as the base model, using just 60% of Intra-Cluster Fusion–generated samples consistently surpasses all baselines from the control group. In the case of LlaMA-3.1-8B, the same subset achieves the second-best average performance, nearly matching the full-set result, and still outperforming the HQ Set. Similarly, according to the Table 9, when using the Mistral-7B-v0.3 model, even a dataset comprised of only 60% Inter-Cluster Fusion–generated samples achieves an average performance of 53.1, on par with the best-performing baseline from the control group. Moreover, increasing this proportion to 100% raises average performance to 54.6, thereby attaining state-of-the-art results across both the experimental and control groups. Therefore, all of our experimental results demonstrate that our proposed paradigm consistently exceeds the performance ceiling of the traditional paradigm trained solely on high-quality data, effectively serving as a viable alternative.

**Additional Observation 4:** **Advanced LLM benefits more from fusion data built on heterogeneous corpora.** As shown in Table 11-12, the more advanced LLM, Qwen2.5-7B, benefits the most from the HQ+Inter setup, in comparison with the HQ+Intra configuration. This is because Inter-Cluster Fusion involves merging corpus pairs with lower similarity, which likely introduces rarer and more diverse information into the merged corpus, thereby enhancing the expressiveness of individual samples. Additionally, advanced LLMs are pre-trained on larger, more diverse, and more up-to-date corpora, leading to a more balanced data distribution. This enables them to better interpret and utilize the rare or novel information produced by heterogeneous corpus fusion, a conclusion also supported by the FuseRL framework (Zhong et al., 2025).

## F Curated Overall Score Distribution Comparison

To visually highlight the quality gap between the original low-quality corpora set and the **ENTP**-generated merged corpora set, we reuse the curated overall score (higher-is-better) employed in Step 1 to distinguish high- from low-quality samples; the resulting distributions for both set are shown in Figure 4.

Table 8: **Performance comparison among the Vanilla Base Model, LQ Set, HQ Set, Full Set, and various proportions of the ENTP -generated dataset from Intra-Cluster Fusion.** The fine-tuning base models are Qwen2.5-7B, Mistral-7B-v0.3, and Llama-3.1-8B. Best and second-best results on average are highlighted in **bold red** and **bold black**, respectively. The average performance changes of **ENTP**, relative to the LQ Set, are also reported.

| Dataset | MMLU (EM ↑) | GSM8K (EM ↑) | BBH (EM ↑) | TydiQA (1-Shot F1 ↑) | Average ↑ |
|---|---|---|---|---|---|
| **Base Model: Qwen2.5-7B** | | | | | |
| Vanilla Base Model | 71.8 | 83.5 | 58.1 | 25.3 | 59.7 |
| LQ Set (123786) | 69.3 | 77.5 | 58.6 | 55.4 | 65.2 |
| HQ Set (131247) | 72.2 | 79.0 | 60.4 | 60.2 | 68.0 |
| Full Set (300932) | 72.0 | 78.0 | 59.8 | 65.2 | 68.8 |
| Intra-20% | 71.2 | 86.5 | 58.1 | 67.8 | **70.9 (+5.7)** |
| Intra-40% | 71.0 | 84.5 | 59.4 | 62.4 | **69.3 (+4.1)** |
| Intra-60% | 70.6 | 81.5 | 61.4 | 69.9 | **70.9 (+5.7)** |
| **Base Model: Mistral-7B-v0.3** | | | | | |
| Vanilla Base Model | 59.7 | 38.0 | 47.6 | 54.8 | 50.0 |
| LQ Set (123786) | 47.5 | 43.5 | 52.7 | 41.1 | 46.3 |
| HQ Set (131247) | 58.4 | 46.0 | 55.5 | 52.5 | **53.1** |
| Full Set (300932) | 60.0 | 43.5 | 52.5 | 53.4 | 52.4 |
| Intra-20% | 59.6 | 40.0 | 52.9 | 56.1 | 52.2 (+5.9) |
| Intra-40% | 59.9 | 39.5 | 54.5 | 55.1 | 52.3 (+6.0) |
| Intra-60% | 60.1 | 43.5 | 53.5 | 57.3 | **53.6 (+7.3)** |
| **Base Model: Llama-3.1-8B** | | | | | |
| Vanilla Base Model | 64.1 | 58.0 | 55.3 | 22.1 | 49.9 |
| LQ Set (123786) | 52.7 | 57.0 | 61.0 | 44.7 | 53.9 |
| HQ Set (131247) | 62.3 | 57.5 | 59.3 | 58.9 | 59.5 |
| Full Set (300932) | 63.5 | 61.0 | 59.1 | 62.8 | **61.6** |
| Intra-20% | 63.9 | 54.5 | 57.5 | 52.0 | 57.0 (+3.1) |
| Intra-40% | 64.0 | 59.5 | 60.6 | 53.9 | 59.5 (+5.6) |
| Intra-60% | 63.6 | 56.5 | 59.6 | 60.1 | **60.0 (+6.1)** |

For each corpora set, we report the frequency of each score level (from 0 to 5) and compute the corresponding average. As illustrated in the Figure 4, the average score of the merged corpora obtained after applying our Step 2 and Step 3 (3.13) is two times larger than the average score of the corpora (1.51) without applying these steps.

Table 9: **Performance comparison among the Vanilla Base Model, LQ Set, HQ Set, Full Set, and various proportions of the ENTP -generated dataset from Inter-Cluster Fusion.** The fine-tuning base models are Mistral-7B-v0.3 and Llama-3.1-8B. Best and second-best results on average are highlighted in **bold red** and **bold black**, respectively. The average performance changes of **ENTP**, relative to the LQ Set, are also reported.

| Dataset | MMLU (EM ↑) | GSM8K (EM ↑) | BBH (EM ↑) | TydiQA (1-Shot F1 ↑) | Average ↑ |
|---|---|---|---|---|---|
| Base Model: Mistral-7B-v0.3 | | | | | |
| Vanilla Base Model | 59.7 | 38.0 | 47.6 | 54.8 | 50.0 |
| LQ Set (123786) | 47.5 | 43.5 | 52.7 | 41.1 | 46.3 |
| HQ Set (131247) | 58.4 | 46.0 | 55.5 | 52.5 | 53.1 |
| Full Set (300932) | 60.0 | 43.5 | 52.5 | 53.4 | 52.4 |
| Inter-20% | 59.4 | 37.0 | 54.3 | 57.5 | 52.1 (+5.8) |
| Inter-40% | 59.1 | 42.5 | 51.8 | 57.1 | 52.6 (+6.3) |
| Inter-60% | 58.9 | 45.0 | 52.1 | 56.3 | 53.1 (+6.8) |
| Inter-80% | 58.9 | 47.0 | 52.2 | 56.7 | **53.7 (+7.4)** |
| Inter-100% | 58.0 | 49.0 | 53.1 | 58.1 | **54.6 (+8.3)** |
| Base Model: Llama-3.1-8B | | | | | |
| Vanilla Base Model | 64.1 | 58.0 | 55.3 | 22.1 | 49.9 |
| LQ Set (123786) | 52.7 | 57.0 | 61.0 | 44.7 | 53.9 |
| HQ Set (131247) | 62.3 | 57.5 | 59.3 | 58.9 | **59.5** |
| Full Set (300932) | 63.5 | 61.0 | 59.1 | 62.8 | **61.6** |
| Inter-20% | 63.6 | 55.0 | 57.7 | 53.2 | 57.4 (+3.5) |
| Inter-40% | 62.1 | 58.5 | 58.6 | 53.4 | 58.2 (+4.3) |
| Inter-60% | 62.3 | 55.5 | 57.8 | 55.9 | 57.9 (+4.0) |
| Inter-80% | 62.3 | 55.5 | 58.4 | 55.8 | 58.0 (+4.1) |
| Inter-100% | 61.9 | 60.5 | 59.9 | 54.0 | 59.1 (+5.2) |

Table 10: **Performance comparison among the Vanilla Base Model, LQ Set, HQ Set, Full Set, and datasets mixing the entire HQ Set with various proportions of the ENTP -generated Inter-Cluster Fusion samples.** The fine-tuning base model is Llama-3.1-8B. Best and second-best results on average are highlighted in **bold red** and **bold black**, respectively. The average performance changes of **ENTP**, relative to the HQ Set, are also reported.

| Dataset | MMLU (EM ↑) | TruthfulQA (EM ↑) | GSM8K (EM ↑) | BBH (EM ↑) | TydiQA (1-Shot F1 ↑) | Average ↑ |
|---|---|---|---|---|---|---|
| Base Model: Llama-3.1-8B | | | | | | |
| Vanilla Base Model | 64.1 | 32.9 | 58.0 | 55.3 | 22.1 | 46.5 |
| LQ Set (123786) | 52.7 | 44.3 | 57.0 | 61.0 | 43.9 | 51.8 |
| HQ Set (131247) | 62.3 | 41.8 | 57.5 | 59.3 | 58.9 | 56.0 |
| Full Set (300932) | 63.5 | 42.0 | 61.0 | 59.1 | 62.8 | 57.7 |
| HQ+Inter-40% | 62.0 | 45.3 | 58.0 | 59.9 | 55.0 | 56.0 (+0.0) |
| HQ+Inter-60% | 63.3 | 44.6 | 61.0 | 62.4 | 57.5 | **57.8 (+1.8)** |
| HQ+Inter-100% | 62.5 | 44.6 | 65.0 | 60.6 | 57.8 | **58.1 (+2.1)** |

Table 11: **Performance comparison among the Vanilla Base Model, LQ Set, HQ Set, Full Set, and datasets mixing the entire HQ Set with various proportions of the ENTP -generated Intra-Cluster Fusion samples.** The fine-tuning base model is Qwen2.5-7B. Best and second-best results on average are highlighted in **bold red** and **bold black**, respectively.

| Dataset | MMLU (EM ↑) | TruthfulQA (EM ↑) | GSM8K (EM ↑) | BBH (EM ↑) | TydiQA (1-Shot F1 ↑) | Average ↑ |
|---|---|---|---|---|---|---|
| Base Model: Qwen2.5-7B | | | | | | |
| Vanilla Base Model | 71.8 | 11.1 | 83.5 | 58.1 | 25.3 | 50.0 |
| LQ Set (123786) | 69.3 | 43.5 | 77.5 | 58.6 | 55.4 | 60.9 |
| HQ Set (131247) | 72.2 | 38.2 | 79.0 | 60.4 | 60.2 | 62.0 |
| Full Set (300932) | 72.0 | 41.4 | 78.0 | 59.8 | 65.2 | **63.3** |
| HQ+Intra-40% | 72.0 | 36.3 | 72.5 | 58.7 | 63.6 | 60.6 |
| HQ+Intra-60% | 72.0 | 36.5 | 80.0 | 58.1 | 64.1 | 62.1 |
| HQ+Intra-100% | 71.0 | 45.8 | 76.5 | 59.4 | 64.7 | **63.5** |

Table 12: **Performance comparison among the Vanilla Base Model, LQ Set, HQ Set, Full Set, and datasets mixing the entire HQ Set with various proportions of the ENTP -generated Inter-Cluster Fusion samples.** The fine-tuning base model is Qwen2.5-7B. Best and second-best results on average are highlighted in **bold red** and **bold black**, respectively.

| Dataset | MMLU (EM ↑) | TruthfulQA (EM ↑) | GSM8K (EM ↑) | BBH (EM ↑) | TydiQA (1-Shot F1 ↑) | Average ↑ |
|---|---|---|---|---|---|---|
| Base Model: Qwen2.5-7B | | | | | | |
| Vanilla Base Model | 71.8 | 11.1 | 83.5 | 58.1 | 25.3 | 50.0 |
| LQ Set (123786) | 69.3 | 43.5 | 77.5 | 58.6 | 55.4 | 60.9 |
| HQ Set (131247) | 72.2 | 38.2 | 79.0 | 60.4 | 60.2 | 62.0 |
| Full Set (300932) | 72.0 | 41.4 | 78.0 | 59.8 | 65.2 | **63.3** |
| HQ+Inter-20% | 71.6 | 37.3 | 73.0 | 59.5 | 64.0 | 61.1 |
| HQ+Inter-40% | 71.7 | 42.0 | 75.0 | 57.0 | 61.7 | 61.5 |
| HQ+Inter-60% | 71.6 | 44.8 | 74.0 | 57.1 | 63.1 | 62.1 |
| HQ+Inter-80% | 71.4 | 41.9 | 81.5 | 59.7 | 64.5 | **63.8** |

Table 13: **Performance comparison among the Vanilla Base Model, LQ Set, HQ Set, Full Set, and datasets mixing the entire HQ Set with various proportions of the ENTP -generated samples from both Inter-Cluster and Intra-Cluster Fusion.** The fine-tuning base models are Mistral-7B-v0.3, Llama-3.1-8B, and Qwen2.5-7B. Best and second-best results on average are highlighted in **bold red** and **bold black**, respectively.

| Dataset | MMLU (EM ↑) | TruthfulQA (EM ↑) | GSM8K (EM ↑) | BBH (EM ↑) | TydiQA (1-Shot F1 ↑) | Average ↑ |
|---|---|---|---|---|---|---|
| **Base Model: Mistral-7B-v0.3** | | | | | | |
| Vanilla Base Model | 59.7 | 30.4 | 38.0 | 47.6 | 54.8 | 46.1 |
| LQ Set (123786) | 47.5 | 43.7 | 43.5 | 52.7 | 41.1 | 45.7 |
| HQ Set (131247) | 58.4 | 39.2 | 46.0 | 55.5 | 52.5 | 50.3 |
| Full Set (300932) | 60.0 | 43.5 | 43.5 | 52.5 | 53.4 | 50.6 |
| HQ+**ENTP**-40% | 57.8 | 42.4 | 45.0 | 54.4 | 55.2 | **51.0** |
| HQ+**ENTP**-60% | 58.2 | 45.8 | 45.0 | 52.4 | 54.5 | **51.2** |
| HQ+**ENTP**-100% | 57.2 | 47.2 | 46.0 | 52.2 | 53.4 | **51.2** |
| **Base Model: Llama-3.1-8B** | | | | | | |
| Vanilla Base Model | 64.1 | 32.9 | 58.0 | 55.3 | 22.1 | 46.5 |
| LQ Set (123786) | 52.7 | 44.3 | 57.0 | 61.0 | 43.9 | 51.8 |
| HQ Set (131247) | 62.3 | 41.8 | 57.5 | 59.3 | 58.9 | 56.0 |
| Full Set (300932) | 63.5 | 42.0 | 61.0 | 59.1 | 62.8 | **57.7** |
| HQ+**ENTP**-40% | 62.5 | 44.6 | 59.0 | 58.7 | 57.4 | 56.4 |
| HQ+**ENTP**-60% | 62.7 | 43.0 | 59.5 | 61.3 | 61.5 | **57.6** |
| HQ+**ENTP**-100% | 62.7 | 47.2 | 61.5 | 61.1 | 56.1 | **57.7** |
| **Base Model: Qwen2.5-7B** | | | | | | |
| Vanilla Base Model | 71.8 | 11.1 | 83.5 | 58.1 | 25.3 | 50.0 |
| LQ Set (123786) | 69.3 | 43.5 | 77.5 | 58.6 | 55.4 | 60.9 |
| HQ Set (131247) | 72.2 | 38.2 | 79.0 | 60.4 | 60.2 | 62.0 |
| Full Set (300932) | 72.0 | 41.4 | 78.0 | 59.8 | 65.2 | **63.3** |
| HQ+**ENTP**-40% | 72.1 | 44.3 | 75.5 | 59.4 | 60.7 | 62.4 |
| HQ+**ENTP**-60% | 72.2 | 43.0 | 74.0 | 60.9 | 59.6 | 61.9 |
| HQ+**ENTP**-100% | 71.7 | 45.1 | 76.0 | 59.0 | 63.4 | **63.1** |

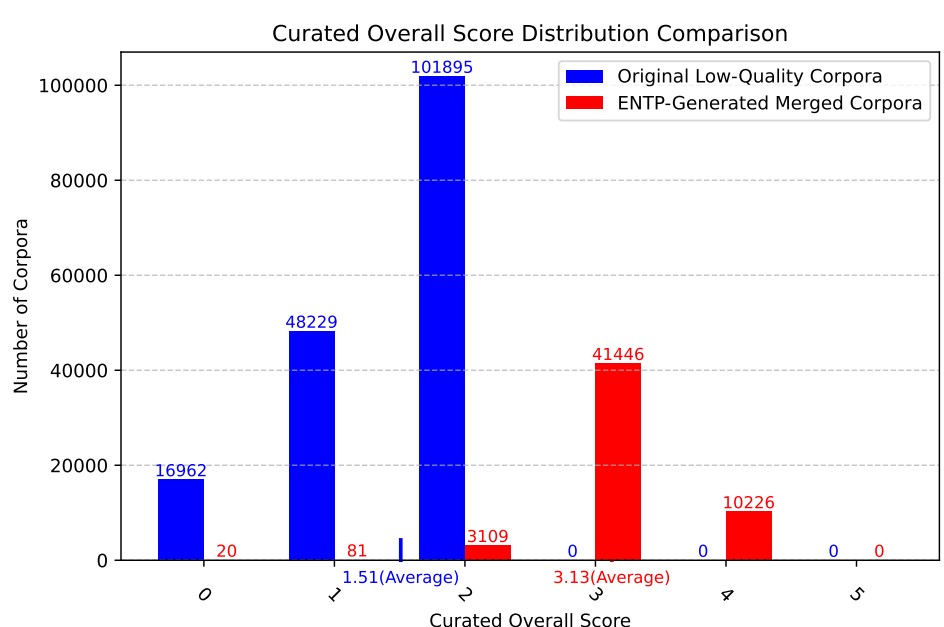

Figure 4: Curated overall score distributions for the low-quality corpora set (obtained from Step 1) and for **ENTP**-generated merged corpora set (obtained from Step 3)

## G  Concrete End-To-End Example

We present a concrete end-to-end example that systematically demonstrates the operational mechanism of the neural–symbolic fusion step (Step 3). The example is described step-by-step and includes all intermediate outputs to facilitate reproducibility and interpretability.

We begin by randomly selecting two corpora from the representative low-quality set which is obtained from the clustering step (Step 2), as the input:

```
Raw Corpus A:
### User
input question: Write a sentence not in English.
output answer: от того, что появляется белая лилия, которая цветет с
↪   июня по сентябрь, который распространяется на соленый воздух
↪   нежный аромат.
input question: Write a sentence not in English.
output answer: При других заболеваниях лечение может идти до 1 года
↪   и далее. Биокоррекция Васильева может осуществляться как в
↪   клинических условиях, так и амбулаторно.
input question: Write a sentence not in English.
output answer: Насколько существенно ее воздействие на важные
↪   бизнес-процессы?
input question: Write a sentence not in English.
output answer:
### Assistant
Ваш входной идентификационный код будет также Вашим кодовым именем
↪   при участии в конкурсе. Информация про введение данных будет
↪   отослана Вам по электронной почте FTP с адресом и паролем.

Raw Corpus B:
### User
input question: Write a sentence not in English.
output answer: Отель Fayal Resort Hotel будет показан на карте, если
↪   Вы включите JavaScript.

Q: Translate "How to eat: Just drink it." to Russian?
Yes: Как употреблять: просто выпейте!

[Q]: За 72 года своего существования "Башнефть" сформировалась как
↪   мощный многопрофильный нефтегазодобывающий комплекс.

Translate this to English?
[A]: Within 72 years of its existence, Bashneft has become powerful
↪   multi-structural oil and gas extracting enterprise.

Question:
Для Профессионального комплекта существуют корпоративные решения с
↪   возможностью определять состав дополнительных специализированных
↪   словарей.

Could you please translate this to English?
Answer:
For the Professional package there exist enterprise solutions with
↪   the possibility to determine the composition of special
↪   dictionaries.
```

```
test: Балахтинский район
English?

translation: Balakhtinsky District

input question: Write a sentence not in English.
output answer:
### Assistant
Лангоун, Майкл
```

Step 1: Domain Analysis (Prompt template is shown in Figure 10)

```
LLM Response:
{
    "corpus_A_domain": "Language Translation",
    "corpus_B_domain": "Language Translation",
    "corpus_A_key_terms_list": [
        "sentence construction",
        "foreign language",
        "biocorrection",
        "business processes"
    ],
    "corpus_B_key_terms_list": [
        "translation",
        "enterprise solutions",
        "specialized dictionaries",
        "multilingual"
    ],
    "matching_rules_derived_from_corpus_A": "Questions from the '###
    ↪  User' section often ask for non-English sentences or
    ↪  contexts, and answers in the '### Assistant' section provide
    ↪  translations or related foreign phrases.",
    "matching_rules_derived_from_corpus_B": "The '### User'
    ↪  questions request either translations or sentences in
    ↪  non-English, while the '### Assistant' responses deliver
    ↪  translations or contextual explanations in both English and
    ↪  Russian.",
    "relationship": "related-domain"
}
```

Once the relationship is identified, the corresponding fusion strategy is selected (all strategies are provided in Figure 11). The MCG template (Figure 13) is then applied to generate the initial merged corpus.

Step 2 & 3: Strategy Selection & Merged Corpus Generation

```
LLM Response:
{
    "overall_response": [
        {
            "domain_type": "related-domain",
            "applied_strategy": "conceptual_analogy",
```

```
1680        "corpus_fusion_variant": "### User\nIn the realm of
1681    ↪    Language Translation, both corpora emphasize the
1682    ↪    importance of constructing sentences in foreign
1683    ↪    languages. For instance, corpus A provides examples
1684    ↪    of non-English sentences, while corpus B focuses on
1685    ↪    translating these sentences into English. This
1686    ↪    highlights the interconnectedness of language
1687    ↪    processing and multilingual communication. How can
1688    ↪    we further enhance the understanding of cultural
1689    ↪    linguistics in translation?\n\n### Assistant\nWe can
1690    ↪    enhance the understanding of cultural linguistics in
1691    ↪    translation by incorporating cultural context into
1692    ↪    the translation process, ensuring that idiomatic
1693    ↪    expressions and cultural references are accurately
1694    ↪    conveyed."
        },
1695    {
1696        "domain_type": "related-domain",
1697        "applied_strategy": "process_mapping",
1698        "corpus_fusion_variant": "### User\nWhen considering
1699    ↪    enterprise solutions for Language Translation,
1700    ↪    corpus B discusses specialized dictionaries that can
1701    ↪    aid in translation tasks. Meanwhile, corpus A
1702    ↪    mentions biocorrection and its relevance to business
1703    ↪    processes. This suggests a need for a structured
1704    ↪    approach to integrating these specialized tools into
1705    ↪    translation workflows. What are the best practices
1706    ↪    for implementing such solutions in a multilingual
1707    ↪    environment?\n\n### Assistant\nBest practices for
1708    ↪    implementing specialized tools in a multilingual
1709    ↪    environment include conducting thorough needs
1710    ↪    assessments, training staff on the use of these
1711    ↪    tools, and continuously evaluating the effectiveness
        ↪    of the solutions in improving translation accuracy."
1712        },
1713    {
1714        "domain_type": "related-domain",
1715        "applied_strategy": "term_bridging",
1716        "corpus_fusion_variant": "### User\nBoth corpora
        ↪    highlight key terms such as 'sentence construction'
1717    ↪    from corpus A and 'translation' from corpus B. This
1718    ↪    overlap indicates a shared focus on the mechanics of
1719    ↪    language. Additionally, terms like 'foreign
1720    ↪    language' and 'multilingual' suggest a broader
1721    ↪    context of language processing. How can we leverage
1722    ↪    these shared terms to create a more cohesive
1723    ↪    language translation framework?\n\n### Assistant\nWe
1724    ↪    can leverage these shared terms by developing a
1725    ↪    unified glossary that defines each term in the
1726    ↪    context of both corpora, facilitating better
1727    ↪    communication and understanding among language
        ↪    professionals."
1728        }
1729    ]
1730  }
1731
```

Since we provide three fusion strategies for each relationship, three different merged corpora will be generated. However, for clarity of presentation, we illustrate the process using only

the initial merged corpus produced by the 'term_bridge' strategy. The following step is to obtain the **Symbolic Loss** for this merged corpus.

Step 4: Information Completeness Detection (Prompt template is shown in Figure 14.)

```
LLM Response:
{
    "technical_report": {
        "term_coverage": {
            "included": {
                "corpus_A_key_terms_list": [
                    "sentence construction",
                    "foreign language",
                    "biocorrection",
                    "business processes"
                ],
                "corpus_B_key_terms_list": [
                    "translation",
                    "enterprise solutions",
                    "specialized dictionaries",
                    "multilingual"
                ]
            },
            "missing": {
                "corpus_A_key_terms_list": [],
                "corpus_B_key_terms_list": []
            }
        },
        "question_analysis": {
            "question_exists": {
                "question_type": "open-ended",
                "needs_extra_knowledge_or_background_supplementation": {
                    "context_contain": "The '### User' section discusses
↪   the overlap of key terms and asks how to
↪   leverage them for a cohesive framework.",
                    "context_missing": "The '### User' section lacks
↪   specific examples or details on what aspects of
↪   the shared terms should be leveraged, which
↪   could help in formulating a more precise
↪   answer."
                }
            }
        },
        "answer_quality": {
            "if_needs_to_re_answer": {
                "explanation": "The answer provided in the '###
↪   Assistant' section does not directly address the
↪   last question from the '### User' section, which
↪   asks how to leverage shared terms to create a
↪   cohesive language translation framework. The
↪   response is more about developing a glossary rather
↪   than providing a cohesive framework."
            }
        }
    }
}
```

According to the resultant symbolic loss, the current merged corpus has **missing necessary context in the '### User' section**, and the '### **Assistant' section does not provide a direct and clear answer** to the question from '### User' section. Thus,

the initial prompt template for Merged Corpus Generation (Figure 13) is updated to the version shown in Figure 24 to generate the enhanced merged corpus.

Step 5: Enhanced Corpus Generation (Prompt template is shown in Figure 24.)

```
LLM Response:
{
    "enhanced_corpus": "### User\nBoth corpora highlight key terms
    ↪   such as 'sentence construction' from corpus A and
    ↪   'translation' from corpus B. This overlap indicates a shared
    ↪   focus on the mechanics of language. Additionally, terms like
    ↪   'foreign language' and 'multilingual' suggest a broader
    ↪   context of language processing. How can we leverage these
    ↪   shared terms, particularly 'sentence construction' and
    ↪   'translation', to create a more cohesive language
    ↪   translation framework? For instance, what specific
    ↪   strategies can we implement that utilize 'business
    ↪   processes' and 'specialized dictionaries' to enhance our
    ↪   approach to multilingual translation?\n\n### Assistant\nWe
    ↪   can leverage these shared terms by developing a unified
    ↪   glossary that defines each term in the context of both
    ↪   corpora, facilitating better communication and understanding
    ↪   among language professionals. In addition, we should
    ↪   consider implementing strategies that integrate 'business
    ↪   processes' with 'sentence construction' techniques to
    ↪   streamline translation workflows. Utilizing 'specialized
    ↪   dictionaries' can also enhance accuracy in translations,
    ↪   particularly in a multilingual context. By aligning these
    ↪   elements, we can create a cohesive language translation
    ↪   framework that not only addresses the mechanics of language
    ↪   but also improves the overall efficiency and effectiveness
    ↪   of translation practices."
}
```

Again, the following step is to obtain the **Symbolic Loss** for the resultant enhanced merged corpus from the last step.

Step 4: Information Completeness Detection (ICD) For Enhanced Corpus (Prompt template has been shown in Figure 14.)

```
LLM Response:
{
    "technical_report": {
        "term_coverage": {
            "included": {
                "corpus_A_key_terms_list": [
                    "biocorrection",
                    "business processes",
                    "foreign language"
                ],
                "corpus_B_key_terms_list": [
                    "translation",
                    "enterprise solutions",
                    "specialized dictionaries",
                    "multilingual"
                ]
            },
            "missing": {
                "corpus_A_key_terms_list": [
                    "sentence construction"
```

```
                    ],
                    "corpus_B_key_terms_list": []
                }
            },
            "question_analysis": {
                "question_exists": {
                    "question_type": "open-ended",
                    "needs_extra_knowledge_or_background_supplementation":
                    ↪   "nope"
                }
            },
            "answer_quality": {
                "if_needs_to_re_answer": "nope"
            }
        }
    }
```

Compared to the symbolic loss of the initial merged corpus, the current symbolic loss only reflects missing key terms from source corpus A. This indicates that the enhanced merged corpus contains clear questions and answers and provides sufficient background knowledge to address the questions. The only remaining issue is that the coverage of key terms is slightly insufficient. Nonetheless, compared to the initial merged corpus, the current version represents a significant improvement. Therefore, to address the issue of missing key terms, we update the current template to the version shown in Figure 17 to generate a better merged corpus covering a broader range of domains.

Step 5: Enhanced Corpus Generation (Prompt template is shown in Figure 17.)

```
LLM Response:
{
    "enhanced_corpus": "### User\nBoth corpora highlight key terms such
    ↪   as 'sentence construction' from corpus A and 'translation' from
    ↪   corpus B. This overlap indicates a shared focus on the mechanics
    ↪   of language. Additionally, terms like 'foreign language' and
    ↪   'multilingual' suggest a broader context of language processing.
    ↪   How can we leverage these shared terms, particularly 'sentence
    ↪   construction' and 'translation', to create a more cohesive
    ↪   language translation framework? For instance, what specific
    ↪   strategies can we implement that utilize 'business processes'
    ↪   and 'specialized dictionaries' to enhance our approach to
    ↪   multilingual translation?\n\n### Assistant\nWe can leverage
    ↪   these shared terms by developing a unified glossary that defines
    ↪   each term in the context of both corpora, facilitating better
    ↪   communication and understanding among language professionals. In
    ↪   particular, the concept of 'sentence construction' is crucial as
    ↪   it involves the mechanisms of structuring sentences effectively
    ↪   in various languages. Additionally, we should consider
    ↪   implementing strategies that integrate 'business processes' with
    ↪   'sentence construction' techniques to streamline translation
    ↪   workflows. For example, application scenarios such as utilizing
    ↪   'specialized dictionaries' can enhance accuracy in translations,
    ↪   particularly in a multilingual context. By aligning these
    ↪   elements, we can create a cohesive language translation
    ↪   framework that not only addresses the mechanics of language but
    ↪   also improves the overall efficiency and effectiveness of
    ↪   translation practices."
}
```

Again, the following step is to obtain the **Symbolic Loss** for the resultant enhanced merged corpus from the last step.

Step 4: Information Completeness Detection (ICD) For Enhanced Corpus

```
LLM Response:
{
    "technical_report": {
        "term_coverage": {
            "included": {
                "corpus_A_key_terms_list": [
                    "biocorrection",
                    "business processes",
                    "foreign language",
                    "sentence construction"
                ],
                "corpus_B_key_terms_list": [
                    "enterprise solutions",
                    "multilingual",
                    "specialized dictionaries",
                    "translation"
                ]
            },
            "missing": {
                "corpus_A_key_terms_list": [],
                "corpus_B_key_terms_list": []
            }
        },
        "question_analysis": {
            "question_exists": {
                "question_type": "open-ended",
                "needs_extra_knowledge_or_background_supplementation":
                ↪   "nope"
            }
        },
        "answer_quality": {
            "if_needs_to_re_answer": "nope"
        }
    }
}
```

According to the current symbolic loss, all conditions have been satisfied: no key terms are missing, the '### User' section provides a clear question along with the necessary background knowledge to answer it, and the '### Assistant' section provides a precise answer to the question. Thus, **the resultant merged corpus successfully completes the entire Cycle 1 process**. The next step is to double-check the quality of the answer in the '### Assistant' section of the current resultant merged corpus (obtaining the **symbolic loss** from the FAC operator).

Step 6: Final Answer Check (Prompt template has been shown in Figure 12.)

```
LLM Response:
{
    "if_containing_direct_answer": {
```

```
        "direct_answer": "We can leverage these shared terms by
    ↪    developing a unified glossary that defines each term in
    ↪    the context of both corpora, facilitating better
    ↪    communication and understanding among language
    ↪    professionals. Additionally, we should consider
    ↪    implementing strategies that integrate 'business
    ↪    processes' with 'sentence construction' techniques to
    ↪    streamline translation workflows. For example, utilizing
    ↪    'specialized dictionaries' can enhance accuracy in
    ↪    translations, particularly in a multilingual context."
    },
    "if_containing_information_needed_to_delete": "nope"
}
```

According to the symbolic loss from the FAC operator, the current '### Assistant' section contains a direct answer to the question from '### User' and includes no irrelevant or redundant information, indicating that **the current merged corpus has successfully passed the entire Cycle 2 process**.

**Final Merged Corpus**   We present the input corpora pair and their resultant merged corpus (**'enhanced_corpus'**) below:

```
{
    "raw_corpus_A": "### User\ninput question: Write a sentence not
    ↪    in English.\noutput answer: от того, что появляется белая
    ↪    лилия, которая цветет с июня по сентябрь, который
    ↪    распространяется на соленый воздух нежный аромат.\ninput
    ↪    question: Write a sentence not in English.\noutput answer:
    ↪    При других заболеваниях лечение может идти до 1 года и
    ↪    далее. Биокоррекция Васильева может осуществляться как в
    ↪    клинических условиях, так и амбулаторно.\ninput question:
    ↪    Write a sentence not in English.\noutput answer: Насколько
    ↪    существенно ее воздействие на важные бизнес-процессы?\ninput
    ↪    question: Write a sentence not in English.\noutput
    ↪    answer:\n### Assistant\nВаш входной идентификационный код
    ↪    будет также Вашим кодовым именем при участии в конкурсе.
    ↪    Информация про введение данных будет отослана Вам по
    ↪    электронной почте FTP с адресом и паролем.",
    "raw_corpus_B": "### User\ninput question: Write a sentence not
    ↪    in English.\noutput answer: Отель Fayal Resort Hotel будет
    ↪    показан на карте, если Вы включите JavaScript.\n\n\nQ:
    ↪    Translate \"How to eat: Just drink it.\" to Russian?\nYes:
    ↪    Как употреблять: просто выпейте!\n\n\n[Q]: За 72 года своего
    ↪    существования \"Башнефть\" сформировалась как мощный
    ↪    многопрофильный нефтегазодобывающий комплекс.\n\nTranslate
    ↪    this to English?\n[A]: Within 72 years of its existence,
    ↪    Bashneft has become powerful multi-structural oil and gas
    ↪    extracting enterprise.\n\n\nQuestion:\nДля Профессионального
    ↪    комплекта существуют корпоративные решения с возможностью
    ↪    определять состав дополнительных специализированных
    ↪    словарей.\n\nCould you please translate this to
    ↪    English?\nAnswer:\nFor the Professional package there exist
    ↪    enterprise solutions with the possibility to determine the
    ↪    composition of special dictionaries.\n\n\ntest: Балахтинский
    ↪    район\nEnglish?\n\ntranslation: Balakhtinsky
    ↪    District\n\n\ninput question: Write a sentence not in
    ↪    English.\noutput answer:\n### Assistant\nЛангоун, Майкл",
```

```
      "enhanced_corpus": "### User\nBoth corpora highlight key terms
  ↪   such as 'sentence construction' from corpus A and
  ↪   'translation' from corpus B. This overlap indicates a shared
  ↪   focus on the mechanics of language. Additionally, terms like
  ↪   'foreign language' and 'multilingual' suggest a broader
  ↪   context of language processing. How can we leverage these
  ↪   shared terms, particularly 'sentence construction' and
  ↪   'translation', to create a more cohesive language
  ↪   translation framework? For instance, what specific
  ↪   strategies can we implement that utilize 'business
  ↪   processes' and 'specialized dictionaries' to enhance our
  ↪   approach to multilingual translation?\n\n### Assistant\nWe
  ↪   can leverage these shared terms by developing a unified
  ↪   glossary that defines each term in the context of both
  ↪   corpora, facilitating better communication and understanding
  ↪   among language professionals. In particular, the concept of
  ↪   'sentence construction' is crucial as it involves the
  ↪   mechanisms of structuring sentences effectively in various
  ↪   languages. Additionally, we should consider implementing
  ↪   strategies that integrate 'business processes' with
  ↪   'sentence construction' techniques to streamline translation
  ↪   workflows. For example, application scenarios such as
  ↪   utilizing 'specialized dictionaries' can enhance accuracy in
  ↪   translations, particularly in a multilingual context. By
  ↪   aligning these elements, we can create a cohesive language
  ↪   translation framework that not only addresses the mechanics
  ↪   of language but also improves the overall efficiency and
  ↪   effectiveness of translation practices."
}
```

## H  MERGED CORPUS EXAMPLE

For illustrative purposes, we randomly select three merged corpora in this section: two derived from Intra-Cluster Fusion and one from Inter-Cluster Fusion.

### H.1  MERGED CORPUS FROM INTRA-CLUSTER FUSION

As illustrated in Figure 7, the two raw corpora originate from the same cluster and contain a large amount of overlapping surface-level information. One focuses on locating positions with alphabetical elements in the input list, while the other counts the total number of such elements. However, both fail to explicitly convey the underlying conceptual principles. This indicates that these raw corpora have very limited capacity to guide the LLM in developing a deeper, principle-based understanding. In contrast, our merged corpus not only makes full use of the background knowledge provided by the raw corpora, but also includes concrete procedural steps in the answer section. Moreover, it explicitly references relevant technical domains and concepts, such as data structures, list traversal, and element evaluation. As a result, our merged corpus is clearly better positioned to guide the LLM toward deeper reasoning and generate outputs that are closer to ground-truth inferences.

Similarly, in another five-to-one corpora fusion example (see Figure 9), all five raw corpora focus on the same task, "Generate a 5-star review for a given software." However, none of them provide any background information about the software itself. While the last two raw corpora contain multiple Q-A pairs, there is little to no semantic connection between the pairs, and in some cases, the answers appear unrelated to the corresponding questions. Such fragmented and context-deficient corpora may negatively impact the LLM's reasoning capabilities. By contrast, our merged corpus not only retains key features from the original raw corpora in the question formulation but also provides substantial contextual background.

Furthermore, the answer section offers clear directions and actionable steps tailored to the question, significantly enhancing the expressive power and utility of each individual merged corpus.

## H.2 Merged Corpus From Inter-Cluster Fusion

Furthermore, as shown in Figure 8, the two raw corpora come from entirely unrelated domains, one focuses on official languages and industries, while the other discusses leeks and grass. In contrast, our merged corpus introduces a hypothetical scenario that not only incorporates elements from both raw corpora, such as Spanish, leeks, and grass, but also raises a more profound question: How do cultural values and language influence agrotourism? The answer section goes further by outlining a concrete strategic plan in response. This further validates the capacity of our merged corpora to guide the LLM in exploring a broader range of reasoning possibilities.

# I Future Work

Looking ahead, we plan to address the boundary cases where fusion consistently fails, particularly for highly structured inputs such as tables, code snippets, and mathematical expressions. Our current approach sidesteps this challenge by filtering out mathematics- and coding-related corpora, but a more general solution is needed. A promising direction is to develop a unified fusion paradigm that can seamlessly handle both structured–structured and structured–unstructured data pairs, enabling robust corpus integration across diverse domains. Additionally, we intend to adopt more up-to-date benchmarks covering a wider range of domains and tasks as evaluation sets, in order to more thoroughly assess LLM performance across different fields.

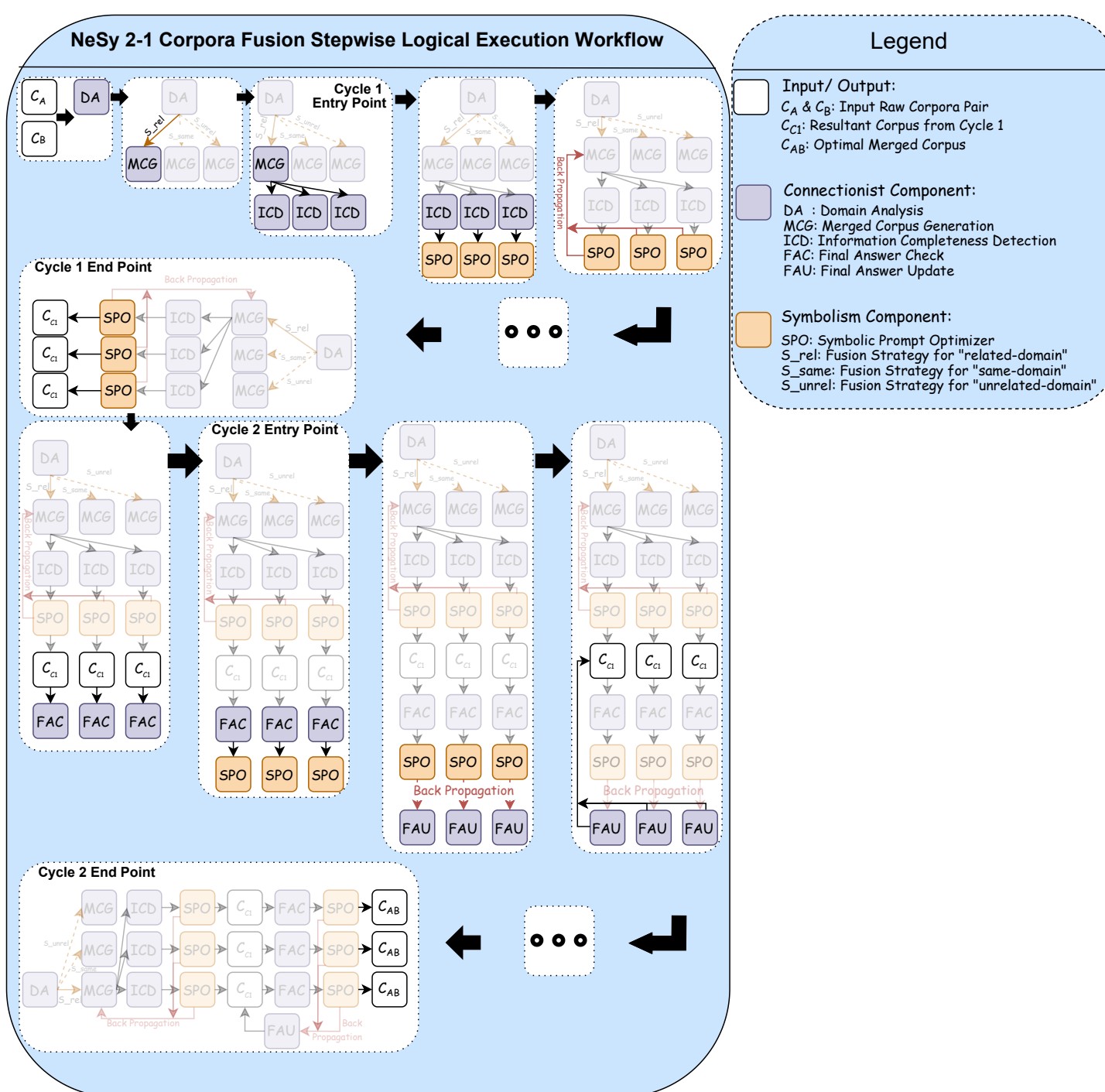

Figure 5: Neural-Symbolic Two-To-One Corpora Fusion Stepwise Logical Execution Workflow

## LLM Rating Prompt Template From DS^2

- - - - - - - - - - - - - - - - - - - - - - - - - - - - - - - - - - - -

<System Prompt>: As a data quality estimator, your task is to assess the quality of the data sample based on the criteria: Rarity, Complexity, and Informativeness. Please rate the sample on a scale from 1 to 10 for each criterion, and return an overall rating on a scale from 1 to 10, where a higher score indicates a higher level of quality. Ensure that the ratings are not overly concentrated around a specific score. If multiple samples have similar qualities, consider spreading the scores more evenly to reflect subtle differences.

<User Prompt>: Now, please carefully evaluate the following data sample and return the integral evaluation scores using the JSON format:
      {"Rarity": <number, 1-10>,
          "Complexity": <number, 1-10>,
          "Informativeness": <number, 1-10>,
          "Overall rating": <number, 1-10>}

Instruction: [Instruction]
Input: [Input]
Response: [Response]

Figure 6: LLM-rating Prompt Template From DS$^2$ (Pang et al., 2025)

# Two-to-One Corpora Fusion Example

**Two Raw Corpora:**

"### User\nYou will be given a definition of a task first, then some input of the task.\nIn this task, you are given an input list. A list contains several comma-separated items written within brackets. You need to return the position of all the alphabetical elements in the given list in order. Assume the position of the 1st element to be 1. Return -1 if no alphabetical element is in the list.\n\n['2385', 'i', 'Y', 'T', '4011', 'f', 'Q']\nOutput:\n### Assistant\n2, 3, 4, 6, 7",

"### User\nQ: In this task, you are given an input list. A list contains several comma-separated items written within brackets. You need to return the count of all the alphabetical elements in the given list. Return 0 if no alphabetical element in the list\n['1029', '9675', '5911', '2311', '781', '1357', 'T', 'p', '2795', 'H', 'G', 'a', 'd', 'u', '6253', 'c', '2287', 'g', 'c', '2881', 'D', '3405', '7357', '7193', 'y', '6545']\nA:\n### Assistant\n12"

**Merged Corpus:**

### User\nIn the context of Data Structures and Algorithms, we often analyze lists to extract specific information. For instance, when given an input list like ['2385', 'i', 'Y', 'T', '4011', 'f', 'Q'], we can identify the positions of alphabetical elements. This process is similar to counting how many alphabetical elements are present in another list, such as ['1029', '9675', '5911', '2311', '781', '1357', 'T', 'p', '2795', 'H', 'G', 'a', 'd', 'u', '6253', 'c', '2287', 'g', 'c', '2881', 'D', '3405', '7357', '7193', 'y', '6545']. How do these two tasks relate in terms of their underlying principles, particularly regarding the concepts of identifying the position of alphabetical elements and counting them in a list? What are the fundamental similarities in their methodologies, and how do they utilize the input list to derive meaningful output?\n\n
### Assistant\nBoth tasks involve processing a list to extract information based on specific criteria, highlighting the importance of character analysis and element identification in list processing. Specifically, identifying the positions of alphabetical elements requires traversing the input list and checking each element against the criteria of being alphabetical, which results in a list of indices where these elements are found. In contrast, counting the alphabetical elements involves a similar traversal but focuses on tallying the total number of elements that meet the alphabetical criteria, ultimately yielding a single numerical output. Thus, both tasks share the underlying principle of list traversal and element evaluation, emphasizing the significance of systematic analysis in data structures.

Figure 7: Merged Corpus Example from Two-To-One Intra-Cluster Fusion

# Two-to-One Corpora Fusion Example

- - - - - - - - - - - - - - - - - - - - - - - - - - - - - - - - - - - - - -

**Two Raw Corpora:**

"### User\nUsing available sources, can you provide the current population, official language, and top 3 industries of the city that serves as the administrative and political center of Bolivia?\n### Assistant\nThe city that serves as the administrative and political center of Bolivia is La Paz.\nAccording to the World Population Review, as of 2021, the population of La Paz is approximately 2.8 million people.\nThe official languages of Bolivia are Spanish, Quechua, and Aymara. However, Spanish is the most widely spoken language in La Paz.\nThe top three industries in La Paz are finance and banking, tourism, and manufacturing.",

"### User\nFor this chain-of-thought reasoning and answer, what was the question?\nChives grow upwards in thin green cylindrical shoots. Grass grows upwards in thin green flat shoots.\n A: yes\n\n### Assistant\nQ: Could chives be mistaken for grass?"

- - - - - - - - - - - - - - - - - - - - - - - - - - - - - - - - - - - - - -

**Merged Corpus:**

### User\nImagine a scenario where La Paz, with its 2.8 million residents, is hosting an international conference on botany. The discussions revolve around the similarities between local plants like chives and grass, which both grow upwards but in different forms—cylindrical and flat shoots, respectively. As the conference unfolds, participants explore how the official languages of Bolivia, primarily Spanish, Quechua, and Aymara, influence the understanding of these plants in cultural contexts. What would be the key takeaways from such a conference regarding the relationship between local flora and the economic sectors in La Paz, particularly in terms of how cultural perceptions and language might affect industries such as agriculture and tourism?\n\n### Assistant\nKey takeaways would likely include the importance of local flora in promoting tourism, particularly through eco-tourism initiatives that highlight unique plant species. Additionally, the role of agriculture in economic stability would be emphasized, showcasing how crops like chives can be marketed both locally and internationally. Furthermore, the conference would reveal how cultural identity, shaped by the official languages of Bolivia, influences the perception and utilization of these plants in various industries, suggesting that a deeper understanding of these relationships could enhance economic opportunities in La Paz."

Figure 8: Merged Corpus Example from Two-To-One Inter-Cluster Fusion

## Five-to-One Corpora Fusion Example

- - - - - - - - - - - - - - - - - - - - - - - - - - - - - - - - - - - - - -

**Five Raw Corpora:**

### User\nQuestion: Generate a 5-star review (1 being lowest and 5 being highest) about an app with package com.google.android.gms.\nAnswer:\n### Assistant\n faster,

### User\nQuestion: Generate a 5-star review (1 being lowest and 5 being highest) about an app with package com.google.android.gms.\nAnswer:\n### Assistant\nIts very great,

### User\nAnswer the following question: Generate a 5-star review (1 being lowest and 5 being highest) about an app with package org.telegram.messenger.\nAnswer: \n### Assistant\nwelldone,

### User\nQuestion: Generate a 5-star review (1 being lowest and 5 being highest) about an app with package com.google.android.gms.\nAnswer: i love it\n\nQuestion: Generate a 5-star review (1 being lowest and 5 being highest) about an app with package com.google.android.gms.\nAnswer: I can't think about it all the best thing to happen to you and your not going out with a smile and laugh at me for the first place in my life \"\"...😊\n\nQuestion: Generate a 5-star review (1 being lowest and 5 being highest) about an app with package com.google.android.gms.\nAnswer:\n ### Assistant\nNAGABABU VASANTHABA 333",

"### User\ninput: Please answer the following: Generate a 5-star review (1 being lowest and 5 being highest) about an app with package com.numix.icons_circle.\n ++++++++++\noutput: Wow Really like it.\n\n\ninput: Please answer the following: Generate a 4-star review (1 being lowest and 5 being highest) about an app with package org.telegram.messenger.\n++++++++++\noutput: Good\n\n\ninput: Please answer the following: Generate a 5-star review (1 being lowest and 5 being highest) about an app with package com.google.android.gms.\n++++++++++\noutput: salidos\n\n\n\ninput: Please answer the following: Generate a 5-star review (1 being lowest and 5 being highest) about an app with package com.mgaetan89.showsrage.\n++++++++++\n output:\n### Assistant\nNice redesign! This app works great! And the new icon is beautiful!"

- - - - - - - - - - - - - - - - - - - - - - - - - - - - - - - - - - - - - -

**Merged Corpus:**

### User\nIn the realm of mobile applications, user feedback acts as a compass guiding developers toward user satisfaction. Just as a well-crafted itinerary enhances a journey, a '5-star review' serves as a beacon of success for an app like com.mgaetan89.showsrage. Users often express their delight, saying things like, \"Nice redesign! This app works great!\" This feedback is crucial for potential users navigating the digital landscape. How can we further explore the metaphor of apps as companions in this journey, and what other terms can we bridge to deepen our understanding of user experiences?\n\n### Assistant\nWe can explore the metaphor of apps as companions by identifying additional terms like 'user-friendly experience' and 'companion apps' that enhance user satisfaction. By analyzing how these terms relate to user feedback, we can gain insights into the emotional connections users form with their apps.

Figure 9: Merged Corpus Example from Five-To-One Intra-Cluster Fusion

## Domain Analysis (DA)

----------------------------------------------------------------

## Role
You are a domain relationship analyzer

## Task
Analyze two QA corpora and output domain relationship analysis in JSON format

## Input
- corpus_A: Full QA block including ### User and ### Assistant
- corpus_B: Full QA block including ### User and ### Assistant

## Output Requirements
1. Identify primary domains for both corpora using professional terminology
2. Extract as many key technical terms as possible from both corpora
3. Identify and extract the potential matching rules or patterns that align questions from the '### User' section with corresponding answers in the '### Assistant' section for both corpora
4. Calculate semantic similarity between corpora based on conceptual overlap (0-1 scale)
5. Propose candidate **bridging_concepts** that enable cross-domain integration
   - Specifically, generate potential bridging concepts that incorporate and unify the key terms from both the **corpus_A_key_terms_list** and the **corpus_B_key_terms_list**
6. Refer to the '## Example Output' section below, and make sure all the key must have a valid value
7. Please ensure that **none of your responses** contain any information related to **sexual explicitness, violence, drug use, threats to social order, or racial prejudice**.

## Example Output
{
    "corpus_A_domain": "Network Security",
    "corpus_B_domain": "Medical Device Regulation",
    "corpus_A_key_terms_list": ["encryption", "firewall", "VPN", ...],
    "corpus_B_key_terms_list": ["sterilization", "FDA", "compliance", ...],
    "matching_rules_derived_from_corpus_A":
        "Provide a detailed description of the potential matching rules or patterns that align questions from the '### User' section with corresponding answers in the '### Assistant' section within **corpus_A**",
    "matching_rules_derived_from_corpus_B":
        "Provide a detailed description of the potential matching rules or patterns that align questions from the '### User' section with corresponding answers in the '### Assistant' section within **corpus_B**",
    "relationship": "same-domain"/ "related-domain"/ "unrelated-domain"
}

## Input
- corpus_A: {corpus_A from user input}
- corpus_B: {corpus_B from user input}

Figure 10:   Prompt template for the LLM-invoking Domain Analysis (DA) Operator

## Symbolic Fusion Strategy Selection (SS)

```
"same-domain": {{
        "knowledge_merging":
            "Combine complementary knowledge points from corpora within the same
            domain to create comprehensive expertise integration",
        "procedure_extension":
            "Enhance operational workflows by integrating detailed steps from
            multiple sources within the same field",
        "case_integration":
            "Develop composite scenarios that unify specialized cases from different
            sub-domains"
}},
"related-domains": {{
        "conceptual_analogy":
            "Establish cross-domain connections through abstract principle
            similarities",
        "process_mapping":
            "Adapt standard processes from one domain to another's framework
            while preserving core logic",
        "term_bridging":
            "Create conceptual links through shared terminology with domain-specific
            interpretations"
}},
"unrelated-domains": {{
        "creative_metaphor":
            "Construct innovative connections using figurative language and symbolic
            representations",
        "hypothetical_scenario":
            "Design artificial situations that force meaningful interaction between
            disparate domains",
        "structural_parallelism":
            "Identify and leverage formal pattern similarities in knowledge
            organization"
}}
```

Figure 11: Symbolic fusion strategy definition for Strategy Selection (SS) Operator

# Final Answer Check (FAC)

- - - - - - - - - - - - - - - - - - - - - - - - - - - - - - - - - - - - - - -

## Role
You are a cross-corpus fusion quality auditor.

## Task
Given the merged corpus containing '### User' and '### Assistant' sections,
assess the quality of **the last '### Assistant' section only** (if the merged
corpus contains multiple '### Assistant' sections) following the
'### Evaluation Criteria':

### Evaluation Criteria For the '### Assistant' section:
1. **Direct Response**:
   - Does the '### Assistant' section offer a clear answer to the unanswered
     question from the '### User' section?
2. **Content Relevance**:
   - Does the '### Assistant' section contain unnecessary, redundant, or unrelated
     information?

### Input Merged Corpus:
<begin>
{merged_corpus}
<end>

### Expected Output Structure:
{{
   "if_containing_direct_answer":
      "nope" (indicating that the last '### Assistant' section **does not provide an
      answer** to the final question posed in the last '### User' section)
   /
   "if_containing_direct_answer": {{
      'direct_answer':
            "Extracting only the direct, complete answer from the last
            '### Assistant' section, ensuring that the extracted information is both
            clear and coherent."
   }},
   "if_containing_information_needed_to_delete":
      "nope" (indicating that the last '### Assistant' section provides
      **a perfect answer** to the final question from the last '### User' section,
      **with no redundant or irrelevant information**)
   /
   "if_containing_information_needed_to_delete": {{
      "information_needs_to_remove":
            "Extracting all redundant, irrelevant, or unnecessary information from the
            last '### Assistant' section that does not contribute to answering the
            final question in the last '### User' section"
   }}
}}

Figure 12:   Prompt template for the Final Answer Check (FAC) Operator

## Merged Corpus Generation (MCG)

- - - - - - - - - - - - - - - - - - - - - - - - - - - - - - - - - - - -

## Role
You are a strategy architect specializing in cross-corpus fusion, skilled in leveraging domain analysis to design effective merging strategies.

## Task
Utilize the provided domain analysis and selected strategies to generate three unique corpus fusion variants. Each variant must employ a distinct strategy to merge corpus_A and corpus_B, ensuring that no strategy is repeated. The fusion for each variant should not only integrate the two corpora but also reflect the specific domain characteristics identified in the analysis.

## Input
{{
    "raw_corpus_A": {raw_corpus_A},
    "raw_corpus_B": {raw_corpus_B}
}}

## Domain Analysis
{{
    "corpus_A_domain": {corpus_A_domain},
    ...
    "relationship": {relationship_label}
}}

## Selected Strategy
{strategy_dict_str}

## Output Requirements
- For all three corpus fusion variants generation:
    1. For the '### User' section:
        a. The '### User' section must conclude with an unanswered question
            - This section should integrate **essential context** with **one or more related, logically connected questions**.
            - If there are multiple questions in the newly generated '### User' section, **do make sure to provide the corresponding direct answer to each question except the final one**
        b. In the '### User' section, ensure that the background information is logically structured and coherently presented. The question posed should be directly related to the provided background, with a natural and seamless transition between the background information and the question, resulting in an overall smooth and readable flow.
    2. '### Assistant' section must provide a direct answer exclusively to the unanswered question posed in the '### User' section
    3. Preserve all key terms from both corpus_A_key_terms_list and corpus_B_key_terms_list in all three corpus fusion variants
    4. Adhere to the matching rules or patterns from both raw corpora (**matching_rules_A** and **matching_rules_B**), ensuring that the resulting matching rules or patterns explicitly encompass those from both original corpora
    5. Ensure logical coherence and semantic fluency throughout the content
    6. Utilize the **Selected Strategies** for the corpus fusion variants generation
    7. Strictly maintain:
        - The '### User' and '### Assistant' markers.
        - The formatting identical to that of the original corpora.
    8. Each corpus fusion variant **must include at leaset** one '### User' section and one '### Assistant' section
        - Which means the resultant corpus fusion variant may contain multiple '### User' and '### Assistant' section pairs if deemed indeed necessary.
    9. Please ensure that **none of your responses** contain any information related to **sexual explicitness, violence, drug use, threats to social order, or racial prejudice**.

## Example Output Structure
{{
    "overall_response": [
        {{
            "domain_type": "same-domain"/ "related-domain"/ "unrelated-domain"
            "applied_strategy": "creative_metaphor"/ ...,
            "corpus_fusion_variant":
                """### User
<The content in the ### User section may be a single question (with or without context) or a series of question-answer pairs that culminate in a final question, in accordance with the original corpora.>

### Instruction
<This section is optional. Sometimes the raw corpora include an Instruction section. For the newly generated corpus fusion variant, you may choose whether or not to include it.>

### Input
<This section is optional. Sometimes the raw corpora include an Input section. For the newly generated corpus fusion variant, you may choose whether or not to include it.>

### Assistant
<The content in the ### Assistant section must provide the answer to the latest question presented in the ### User section.>"""

        or you can also include multiple '### User' and '### Assistant' section pairs derived from some raw corpus if you deem it indeed necessary

            "corpus_fusion_variant":
                """### User
<The content in the ### User section may be a single question (with or without context) or a series of question-answer pairs that culminate in a final question, in accordance with the original corpora.>

### Assistant
<The content in the ### Assistant section must provide the answer to the latest question presented in the ### User section.>

...

### User
<The content in the ### User section may be a single question (with or without context) or a series of question-answer pairs that culminate in a final question, in accordance with the original corpora.>

### Assistant
<The content in the ### Assistant section must provide the answer to the latest question presented in the ### User section.>"""
        }},
        {{
            ...
        }},
        {{
            ...
        }}
    ]
}}

Figure 13: Prompt design for the Merged Corpus Generation (MCG) Operator.

**Information Completeness Detection (ICD)**

- - - - - - - - - - - - - - - - - - - - - - - - - - - - - - - - - -

## Role
You are a cross-corpus fusion quality auditor.

## Task
Evaluate the completeness of merged corpus content against strict quality criteria
and generate a detailed supplementation report.

### Evaluation Criteria:
1. **Key Term Coverage**
   - Verify inclusion of ALL key technical terms from:
     - Corpus A Key Technical Terms List: {corpus_A_key_terms_list}
     - Corpus B Key Technical Terms List: {corpus_B_key_terms_list}
   - Categorize terms as included/missing

2. **'### User' Section Quality Check**
   a. Unanswered Questions Presence Check:
     - Does the '### User' section end with an unanswered question?
   b. Unanswered Question Type Analysis:
     - Open-ended: This type of question normally does not have an unique golden
       answer. So, it requires **no additional context** (such as: why do you like
       Spring?).
     - Closed-ended: This type of question normally does have an unique golden
       answer. So, it **requires specific context** (such as: what is the first
       sentence of the input paragraph? The 'input' paragraph is the specific
       context in this case.).
   c. Background Provision:
     - For open-ended unanswered question: Verify self-contained background
     - For closed-ended unanswered question: Does the '### User' section provide
       sufficient background to address the unanswered question?
   d. Multiple Questions Handling:
     - If multiple questions are present, does '### User' section answer all except
       the final one explicitly?

3. **'### Assistant' Section Evaluation**:
   a. **Direct Response**:
     - Does the '### Assistant' section offer a clear answer to the unanswered
       question from the '### User' section?
   b. **Content Relevance**:
     - Does the '### Assistant' section contain unnecessary, redundant, or
       unrelated information?

4. **Matching Rules Or Patterns Verification**
   - Confirm that the merged corpus's mapping from the question (from '### User'
     section) to the answer (from '### Assistant' section) preserves the implicit
     patterns observed in both original corpora:
     - Matching Rules or Patterns derived from Corpus A:
       {matching_rules_derived_from_corpus_A}
     - Matching Rules or Patterns derived from Corpus B:
       {matching_rules_derived_from_corpus_B}

### Input Data:
- Source Corpus A:
<begin>
{corpus_A}
<end>

- Source Corpus B:
<begin>
{corpus_B}
<end>

- Merged Corpus:
<begin>
{merged_corpus}
<end>

### Output Requirements:
- Strict JSON format
- Detailed technical breakdown
- Missing elements must be explicitly listed

### Evaluation Process:
1. **Phase 1: Term Inventory Audit**
   a. Cross-reference terms from both corpora
   b. Generate inclusion/missing lists

2. **Phase 2: Question Analysis**
   a. Question existence verification
   b. Question type classification
   c. Background context assessment

3. **Phase 3: Answer Validation**
   a. Directness of answer to question
   b. Completeness for question type

### Example Output:
```
{{
  "technical_report": {{
    "term_coverage": {{
      "included": {{
        "corpus_A_key_terms_list": ["term_1", "term_3", ...],
        "corpus_B_key_terms_list": ["term_2", "term_3", ...]
      }},
      "missing": {{
        "corpus_A_key_terms_list": ["term_2", ...],
        "corpus_B_key_terms_list": ["term_1", ...]
      }}
    }},
    "question_analysis": {{
      "question_exists":
        "no_questions_found" (indicating that the '### User' section dose not
        provide any unanswered questions)
        /
      "question_exists": {{
        "question_type": "open-ended"/ "close-ended",
        "needs_extra_knowledge_or_background_supplementation": {{
          "context_contain":
            "Provide a detailed description about what context information had
            been provided by the '### User' section of the current merged
            corpus",
          "context_missing":
            "Provide a detailed explanation of the necessary context information
            that is still absent from the '### User' section, which is required
            to answer this question."
        }}
        /
        "needs_extra_knowledge_or_background_supplementation":
            "nope" (indicating that the context provided in the '### User'
            section is sufficient to answer the question without any additional
            background information)
      }}
    }},
    "answer_quality": {{
      "if_needs_to_re_answer":
        "nope" (indicating that the current answer provided in the
        '### Assistant' section had directly addressed **the last question**
        from the '### User' section)
        /
      "if_needs_to_re_answer": {{
        "explanation": "Offer a comprehensive rationale explaining why the answer
        provided in the '### Assistant' section does not adequately address the
        question posed in the '### User' section. This explanation should detail
        whether the response is incomplete, only partially addresses the question,
        or is entirely irrelevant."
      }}
    }}
  }}
}}
```

Figure 14: Prompt design for the Information Completeness Detection (ICD) Operator.

## Candidate FAU Prompt Template

**Symbolic Loss:**
**Omission of Directed Answer**

- - - - - - - - - - - - - - - - - - - - - - - - - - - - - - - - -

## Role
You are an expert assistant.

## Task
Below is a conversation that may contain one or more pairs of "### User" and "### Assistant" sections. The final "### User" section ends with an unanswered question. Please review the conversation and provide a concise, direct answer to that unanswered question without any unnecessary filler. Your answer should be concise and directly address that unanswered question.

## Input Conversation
{conversation}

## Expected Output Structure
{{
    "answer": "providing a direct answer to the unanswered question from the
              '### User' section only"
}}

Figure 15:   Candidate FAU Prompt Template for The Case of Omission of Directed Answer

**Candidate FAU Prompt Template**

**Symbolic Loss:**
**Existing Irrelevant or Redundant Information**

## Role
You are an expert assistant.

## Task
You are provided with a conversation that contains one or more pairs of "### User" and "### Assistant" sections. The final "### Assistant" section includes an answer that not only addresses the question from the last "### User" section but also contains redundant or irrelevant information. Additionally, you are given feedback specifying the **direct_answer** (the essential part to keep) and the **information_needs_to_remove** (the parts to discard).

Your task is to review the conversation and the feedback, then provide a revised answer that is concise and contains only the direct answer to the question from the last "### User" section, with all extraneous content removed.

## Input Conversation
{conversation}

## Feedback For The Answer From The Final '### Assistant' Section
{{
    "direct_answer": {direc_ans},
    "information_needs_to_remove": {removed_infor}
}}

## Expected Output Structure
{{
    "answer": "provide a revised answer that is concise and contains only the direct
                answer to the question from the last "### User" section, with all
                extraneous content removed."
}}

Figure 16: Candidate FAU Prompt Template for The Case of Existing Irrelevant or Redundant Information

## Candidate MCG Prompt Template

**Symbolic Loss:**
**Omission of Key Technical Terms**
- - - - - - - - - - - - - - - - - - - - - - - - - - - - - - - - - - - - -

## Role
You are a Corpus Enhancement Specialist.

## Task
Intelligently expand existing merged corpus based on term coverage reports while strictly adhering to the following rules:

### Input Data
1. Current Merged Corpus:
<begin>
{merged_corpus}
<end>

2. Term Coverage Report:
{{
    "included_terms_list": {included_terms_list},
    "missing_terms_list": {missing_terms_list}
}}

### Processing Rules
1. **Content Preservation Principle**
   - Preserve all unanswered questions from the '### User' section and their
     corresponding answers in the '### Assistant' section without alteration.
   - Ensure that any modifications to the existing content do not exceed 20%% of
     the original content.

2. **Term Integration Guidelines**
   - **Insertion of Missing Terms**: Insert each missing term from
     **missing_terms_list** using one of the following methods:
       a. Integrate the term naturally within an explanatory statement
          (e.g., "...which involves {{term}} mechanisms...").
       b. Incorporate the term into practical examples (e.g., "Application scenarios
          such as {{term}}...").

   - **Handling of Already Included Terms**: For every term listed in
     **included_terms_list** that is present in the current merged corpus, choose
     one of the following approaches:
       a. Retain the original content from the **Current Merged Corpus** if it is
          relevant to the term—meaning the content contains either an explicit
          mention or an implicit reference to the concept represented by the term.
       b. Rephrase the original content from the **Current Merged Corpus** that
          pertains to the term, ensuring that the revised version explicitly includes
          the term while also integrating all the missing terms.

   - **Prohibition**: Do not simply list terms without integrating them into the
                        context.

3. **Coherence Assurance**
   - Ensure that all newly inserted or rephrased content is seamlessly integrated
     using explicit transitional phrases (e.g., "Considering", "In light of", "Particular
     attention should be paid to", etc.).
   - Preserve the original paragraph structure to maintain the logical flow and
     organization of the content.

### Expected Output Sturcture
{{
    "enhanced_corpus": "### User\n...\n### Assistant\n...",
}}

Figure 17: Candidate MCG Prompt Template for The Case of Omission of Key Technical Terms

**Candidate MCG Prompt Template**

**Symbolic Loss:**
**Omission of Key Technical Terms & Directed Answer** – – – – – – – – – – – –

## Role
You are a Corpus Enhancement Specialist.

## Task
Intelligently expand existing merged corpus based on targeted supplementation
while preserving original structure

### Input Data
1. Current Merged Corpus:
<begin>
{merged_corpus}
<end>

2. Supplementation Requirements:
{{
    "terms_coverage": {{
        "included_terms_list": {included_terms_list},
        "missing_terms_list": {missing_terms_list}
    }},
    "answer_quality_feedback": {assistant_feedback}
}}

### Processing Rules
1. **Content Preservation Principle**
   - Preserve all unanswered questions from the '### User' section.
   - Ensure that any modifications to the existing content do not exceed 20%% of
     the original content.

2. **Term Integration Guidelines**
   - **Insertion of Missing Terms**: Insert each missing term from
     **missing_terms_list** using one of the following methods:
       a. Integrate the term naturally within an explanatory statement
          (e.g., "...which involves {{term}} mechanisms...").
       b. Incorporate the term into practical examples (e.g., "Application scenarios
          such as {{term}}...").

   - **Handling of Already Included Terms**: For every term listed in
     **included_terms_list** that is present in the current merged corpus, choose
     one of the following approaches:
       a. Retain the original content from the **Current Merged Corpus** if it is
          relevant to the term—meaning the content contains either an explicit
          mention or an implicit reference to the concept represented by the term.
       b. Rephrase the original content from the **Current Merged Corpus** that
          pertains to the term, ensuring that the revised version explicitly includes
          the term while also integrating all the missing terms.

   - **Prohibition**: Do not simply list terms without integrating them into the
                      context.

3. **Answer Regeneration Guidelines**
   - Regenerate only the responses in the '### Assistant' section.
   - Ensure that the regenerated answer explicitly addresses the unanswered
     questions from the '### User' section.
   - Utilize the insights from "answer_quality_feedback" to inform the regeneration
     process, ensuring that the newly regenerated answers do not repeat the issues
     identified in the feedback.
   - Maintain logical coherence and consistent terminology throughout the
     regenerated response.

4. **Coherence Assurance**
   - Ensure that all newly inserted or rephrased content is seamlessly integrated
     using explicit transitional phrases (e.g., "Considering", "In light of", "Particular
     attention should be paid to", etc.).
   - Preserve the original paragraph structure to maintain the logical flow and
     organization of the content.

### Expected Output Sturcture
{{
    "enhanced_corpus": "### User\n...\n### Assistant\n...",
}}

Figure 18: Candidate MCG Prompt Template for The Case of Omission of Key Technical Terms & Directed Answer

**Candidate MCG Prompt Template**

**Symbolic Loss:**
**Omission of Key Technical Terms & Essential Knowledge**

## Role
You are a Corpus Enhancement Specialist.

## Task
Intelligently expand existing merged corpus based on targeted supplementation while preserving original structure

### Input Data
1. Current Merged Corpus:
<begin>
{merged_corpus}
<end>

4. Supplementation Requirements:
{{
    "terms_coverage": {{
        "included_terms_list": {included_terms_list},
        "missing_terms_list": {missing_terms_list}
    }},
    "question-answer_matching_rules": {{
        "matching_rules_derived_from_corpus_A":
            {matching_rules_derived_from_corpus_A},
        "matching_rules_derived_from_corpus_B":
            {matching_rules_derived_from_corpus_B}
    }},
    "question_feedback": {{
        "question_type": {question_type},
        "context_contain": {context_contain},
        "context_missing": {context_missing}
    }}
}}

### Processing Rules
1. **Content Preservation Principle**
   - Avoid regenerating the entire content of both the '### User' and
     '### Assistant' sections:
     - Instead, only regenerate the unanswered questions from the '### User'
       section (expanding necessary background details as needed) and update the
       corresponding answers in the '### Assistant' section based on the newly
       generated questions.
   - Ensure that the total modifications do not exceed 20%% of the original content.

2. **Term Integration Guidelines**
   - **Insertion of Missing Terms**: Insert each missing term from
     **missing_terms_list** using one of the following methods:
     a. Integrate the term naturally within an explanatory statement
        (e.g., "...which involves {{term}} mechanisms...").
     b. Incorporate the term into practical examples (e.g., "Application scenarios
        such as {{term}}...").

   - **Handling of Already Included Terms**: For every term listed in
     **included_terms_list** that is present in the current merged corpus, choose
     one of the following approaches:
     a. Retain the original content from the **Current Merged Corpus** if it is
        relevant to the term—meaning the content contains either an explicit
        mention or an implicit reference to the concept represented by the term.
     b. Rephrase the original content from the **Current Merged Corpus** that
        pertains to the term, ensuring that the revised version explicitly includes
        the term while also integrating all the missing terms.

   - **Prohibition**: Do not simply list terms without integrating them into the
                       context.

3. **Unanswered Question From '### User' Section Regeneration Guidelines**
   - Enhance the original unanswered question by incorporating additional background
     knowledge:
     - Specifically, based on the provided **context_missing**, the regenerated
       question must integrate both the existing context (**context_contain**) and
       the additional required context (**context_missing**).
   - Ensure that the regenerated unanswered question retains the same question
     type as specified by the provided **question_type**.
   - Fuse the matching rules or patterns from source corpus A
     (**matching_rules_derived_from_corpus_A**) and source corpus B
     (**matching_rules_derived_from_corpus_B**) into the regenerated unanswered
     question and its corresponding answer from the '### Assistant' section.

4. **Answer Regeneration Guidelines**
   - Regenerate only the responses in the '### Assistant' section.
   - Ensure that the regenerated answer explicitly addresses the unanswered
     questions from the '### User' section.
   - Maintain logical coherence and consistent terminology throughout the
     regenerated response.

5. **Coherence Assurance**
   - Ensure that all newly inserted or rephrased content is seamlessly integrated
     using explicit transitional phrases (e.g., "Considering", "In light of", "Particular
     attention should be paid to", etc.).
   - Preserve the original paragraph structure to maintain the logical flow and
     organization of the content.

### Expected Output Sturcture
{{
    "enhanced_corpus": "### User\n...\n### Assistant\n...",
}}

Figure 19: Candidate MCG Prompt Template for The Case of Omission of Key Technical Terms & Essential Knowledge

**Candidate MCG Prompt Template**

**Symbolic Loss:**
**Omission of Key Technical Terms &**
**Essential Knowledge &**
**Directed Answer**
- - - - - - - - - - - - - - - - - - - - - - - - - - - - - - - - - - - -

## Role
You are a *Corpus Enhancement Specialist*.

## Task
Intelligently expand existing merged corpus based on targeted supplementation
while preserving original structure

### Input Data
1. Current Merged Corpus:
<begin>
{merged_corpus}
<end>

4. Supplementation Requirements:
{{
    "terms_coverage": {{
        "included_terms_list": {included_terms_list},
        "missing_terms_list": {missing_terms_list}
    }},
    "question-answer_matching_rules": {{
        "matching_rules_derived_from_corpus_A":
            {matching_rules_derived_from_corpus_A},
        "matching_rules_derived_from_corpus_B":
            {matching_rules_derived_from_corpus_B}
    }},
    "question_feedback": {{
        "question_type": {question_type},
        "context_contain": {context_contain},
        "context_missing": {context_missing}
    }},
    "answer_quality_feedback": {assistant_feedback}
}}

### Processing Rules
1. **Content Preservation Principle**
   - Avoid regenerating the entire content of both the '### User' and
     '### Assistant' sections:
       - Instead, only regenerate the unanswered questions from the '### User'
         section (expanding necessary background details as needed) and update the
         corresponding answers in the '### Assistant' section based on the newly
         generated questions.
   - Ensure that the total modifications do not exceed 20%% of the original content.

2. **Term Integration Guidelines**
   - **Insertion of Missing Terms**: Insert each missing term from
     **missing_terms_list** using one of the following methods:
       a. Integrate the term naturally within an explanatory statement (e.g., "...which
          involves {{term}} mechanisms...").
       b. Incorporate the term into practical examples (e.g., "Application scenarios
          such as {{term}}...").

   - **Handling of Already Included Terms**: For every term listed in
     **included_terms_list** that is present in the current merged corpus, choose
     one of the following approaches:
       a. Retain the original content from the **Current Merged Corpus** if it is
          relevant to the term—meaning the content contains either an explicit
          mention or an implicit reference to the concept represented by the term.
       b. Rephrase the original content from the **Current Merged Corpus** that
          pertains to the term, ensuring that the revised version explicitly includes the
          term while also integrating all the missing terms.

   - **Prohibition**:
          Do not simply list terms without integrating them into the context.

3. **Unanswered Question From '### User' Section Regeneration Guidelines**
   - Enhance the original unanswered question by incorporating additional background
     knowledge:
       - Specifically, based on the provided **context_missing**, the regenerated
         question must integrate both the existing context (**context_contain**) and
         the additional required context (**context_missing**).
   - Ensure that the regenerated unanswered question retains the same question
     type as specified by the provided **question_type**.
   - Fuse the matching rules or patterns from source corpus A
     (**matching_rules_derived_from_corpus_A**) and source corpus B
     (**matching_rules_derived_from_corpus_B**) into the regenerated
     unanswered question and its corresponding answer from the '### Assistant'
     section.

4. **Answer Regeneration Guidelines**
   - Regenerate only the responses in the '### Assistant' section.
   - Ensure that the regenerated answer explicitly addresses the unanswered
     questions from the '### User' section.
   - Utilize the insights from "answer_quality_feedback" to inform the regeneration
     process, ensuring that the newly regenerated answers do not repeat the issues
     identified in the feedback.
   - Maintain logical coherence and consistent terminology throughout the
     regenerated response.

5. **Coherence Assurance**
   - Ensure that all newly inserted or rephrased content is seamlessly integrated
     using explicit transitional phrases (e.g., "Considering", "In light of",
     "Particular attention should be paid to", etc.).
   - Preserve the original paragraph structure to maintain the logical flow and
     organization of the content.

### Expected Output Sturcture
{{
    "enhanced_corpus": "### User\n...\n### Assistant\n...",
}}

Figure 20: Candidate MCG Prompt Template for The Case of Omission of Key Technical Terms & Essential Knowledge & Directed Answer

**Candidate MCG Prompt Template**

**Symbolic Loss:**

**Omission of Key Technical Terms & Question Feedback**

## Role
You are a strategy architect specializing in cross-corpus fusion, skilled in leveraging domain analysis to design effective merging strategies.

## Task
Utilize the provided **Domain Analysis** and **Selected Strategy** to generate one corpus fusion variant. This variant must employ the given **Selected Strategy** to merge corpus_A and corpus_B. The fusion for this variant should not only integrate the two corpora but also reflect the specific domain characteristics identified in the analysis.

## Input
{{
    "raw_corpus_A": {raw_corpus_A},
    "raw_corpus_B": {raw_corpus_B}
}}

## Domain Analysis
{{
    "corpus_A_domain": {corpus_A_domain},
    "corpus_B_domain": {corpus_B_domain},
    "corpus_A_key_terms_list": {corpus_A_key_terms_list},
    "corpus_B_key_terms_list": {corpus_B_key_terms_list},
    "matching_rules_derived_from_corpus_A":
            {matching_rules_derived_from_corpus_A},
    "matching_rules_derived_from_corpus_B":
            {matching_rules_derived_from_corpus_B},
    "relationship": {relationship}
}}

## Selected Strategy
{{
    "Domain-Specific Relationship Between Raw Corpus A and Raw Corpus B":
            {domain_type},
    "Applied Strategy":
            {applied_strategy},
    "Strategy Definition":
            {strategy_definition}
}}

## Output Requirements
- For the corpus fusion variant (enhanced_corpus) generation:
    1. For the '### User' section:
        a. The '### User' section must conclude with an unanswered question
            - This section should integrate **essential context** with **one or more related, logically connected questions**.
            - If there are multiple questions in the newly generated '### User' section, **do make sure to provide the corresponding direct answer to each question except the final one**
        b. In the '### User' section, ensure that the background information is logically structured and coherently presented. The question posed should be directly related to the provided background, with a natural and seamless transition between the background information and the question, resulting in an overall smooth and readable flow.
    2. '### Assistant' section must provide a **direct answer exclusively** to the unanswered question posed in the '### User' section
    3. Preserve all key terms from both **corpus_A_key_terms_list** and **corpus_B_key_terms_list** in this corpus fusion variant
    4. Adhere to the matching rules or patterns from both raw corpora (**matching_rules_A** and **matching_rules_B**), ensuring that the resulting matching rules or patterns explicitly encompass those from both original corpora
    5. Ensure logical coherence and semantic fluency throughout the content
    6. Utilize **bridging_concepts** and the **Selected Strategy** for the corpus fusion variants generation
    7. Strictly maintain:
        - The '### User' and '### Assistant' markers.
        - The formatting identical to that of the original corpora.
    8. Each corpus fusion variant **must include at least** one '### User' section and one '### Assistant' section
        - Which means the resultant corpus fusion variant may contain multiple '### User' and '### Assistant' section pairs if deemed indeed necessary.

## Example Output Structure
{{
    "enhanced_corpus": "### User
‹The content in the ### User section may be a single question (with or without context) or a series of question-answer pairs that culminate in a final question, in accordance with the original corpora.›

### Instruction
‹This section is optional. Sometimes the raw corpora include an Instruction section. For the newly generated corpus fusion variant, you may choose whether or not to include it.›

### Input
‹This section is optional. Sometimes the raw corpora include an Input section. For the newly generated corpus fusion variant, you may choose whether or not to include it.›

### Assistant
‹The content in the ### Assistant section must provide the answer to the latest question presented in the ### User section.›"

    or you can also include multiple '### User' and '### Assistant' section pairs derived from some raw corpus if you deem it indeed necessary

    "enhanced_corpus": "### User
‹The content in the ### User section may be a single question (with or without context) or a series of question-answer pairs that culminate in a final question, in accordance with the original corpora.›

### Assistant
‹The content in the ### Assistant section must provide the answer to the latest question presented in the ### User section.›

...

### User
‹The content in the ### User section may be a single question (with or without context) or a series of question-answer pairs that culminate in a final question, in accordance with the original corpora.›

### Assistant
‹The content in the ### Assistant section must provide the answer to the latest question presented in the ### User section.›"
}}

Figure 21: Candidate MCG Prompt Template for The Case of Omission of Key Technical Terms & Question Feedback

**Candidate MCG Prompt Template**

**Symbolic Loss:**
**Omission of Question Feedback**

## Role
You are a strategy architect specializing in cross-corpus fusion, skilled in leveraging domain analysis to design effective merging strategies.

## Task
Utilize the provided **Domain Analysis** and **Selected Strategy** to generate one corpus fusion variant. This variant must employ the given **Selected Strategy** to merge corpus_A and corpus_B. The fusion for this variant should not only integrate the two corpora but also reflect the specific domain characteristics identified in the analysis.

## Input
{{
    "raw_corpus_A": {raw_corpus_A},
    "raw_corpus_B": {raw_corpus_B}
}}

## Domain Analysis
{{
    "corpus_A_domain": {corpus_A_domain},
    "corpus_B_domain": {corpus_B_domain},
    "corpus_A_key_terms_list": {corpus_A_key_terms_list},
    "corpus_B_key_terms_list": {corpus_B_key_terms_list},
    "matching_rules_derived_from_corpus_A":
            {matching_rules_derived_from_corpus_A},
    "matching_rules_derived_from_corpus_B":
            {matching_rules_derived_from_corpus_B},
    "relationship": {relationship}
}}

## Selected Strategy
{{
    "Domain-Specific Relationship Between Raw Corpus A and Raw Corpus B":
            {domain_type},
    "Applied Strategy":
            {applied_strategy},
    "Strategy Definition":
            {strategy_definition}
}}

## Output Requirements
- For the corpus fusion variant (enhanced_corpus) generation:
    1. For the '### User' section:
        a. The '### User' section must conclude with an unanswered question
            - This section should integrate **essential context** with **one or more
              related, logically connected questions**.
            - If there are multiple questions in the newly generated '### User' section,
              **do make sure to provide the corresponding direct answer to each
              question except the final one**
        b. In the '### User' section, ensure that the background information is
              logically structured and coherently presented. The question posed should
              be directly related to the provided background, with a natural and
              seamless transition between the background information and the question,
              resulting in an overall smooth and readable flow.
    2. '### Assistant' section must provide a **direct answer exclusively** to the
       unanswered question posed in the '### User' section
    3. Preserve all key terms from both **corpus_A_key_terms_list** and
       **corpus_B_key_terms_list** in this corpus fusion variant
    4. Adhere to the matching rules or patterns from both raw corpora
       (**matching_rules_A** and **matching_rules_B**), ensuring that the resulting
       matching rules or patterns explicitly encompass those from both original corpora
    5. Ensure logical coherence and semantic fluency throughout the content
    6. Utilize **bridging_concepts** and the **Selected Strategy** for the corpus
       fusion variants generation
    7. Strictly maintain:
        - The '### User' and '### Assistant' markers.
        - The formatting identical to that of the original corpora.
    8. Each corpus fusion variant **must include at leaset** one '### User' section
       and one '### Assistant' section
        - Which means the resultant corpus fusion variant may contain multiple
          '### User' and '### Assistant' section pairs if deemed indeed necessary.

## Example Output Structure
{{
    "enhanced_corpus": "### User
‹The content in the ### User section may be a single question (with or without
context) or a series of question-answer pairs that culminate in a final question, in
accordance with the original corpora.›

### Instruction
‹This section is optional. Sometimes the raw corpora include an Instruction section.
For the newly generated corpus fusion variant, you may choose whether or not to
include it.›

### Input
‹This section is optional. Sometimes the raw corpora include an Input section. For
the newly generated corpus fusion variant, you may choose whether or not to include
it.›

### Assistant
‹The content in the ### Assistant section must provide the answer to the latest
question presented in the ### User section.›"

    or you can also include multiple '### User' and '### Assistant' section pairs
    derived from some raw corpus if you deem it indeed necessary

    "enhanced_corpus": "### User
‹The content in the ### User section may be a single question (with or without
context) or a series of question-answer pairs that culminate in a final question, in
accordance with the original corpora.›

### Assistant
‹The content in the ### Assistant section must provide the answer to the latest
question presented in the ### User section.›

...

### User
‹The content in the ### User section may be a single question (with or without
context) or a series of question-answer pairs that culminate in a final question, in
accordance with the original corpora.›

### Assistant
‹The content in the ### Assistant section must provide the answer to the latest
question presented in the ### User section.›"
}}

Figure 22: Candidate MCG Prompt Template for The Case of Omission of Question Feedback

**Candidate MCG Prompt Template**

**Symbolic Loss:**
**Omission of Essential Knowledge**

– – – – – – – – – – – – – – – – – – – – – – – – – – – –

```
## Role
You are a Corpus Enhancement Specialist.

## Task
Intelligently expand existing merged corpus based on targeted supplementation
while preserving original structure

### Input Data
1. Current Merged Corpus:
<begin>
{merged_corpus}
<end>

4. Supplementation Requirements:
{{
    "terms_coverage": {{
        "included_terms_list": {included_terms_list}
    }},
    "question-answer_matching_rules": {{
        "matching_rules_derived_from_corpus_A":
            {matching_rules_derived_from_corpus_A},
        "matching_rules_derived_from_corpus_B":
            {matching_rules_derived_from_corpus_B}
    }},
    "question_feedback": {{
        "question_type": {question_type},
        "context_contain": {context_contain},
        "context_missing": {context_missing}
    }}
}}

### Processing Rules
1. **Content Preservation Principle**
   - Avoid regenerating the entire content of both the '### User' and
     '### Assistant' sections:
       - Instead, only regenerate the unanswered questions from the '### User'
         section (expanding necessary background details as needed) and update the
         corresponding answers in the '### Assistant' section based on the newly
         generated questions.
   - Ensure that the total modifications do not exceed 20%% of the original content.

2. **Term Preservation Guidelines**
   - **Retention of Included Terms**: Although the current merged corpus already
     contains all the terms listed in **included_terms_list**, the expansion of the
     unanswered questions in the '### User' section must be conducted in a way
     that preserves these terms. For any content related to these terms, choose
     one of the following approaches:
       a. Retain the original content if it explicitly or implicitly references the term.
       b. Rephrase and expand the original content, ensuring that the final version
          explicitly includes the term while incorporating any additional necessary
          context.

   - **Prohibition**: Avoid merely listing the terms; they must be seamlessly
                        integrated within the expanded content.

3. **Unanswered Question From '### User' Section Regeneration Guidelines**
   - Enhance the original unanswered question by incorporating additional background
     knowledge:
       - Specifically, based on the provided **context_missing**, the regenerated
         question must integrate both the existing context (**context_contain**) and
         the additional required context (**context_missing**).
   - Ensure that the regenerated unanswered question retains the same question
     type as specified by the provided **question_type**.
   - Fuse the matching rules or patterns from source corpus A (
     **matching_rules_derived_from_corpus_A**) and source corpus B
     (**matching_rules_derived_from_corpus_B**) into the regenerated unanswered
     question and its corresponding answer from the '### Assistant' section.

4. **Answer Regeneration Guidelines**
   - Regenerate only the responses in the '### Assistant' section.
   - Ensure that the regenerated answer explicitly addresses the unanswered
     questions from the '### User' section.
   - Maintain logical coherence and consistent terminology throughout the
     regenerated response.

5. **Coherence Assurance**
   - Ensure that all newly inserted or rephrased content is seamlessly integrated
     using explicit transitional phrases (e.g., "Considering", "In light of",
     "Particular attention should be paid to", etc.).
   - Preserve the original paragraph structure to maintain the logical flow and
     organization of the content.

### Expected Output Sturcture
{{
    "enhanced_corpus": "### User\n...\n### Assistant\n...",
}}
```

Figure 23:   Candidate MCG Prompt Template for The Case of Omission of Essential Knowledge

**Candidate MCG Prompt Template**

**Symbolic Loss:**
**Omission of Essential Knowledge & Directed Answer**

- - - - - - - - - - - - - - - - - - - - - - - - - - - - - - - - -

```
## Role
You are a Corpus Enhancement Specialist.

## Task
Intelligently expand existing merged corpus based on targeted supplementation
while preserving original structure

### Input Data
1. Current Merged Corpus:
<begin>
{merged_corpus}
<end>

4. Supplementation Requirements:
{{
    "terms_coverage": {{
        "included_terms_list": {included_terms_list}
    }},
    "question-answer_matching_rules": {{
        "matching_rules_derived_from_corpus_A":
            {matching_rules_derived_from_corpus_A},
        "matching_rules_derived_from_corpus_B":
            {matching_rules_derived_from_corpus_B}
    }},
    "question_feedback": {{
        "question_type": {question_type},
        "context_contain": {context_contain},
        "context_missing": {context_missing}
    }},
    "answer_quality_feedback": {assistant_feedback}
}}

### Processing Rules
1. **Content Preservation Principle**
    - Avoid regenerating the entire content of both the '### User' and
      '### Assistant' sections:
        - Instead, only regenerate the unanswered questions from the '### User'
          section (expanding necessary background details as needed) and update the
          corresponding answers in the '### Assistant' section based on the newly
          generated questions.
    - Ensure that the total modifications do not exceed 20%% of the original content.

2. **Term Preservation Guidelines**
    - **Retention of Included Terms**: Although the current merged corpus already
      contains all the terms listed in **included_terms_list**, the expansion of the
      unanswered questions in the '### User' section must be conducted in a way
      that preserves these terms. For any content related to these terms, choose one
      of the following approaches:
        a. Retain the original content if it explicitly or implicitly references the term.
        b. Rephrase and expand the original content, ensuring that the final version
           explicitly includes the term while incorporating any additional necessary
           context.

    - **Prohibition**: Avoid merely listing the terms; they must be seamlessly
                       integrated within the expanded content.

3. **Unanswered Question From '### User' Section Regeneration Guidelines**
    - Enhance the original unanswered question by incorporating additional background
      knowledge:
        - Specifically, based on the provided **context_missing**, the regenerated
          question must integrate both the existing context (**context_contain**) and
          the additional required context (**context_missing**).
    - Ensure that the regenerated unanswered question retains the same question
      type as specified by the provided **question_type**.
    - Fuse the matching rules or patterns from source corpus A
      (**matching_rules_derived_from_corpus_A**) and source corpus B
      (**matching_rules_derived_from_corpus_B**) into the regenerated
      unanswered question and its corresponding answer from the
      '### Assistant' section.

4. **Answer Regeneration Guidelines**
    - Regenerate only the responses in the '### Assistant' section.
    - Ensure that the regenerated answer explicitly addresses the unanswered
      questions from the '### User' section.
    - Utilize the insights from "answer_quality_feedback" to inform the regeneration
      process, ensuring that the newly regenerated answers do not repeat the issues
      identified in the feedback.
    - Maintain logical coherence and consistent terminology throughout the
      regenerated response.

5. **Coherence Assurance**
    - Ensure that all newly inserted or rephrased content is seamlessly integrated
      using explicit transitional phrases (e.g., "Considering", "In light of", "Particular
      attention should be paid to", etc.).
    - Preserve the original paragraph structure to maintain the logical flow and
      organization of the content.

### Expected Output Sturcture
{{
    "enhanced_corpus": "### User\n...\n### Assistant\n...",
}}
```

Figure 24: Candidate MCG Prompt Template for The Case of Omission of Essential Knowledge & Directed Answer

## Candidate MCG Prompt Template

**Symbolic Loss:**
**Omission of Directed Answer**

- - - - - - - - - - - - - - - - - - - - - - - - - - - - - - - - -

## Role
You are a Corpus Enhancement Specialist.

## Task
Intelligently expand existing merged corpus based on targeted supplementation while preserving original structure

### Input Data
1. Current Merged Corpus:
<begin>
{merged_corpus}
<end>

2. Supplementation Requirements:
{{
    "answer_quality_feedback": {assistant_feedback}
}}

### Processing Rules
1. **Content Preservation Principle**
    - Avoid regenerating the entire content of both the '### User' and
      '### Assistant' sections.
    - Preserve the whole '### User' section.
    - Modify the content from the '### Assistant' section only.
    - Ensure that any modifications to the existing content do not exceed 20%% of
      the original content.

2. **Answer Regeneration Guidelines**
    - Regenerate only the responses in the '### Assistant' section.
    - Ensure that the regenerated answer explicitly addresses the unanswered
      questions from the '### User' section.
    - Utilize the insights from "answer_quality_feedback" to inform the regeneration
      process, ensuring that the newly regenerated answers do not repeat the issues
      identified in the feedback.
    - Maintain logical coherence and consistent terminology throughout the
      regenerated response.

3. **Coherence Assurance**
    - Ensure that all newly inserted or rephrased content is seamlessly integrated
      using explicit transitional phrases (e.g., "Considering", "In light of", "Particular
      attention should be paid to", etc.).
    - Preserve the original paragraph structure to maintain the logical flow and
      organization of the content.

### Expected Output Sturcture
{{
    "enhanced_corpus": "### User\n...\n### Assistant\n...",
}}

Figure 25: Candidate MCG Prompt Template for The Case of Omission of Directed Answer

## Symbolic Loss Example From ICD Operator

```
{
    "technical_report": {
        "term_coverage": {
            "included": {
                "corpus_A_key_terms_list": [
                    "emotional reaction",
                    "satisfaction",
                    "Head-Tail events",
                    "classification"
                ],
                "corpus_B_key_terms_list": [
                    "intention",
                    "desire",
                    "Head-Tail events",
                    "classification"
                ]
            },
            "missing": {
                "corpus_A_key_terms_list": [],
                "corpus_B_key_terms_list": []
            }
        },
        "question_analysis": {
            "question_exists": {
                "question_type": "open-ended",
                "needs_extra_knowledge_or_background_supplementation": {
                    "context_contain":
                        "The '### User' section provides a context
                        discussing emotional reactions and intentions in
                        social interactions, including examples of how these
                        concepts manifest.",
                    "context_missing":
                        "The '### User' section lacks specific examples or
                        scenarios that illustrate the interplay of emotional
                        reactions and intentions, which would help clarify
                        the question."
                }
            }
        },
        "answer_quality": {
            "if_needs_to_re_answer": {
                "explanation":
                    "The answer provided in the '### Assistant' section
                    does not directly address the last question from the
                    '### User' section. The question asks how emotional
                    reactions and intentions interplay in social interactions,
                    while the assistant's response focuses more on the
                    general relationship between emotional reactions and
                    intentions without specifically addressing the interplay
                    aspect."
            }
        }
    }
}
```

Figure 26: Symbolic Loss Example From ICD Operator

## Symbolic Loss Example From FAC Operator

```
{
    "if_containing_direct_answer": {
        "direct_answer":
            "We can identify processes such as recognition of motivations,
            emotional triggers, and the development of interpersonal
            relationships that shape social interactions and their emotional
            outcomes."
    }
    "if_containing_information_needed_to_delete": {
        "information_needs_to_remove":
            "In particular, the interplay of empathy and gratitude can
            significantly enhance the understanding of social dynamics.
            Furthermore, the psychological principle of generosity often
            serves as a catalyst for positive emotional reactions, fostering
            deeper connections among individuals. Considering the
            interconnectedness of these elements, exploring the interplay
            of empathy and gratitude provides valuable insights into how
            emotional responses are influenced by social contexts. By
            examining these aspects, we can gain insights into the
            interconnectedness of emotional responses and social
            contexts."
    }
}
```

Figure 27: Symbolic Loss Example From FAC Operator

