# OpenReview forum: "ENTP: Enhancing Low-Quality SFT Data via Neural-Symbolic Text Purge-Mix"
_ICLR.cc/2026/Conference — Submitted to ICLR 2026_

### Official Review · Reviewer_dHmZ · 2025-10-26

**Soundness:** 2
**Presentation:** 2
**Contribution:** 2
**Rating:** 4
**Confidence:** 5

**Summary:**

This paper challenges the prevailing "quality-first" paradigm in Supervised Fine-Tuning (SFT), where models are trained on small, curated high-quality (HQ) datasets while large amounts of low-quality (LQ) data are discarded. The authors argue this approach is hitting a bottleneck due to the scarcity of untapped HQ data and that LQ data contains valuable, recoverable signals. They introduce ENTP, a novel framework designed to enhance and leverage LQ data through a "Neural-Symbolic Text Purge-Mix" approach.

**Strengths:**

The paper addresses a critical bottleneck in SFT. The shift from merely selecting high-quality data to actively enhancing and synthesizing low-quality data is a compelling and important contribution, offering a path to utilize data that is typically discarded.

The ability of the ENTP-generated dataset (54K samples from the LQ pool) to consistently outperform all data selection baselines applied to the LQ pool, and frequently surpass the performance of the HQ set (131K samples) and even the Full set (300K samples), strongly validates the core hypothesis.

**Weaknesses:**

The ENTP framework is exceedingly complex, involving numerous stages: LLM rating, score correction, embedding, multi-stage clustering, MMR selection, and the iterative fusion. Critically, Step 3 requires multiple iterative calls (Cycles 1 and 2) to an LLM using five different operators (DA, MCG, ICD, FAC, FAU) for every pair of fused samples. Generating the 54K dataset likely required an enormous number of LLM inferences (using gpt-4o-mini). The paper completely omits any analysis of the computational cost, time, or financial expense of the data generation process. Without this analysis, the practicality and scalability of the approach are highly questionable, as the generation cost may outweigh the SFT efficiency gains.

The entire pipeline heavily relies on a powerful proprietary model (gpt-4o-mini) for both initial rating and the complex synthesis/evaluation tasks in Step 3. This raises significant concerns that the observed gains are primarily due to knowledge distillation from the oracle model, rather than the specific effectiveness of the neural-symbolic fusion architecture. ENTP appears to be an elaborate method for prompting the oracle until it produces high-quality output based on the LQ inputs. The paper does not adequately distinguish the benefits of the framework from simpler distillation or data augmentation techniques using the same LLM.

The paper positions ENTP against data selection methods. However, ENTP is fundamentally a data synthesis method. The evaluation is incomplete as it does not compare against other established data synthesis techniques (e.g., Self-Instruct, Evol-Instruct, or simpler mixing/rewriting strategies).

**Questions:**

In addition to the weakness section, I have the below questions:

How do you disentangle the effects of knowledge distillation from the benefits of the complex neural-symbolic framework? Have you compared ENTP against a simpler baseline, such as using the same LLM to rewrite or improve individual low-quality samples without the fusion and iterative optimization steps?

Can you provide a concrete example in the main text illustrating the iteration process? Please show the generated corpus, the resulting Symbolic Loss (JSON), the specific prompt update made by the SPO, and the improved corpus in the subsequent iteration.

---

> ### Author Response · Authors · 2025-11-26
> **Response 1/6**
>
> We want to thank the reviewer for their positive feedback and comments. We will address individual comments below.
>
> # W1: *... The paper completely omits any analysis of the computational cost ...*
>
> Thank you very much for the reminder. Conducting a cost analysis is indeed an important component of any project, we are happy to provide approximate GPU runtime and API costs of several baselines for reference.
> | | LESS[1] | MathFusion[2] | Evol-Instruct[3] | ENTP |
> |:----:|:----:|:----:|:----:|:----:|
> |Average API Cost Per Resultant Corpus (in USD)|0|0.004|0.003|0.005|
> |GPU Runtime (in GPU-hours)|152.5|17|17|17|
>
> [1] LESS: Selecting Influential Data for Targeted Instruction Tuning
> [2] Mathfusion: Enhancing mathematic problem-solving of llm through instruction fusion
> [3] WizardLM: Empowering large pre-trained language models to follow complex instructions
>
> # W2 & Q1-1: *The paper does not adequately distinguish the benefits of the framework from simpler distillation or data augmentation techniques using the same LLM. ... Have you compared ENTP against a simpler baseline, such as using the same LLM to rewrite or improve individual low-quality samples without the fusion and iterative optimization steps*
>
> We fully understand your concern. To provide a clearer view of ENTP’s performance, we use gpt-4o-mini-2024-07-18 to perform direct rewriting on each low-quality corpus and to directly fuse randomly paired corpora, bypassing both our clustering step (Step 2) and neural-symbolic fusion step (Step 3). We then randomly select 54888 resultant corpora to SFT Mistral-7B-v0.3. Additional ablation studies, along with the results of the rewriting baseline, are reported below:
> | | mmlu | truthfulqa | gsm | bbh | tydiqa | Average |
> |:----:|:----:|:----:|:----:|:----:|:----:|:----:|
> |Rewriting|47.3|42.4|41.5|49.4|49.9|46.1|
> |direct_corpora_fusion_wo_step2&3|40.4|41.4|37.5|48.8|50.7|43.8|
> |direct_corpora_fusion_wo_step3|45.9|42.8|40.0|50.0|50.3|45.8|
> |ENTP|58.6|43.0|44.0|53.8|58.3|51.5|
>
> # W3: *... The evaluation is incomplete as it does not compare against other established data synthesis techniques (e.g., Self-Instruct, Evol-Instruct, or simpler mixing/rewriting strategies) ...*
>
> Thank you for pointing this out, and we are happy to provide more baselines for comparison. All executable baseline results will be included in our revised paper.
>
> Experimental Setup: For a fair comparison, all baselines involving LLM calls use GPT-4o-mini-2024-0718. Each baseline generates a resultant dataset containing only 54888 corpora. Additionally, the base LLM used for SFT is Mistral-7B-v0.3.
> - Rewriting: We use LLM to directly rewrite each low-quality corpus.
> - direct_corpora_fusion_wo_step3: We use LLM to directly fuse two corpora via random pairing from the low-quality corpora.
> - MathFusion[1]: This baseline is requested by the reviewer PTLK.
> - Evol-Instruct[2]: Requested baseline.
> - Self-Instruct[3]: Requested baseline.
>
>
> | | mmlu | truthfulqa | gsm | bbh | tydiqa | Average |
> |:----:|:----:|:----:|:----:|:----:|:----:|:----:|
> |Rewriting|47.3|42.4|41.5|49.4|49.9|46.1|
> |direct_corpora_fusion_wo_step3|45.9|42.8|40.0|50.0|50.3|45.8|
> |MathFusion[1]|50.8|59.6|44.5|52.8|41.4|49.8|
> |Evol-Instruct[2]|54.0|57.5|33.5|53.1|42.8|48.2|
> |Self-Instruct[3]|53.1|43.8|45.0|55.2|50.9|49.6|
> |ENTP|58.6|43.0|44.0|53.8|58.3|51.5|
>
> [1] Mathfusion: Enhancing mathematic problem-solving of llm through instruction fusion
> [2] WizardLM: Empowering large pre-trained language models to follow complex instructions
> [3] Self-Instruct: Aligning Language Model with Self Generated Instructions
>
> # Q1-0: *How do you disentangle the effects of knowledge distillation from the benefits of the complex neural-symbolic framework?*
>
> Thank you very much for the question. It is indeed very thoughtful.
>
> Our neural-symbolic fusion step follows a strictly staged architecture in which the output of each step is a required input to the subsequent stage. The knowledge distillation step is not merely a performance enhancer but a **structural prerequisite**: without this distilled knowledge, the next stage receives no input and thus cannot proceed. As a result, this entired fusion step cannot operate if the knowledge distaillation step is removed, and a conventional ablation study that excludes the distillation step is not feasible in our setting.
>
> However, this does not imply that the capability of Step 3 relies entirely on knowledge distillation. Without the predefined symbolic rules (**the prompt template of the Domain Analysis operator shown in Figure 10**), it is completely unclear what knowledge the distillation step should extract. Moreover, without the prior knowledge we provide regarding how to fuse corpora (**the symbolic fusion strategy illustrated in Figure 11**), Step 3 would have no guidance on which knowledge to retain, merge, or discard, and would therefore be incapable of performing the corpora fusion task.

---

> ### Author Response · Authors · 2025-11-26
> **Response 2/6**
>
> # Q2: *Can you provide a concrete example ... ?*
>
> No problem. We can provide a concrete example to illustrate ENTP’s Step 3 operational mechanism step by step. This example will also be included in our revised paper.
>
> ## Step 3: Neural-Symbolic Two-to-One Corpus Fusion
> Randomly select two corpora from the representative low-quality set as input:
> ```
> Raw Corpus A:
> ### User
> input question: Write a sentence not in English.
> output answer: от того, что появляется белая лилия, которая цветет с июня по сентябрь, который распространяется на соленый воздух нежный аромат.
> input question: Write a sentence not in English.
> output answer: При других заболеваниях лечение может идти до 1 года и далее. Биокоррекция Васильева может осуществляться как в клинических условиях, так и амбулаторно.
> input question: Write a sentence not in English.
> output answer: Насколько существенно ее воздействие на важные бизнес-процессы?
> input question: Write a sentence not in English.
> output answer:
> ### Assistant
> Ваш входной идентификационный код будет также Вашим кодовым именем при участии в конкурсе. Информация про введение данных будет отослана Вам по электронной почте FTP с адресом и паролем.
>
> Raw Corpus B:
> ### User
> input question: Write a sentence not in English.
> output answer: Отель Fayal Resort Hotel будет показан на карте, если Вы включите JavaScript.
>
>
> Q: Translate "How to eat: Just drink it." to Russian?
> Yes: Как употреблять: просто выпейте!
>
>
> [Q]: За 72 года своего существования "Башнефть" сформировалась как мощный многопрофильный нефтегазодобывающий комплекс.
>
> Translate this to English?
> [A]: Within 72 years of its existence, Bashneft has become powerful multi-structural oil and gas extracting enterprise.
>
>
> Question:
> Для Профессионального комплекта существуют корпоративные решения с возможностью определять состав дополнительных специализированных словарей.
>
> Could you please translate this to English?
> Answer:
> For the Professional package there exist enterprise solutions with the possibility to determine the composition of special dictionaries.
>
>
> test: Балахтинский район
> English?
>
> translation: Balakhtinsky District
>
>
> input question: Write a sentence not in English.
> output answer:
> ### Assistant
> Лангоун, Майкл
> ```
> > Step 1: Domain Analysis (Prompt template has been shown in Figure 10.)
> ```
> LLM Response:
> {
>     "corpus_A_domain": "Language Translation",
>     "corpus_B_domain": "Language Translation",
>     "corpus_A_key_terms_list": [
>         "sentence construction",
>         "foreign language",
>         "biocorrection",
>         "business processes"
>     ],
>     "corpus_B_key_terms_list": [
>         "translation",
>         "enterprise solutions",
>         "specialized dictionaries",
>         "multilingual"
>     ],
>     "matching_rules_derived_from_corpus_A": "Questions from the '### User' section often ask for non-English sentences or contexts, and answers in the '### Assistant' section provide translations or related foreign phrases.",
>     "matching_rules_derived_from_corpus_B": "The '### User' questions request either translations or sentences in non-English, while the '### Assistant' responses deliver translations or contextual explanations in both English and Russian.",
>     "relationship": "related-domain"
> }
> ```

---

> ### Author Response · Authors · 2025-11-26
> **Response 3/6**
>
> Once the relationship is identified, the corresponding fusion strategy is selected (all strategies are provided in Figure 11). The MCG template (Figure 13) is then applied to generate the initial merged corpus.
> > Step 2 & 3: Strategy Selection & Merged Corpus Generation
> ```
> LLM Response:
> {
>     "overall_response": [
>         {
>             "domain_type": "related-domain",
>             "applied_strategy": "conceptual_analogy",
>             "corpus_fusion_variant": "### User\nIn the realm of Language Translation, both corpora emphasize the importance of constructing sentences in foreign languages. For instance, corpus A provides examples of non-English sentences, while corpus B focuses on translating these sentences into English. This highlights the interconnectedness of language processing and multilingual communication. How can we further enhance the understanding of cultural linguistics in translation?\n\n### Assistant\nWe can enhance the understanding of cultural linguistics in translation by incorporating cultural context into the translation process, ensuring that idiomatic expressions and cultural references are accurately conveyed."
>         },
>         {
>             "domain_type": "related-domain",
>             "applied_strategy": "process_mapping",
>             "corpus_fusion_variant": "### User\nWhen considering enterprise solutions for Language Translation, corpus B discusses specialized dictionaries that can aid in translation tasks. Meanwhile, corpus A mentions biocorrection and its relevance to business processes. This suggests a need for a structured approach to integrating these specialized tools into translation workflows. What are the best practices for implementing such solutions in a multilingual environment?\n\n### Assistant\nBest practices for implementing specialized tools in a multilingual environment include conducting thorough needs assessments, training staff on the use of these tools, and continuously evaluating the effectiveness of the solutions in improving translation accuracy."
>         },
>         {
>             "domain_type": "related-domain",
>             "applied_strategy": "term_bridging",
>             "corpus_fusion_variant": "### User\nBoth corpora highlight key terms such as 'sentence construction' from corpus A and 'translation' from corpus B. This overlap indicates a shared focus on the mechanics of language. Additionally, terms like 'foreign language' and 'multilingual' suggest a broader context of language processing. How can we leverage these shared terms to create a more cohesive language translation framework?\n\n### Assistant\nWe can leverage these shared terms by developing a unified glossary that defines each term in the context of both corpora, facilitating better communication and understanding among language professionals."
>         }
>     ]
> }
> ```
> Since we provide three fusion strategies for each relationship, three different merged corpora will be generated. However, for clarity of presentation, we illustrate the process using only the initial merged corpus produced by the *'term_bridge'* strategy. The following step is to obtain the **Symbolic Loss** for this merged corpus.
> > Step 4: Information Completeness Detection (prompt template in Figure 14)
> ```
> LLM Response:
> {
>     "technical_report": {
>         "term_coverage": {
>             "included": {
>                 "corpus_A_key_terms_list": [
>                     "sentence construction",
>                     "foreign language",
>                     "biocorrection",
>                     "business processes"
>                 ],
>                 "corpus_B_key_terms_list": [
>                     "translation",
>                     "enterprise solutions",
>                     "specialized dictionaries",
>                     "multilingual"
>                 ]
>             },
>             "missing": {
>                 "corpus_A_key_terms_list": [],
>                 "corpus_B_key_terms_list": []
>             }
>         },
>         "question_analysis": {
>             "question_exists": {
>                 "question_type": "open-ended",
>                 "needs_extra_knowledge_or_background_supplementation": {
>                     "context_contain": "The '### User' section discusses the overlap of key terms and asks how to leverage them for a cohesive framework.",
>                     "context_missing": "The '### User' section lacks specific examples or details on what aspects of the shared terms should be leveraged, which could help in formulating a more precise answer."
>                 }
>             }
>         },
>         "answer_quality": {
>             "if_needs_to_re_answer": {
>                 "explanation": "The answer provided in the '### Assistant' section does not directly address the last question from the '### User' section, which asks how to leverage shared terms to create a cohesive language translation framework. The response is more about developing a glossary rather than providing a cohesive framework."
>             }
>         }
>     }
> }
> ```

---

> ### Author Response · Authors · 2025-11-26
> **Response 4/6**
>
> According to the resultant symbolic loss, the current merged corpus has **missing necessary context in the '### User' section**, and the **'### Assistant' section does not provide a direct and clear answer** to the question from '### User' section. Thus, the initial prompt template for Merged Corpus Generation (Figure 13) is updated to the version shown in Figure 24 to generate the enhanced merged corpus.
> > Step 5: Enhanced Corpus Generation (Prompt template has been shown in Figure 24.)
> ```
> LLM Response:
> {
>     "enhanced_corpus": "### User\nBoth corpora highlight key terms such as 'sentence construction' from corpus A and 'translation' from corpus B. This overlap indicates a shared focus on the mechanics of language. Additionally, terms like 'foreign language' and 'multilingual' suggest a broader context of language processing. How can we leverage these shared terms, particularly 'sentence construction' and 'translation', to create a more cohesive language translation framework? For instance, what specific strategies can we implement that utilize 'business processes' and 'specialized dictionaries' to enhance our approach to multilingual translation?\n\n### Assistant\nWe can leverage these shared terms by developing a unified glossary that defines each term in the context of both corpora, facilitating better communication and understanding among language professionals. In addition, we should consider implementing strategies that integrate 'business processes' with 'sentence construction' techniques to streamline translation workflows. Utilizing 'specialized dictionaries' can also enhance accuracy in translations, particularly in a multilingual context. By aligning these elements, we can create a cohesive language translation framework that not only addresses the mechanics of language but also improves the overall efficiency and effectiveness of translation practices."
> }
> ```
> Again, the following step is to obtain the **Symbolic Loss** for the resultant enhanced merged corpus from the last step.
> > Step 4: Information Completeness Detection (ICD) For Enhanced Corpus (Prompt template has been shown in Figure 14.)
> ```
> LLM Response:
> {
>     "technical_report": {
>         "term_coverage": {
>             "included": {
>                 "corpus_A_key_terms_list": [
>                     "biocorrection",
>                     "business processes",
>                     "foreign language"
>                 ],
>                 "corpus_B_key_terms_list": [
>                     "translation",
>                     "enterprise solutions",
>                     "specialized dictionaries",
>                     "multilingual"
>                 ]
>             },
>             "missing": {
>                 "corpus_A_key_terms_list": [
>                     "sentence construction"
>                 ],
>                 "corpus_B_key_terms_list": []
>             }
>         },
>         "question_analysis": {
>             "question_exists": {
>                 "question_type": "open-ended",
>                 "needs_extra_knowledge_or_background_supplementation": "nope"
>             }
>         },
>         "answer_quality": {
>             "if_needs_to_re_answer": "nope"
>         }
>     }
> }
> ```

---

> ### Author Response · Authors · 2025-11-26
> **Response 5/6**
>
> Compared to the symbolic loss of the initial merged corpus, the current symbolic loss only reflects missing key terms from source corpus A. This indicates that the enhanced merged corpus contains clear questions and answers and provides sufficient background knowledge to address the questions. The **only remaining issue is that the coverage of key terms is slightly insufficient**. Nonetheless, compared to the initial merged corpus, the current version represents a significant improvement. Therefore, to address the issue of missing key terms, we update the current template to the version shown in Figure 17 to generate a better merged corpus covering a broader range of domains.
> > Step 5: Enhanced Corpus Generation (Prompt template has been shown in Figure 17.)
> ```
> LLM Response:
> {
>     "enhanced_corpus": "### User\nBoth corpora highlight key terms such as 'sentence construction' from corpus A and 'translation' from corpus B. This overlap indicates a shared focus on the mechanics of language. Additionally, terms like 'foreign language' and 'multilingual' suggest a broader context of language processing. How can we leverage these shared terms, particularly 'sentence construction' and 'translation', to create a more cohesive language translation framework? For instance, what specific strategies can we implement that utilize 'business processes' and 'specialized dictionaries' to enhance our approach to multilingual translation?\n\n### Assistant\nWe can leverage these shared terms by developing a unified glossary that defines each term in the context of both corpora, facilitating better communication and understanding among language professionals. In particular, the concept of 'sentence construction' is crucial as it involves the mechanisms of structuring sentences effectively in various languages. Additionally, we should consider implementing strategies that integrate 'business processes' with 'sentence construction' techniques to streamline translation workflows. For example, application scenarios such as utilizing 'specialized dictionaries' can enhance accuracy in translations, particularly in a multilingual context. By aligning these elements, we can create a cohesive language translation framework that not only addresses the mechanics of language but also improves the overall efficiency and effectiveness of translation practices."
> }
> ```
> Again, the following step is to obtain the **Symbolic Loss** for the resultant enhanced merged corpus from the last step.
> > Step 4: Information Completeness Detection (ICD) For Enhanced Corpus
> ```
> LLM Response:
> {
>     "technical_report": {
>         "term_coverage": {
>             "included": {
>                 "corpus_A_key_terms_list": [
>                     "biocorrection",
>                     "business processes",
>                     "foreign language",
>                     "sentence construction"
>                 ],
>                 "corpus_B_key_terms_list": [
>                     "enterprise solutions",
>                     "multilingual",
>                     "specialized dictionaries",
>                     "translation"
>                 ]
>             },
>             "missing": {
>                 "corpus_A_key_terms_list": [],
>                 "corpus_B_key_terms_list": []
>             }
>         },
>         "question_analysis": {
>             "question_exists": {
>                 "question_type": "open-ended",
>                 "needs_extra_knowledge_or_background_supplementation": "nope"
>             }
>         },
>         "answer_quality": {
>             "if_needs_to_re_answer": "nope"
>         }
>     }
> }
> ```
> According to the current symbolic loss, **all conditions have been satisfied: no key terms are missing**, the '### User' section provides a clear question along with the necessary background knowledge to answer it, and the '### Assistant' section provides a precise answer to the question. Thus, **the resultant merged corpus successfully completes the entire Cycle 1 process**. The next step is to double-check the quality of the answer in the '### Assistant' section of the current resultant merged corpus (obtaining the **symbolic loss** from the FAC operator).
> > Step 6: Final Answer Check (prompt template is shown in Figure 12)
> ```
> LLM Response:
> {
>     "if_containing_direct_answer": {
>         "direct_answer": "We can leverage these shared terms by developing a unified glossary that defines each term in the context of both corpora, facilitating better communication and understanding among language professionals. Additionally, we should consider implementing strategies that integrate 'business processes' with 'sentence construction' techniques to streamline translation workflows. For example, utilizing 'specialized dictionaries' can enhance accuracy in translations, particularly in a multilingual context."
>     },
>     "if_containing_information_needed_to_delete": "nope"
> }
> ```

---

> ### Author Response · Authors · 2025-11-26
> **Response 6/6**
>
> According to the symbolic loss from the FAC operator, the current '### Assistant' section contains a direct answer to the question from '### User' and includes no irrelevant or redundant information, indicating that **the current merged corpus has successfully passed the entire Cycle 2 process**.
>
> ## Final Merged Corpus
> We present the input corpora pair and their resultant merged corpus (**'enhanced_corpus'**) below:
> ```
> {
>     "raw_corpus_A": "### User\ninput question: Write a sentence not in English.\noutput answer: от того, что появляется белая лилия, которая цветет с июня по сентябрь, который распространяется на соленый воздух нежный аромат.\ninput question: Write a sentence not in English.\noutput answer: При других заболеваниях лечение может идти до 1 года и далее. Биокоррекция Васильева может осуществляться как в клинических условиях, так и амбулаторно.\ninput question: Write a sentence not in English.\noutput answer: Насколько существенно ее воздействие на важные бизнес-процессы?\ninput question: Write a sentence not in English.\noutput answer:\n### Assistant\nВаш входной идентификационный код будет также Вашим кодовым именем при участии в конкурсе. Информация про введение данных будет отослана Вам по электронной почте FTP с адресом и паролем.",
>     "raw_corpus_B": "### User\ninput question: Write a sentence not in English.\noutput answer: Отель Fayal Resort Hotel будет показан на карте, если Вы включите JavaScript.\n\n\nQ: Translate \"How to eat: Just drink it.\" to Russian?\nYes: Как употреблять: просто выпейте!\n\n\n[Q]: За 72 года своего существования \"Башнефть\" сформировалась как мощный многопрофильный нефтегазодобывающий комплекс.\n\nTranslate this to English?\n[A]: Within 72 years of its existence, Bashneft has become powerful multi-structural oil and gas extracting enterprise.\n\n\nQuestion:\nДля Профессионального комплекта существуют корпоративные решения с возможностью определять состав дополнительных специализированных словарей.\n\nCould you please translate this to English?\nAnswer:\nFor the Professional package there exist enterprise solutions with the possibility to determine the composition of special dictionaries.\n\n\ntest: Балахтинский район\nEnglish?\n\ntranslation: Balakhtinsky District\n\n\ninput question: Write a sentence not in English.\noutput answer:\n### Assistant\nЛангоун, Майкл",
>     "enhanced_corpus": "### User\nBoth corpora highlight key terms such as 'sentence construction' from corpus A and 'translation' from corpus B. This overlap indicates a shared focus on the mechanics of language. Additionally, terms like 'foreign language' and 'multilingual' suggest a broader context of language processing. How can we leverage these shared terms, particularly 'sentence construction' and 'translation', to create a more cohesive language translation framework? For instance, what specific strategies can we implement that utilize 'business processes' and 'specialized dictionaries' to enhance our approach to multilingual translation?\n\n### Assistant\nWe can leverage these shared terms by developing a unified glossary that defines each term in the context of both corpora, facilitating better communication and understanding among language professionals. In particular, the concept of 'sentence construction' is crucial as it involves the mechanisms of structuring sentences effectively in various languages. Additionally, we should consider implementing strategies that integrate 'business processes' with 'sentence construction' techniques to streamline translation workflows. For example, application scenarios such as utilizing 'specialized dictionaries' can enhance accuracy in translations, particularly in a multilingual context. By aligning these elements, we can create a cohesive language translation framework that not only addresses the mechanics of language but also improves the overall efficiency and effectiveness of translation practices."
> }
> ```

---

### Official Review · Reviewer_CanP · 2025-10-30

**Soundness:** 3
**Presentation:** 3
**Contribution:** 3
**Rating:** 6
**Confidence:** 4

**Summary:**

The work proposes ENTP, a data-construction pipeline that reclaims "low-quality" instruction data instead of discarding it. After a) LLM-based quality scoring with a Score Transition Matrix; b) Clustering + MMR selection, ENTP performs a neural-symbolic "two-to-one" fusion of pairs of low-quality samples using iterative LLM generation and symbolic feedback. The authors craft an ENTP dataset (default 54,888 pairs) and show that models fine-tuned on ENTP outperform LQ/HQ subsets and sometimes or surpass the Full Set (≈300k) on OpenLLM benchmarks.

**Strengths:**

1. The problem of evaluating and training LLMs without enough high-quality supervision is pressing, revalorizing low-quality data is a fresh, pragmatic angle that complements selection-only pipelines.
2. Experiments span multiple model families, datasets, and baselines, providing convincing evidence of generality. The consistent performance gain even over full-dataset fine-tuning is noteworthy.
3. The work highlights a structural limitation in classical data selection and empirically verifies that low-quality data retain useful signals.

**Weaknesses:**

1. The "symbolic loss" component remains conceptually abstract. The loss is an LLM-authored checklist driving prompt edits, not a defined objective with measurable descent, no ablation isolates its incremental contribution.
2. The method involves multiple nested steps (scoring correction, clustering, two-stage fusion), which may pose high computational cost and reproducibility challenges. Runtime (or API token usage) analysis is missing.
3. The paper does not provide a systematic evaluation of the actual quality of the generated instruction–response pairs. There is no human annotation, automatic text-quality scoring, or factuality/toxicity audit of the ENTP corpus. The only evidence of improvement comes indirectly from downstream benchmark gains and a few anecdotal examples in the appendix.It is unclear whether ENTP truly produces higher-quality data or merely data that better fits the evaluation distributions.

**Questions:**

1. The reported sizes (LQ = 123,786, HQ = 131,247, Full = 300,932) do not sum correctly. Is there overlap, filtering, or an unlabeled "medium-quality" portion?

---

> ### Author Response · Authors · 2025-11-26
> **Response 1/2**
>
> We want to thank the reviewer for their positive feedback and comments. We will address individual comments below.
>
> # W1-0: *The "symbolic loss" component remains conceptually abstract.*
>
> First, we sincerely apologize for any confusion caused by the technical terminology (e.g., "symbolic loss") in the paper. Concretely, for every intermediate merged corpus the ICD (Information-Completeness Detection) operator generates a symbolic loss, essentially, a structured JSON object that functions as an information-bearing carrier. Taking one of its keys, "questions_exists" as an example, it enumerates (**all detailes had been shown in Figure 14 in Appendix**):
> 1, whether the "### User" session contains an explicit question;
> 2, if so, whether the question is open- or closed-ended;
> 3, whether sufficient background knowledge has been supplied to answer it;
> 4, what background knowledge is currently provided and what remains missing; and
> 5, the exact supplementary knowledge required.
>
> Consequently, this JSON object simultaneously **assesses the present merged corpus** and **prescribes the precise direction for revision**, an explicit **natural-language analogue of a loss value and its gradient**. We term it symbolic loss because every key is an information-laden symbol, and, in our ENTP, every rule is likewise encoded as a symbol.
>
> # W1-1: *... no ablation isolates its incremental contribution.*
>
> Thank you very much for your suggestion. The most important aspect of our Step 3 is the refinement of the merged corpora based on symbolic loss. Therefore, instead of performing an ablation specifically on the symbolic loss, we can provide ablation studies for each step. We hope this will offer a more intuitive understanding of the contribution of each component.
>
> To isolate the individual contribution of each ENTP component, we ablate both the clustering step (Step 2) and the neural–symbolic fusion step (Step 3). Concretely, we employ GPT-4o-mini-2024-07-18 to directly fuse randomly paired corpora, thereby bypassing clustering and neural-symbolic fusion steps. From the resulting pool we randomly subsample 54888 merged corpora to SFT Mistral-7B-v0.3. Downstream evaluation results are reported below:
> | | mmlu | truthfulqa | gsm | bbh | tydiqa | Average |
> |:----:|:----:|:----:|:----:|:----:|:----:|:----:|
> |direct_corpora_fusion_wo_step2&3|40.4|41.4|37.5|48.8|50.7|43.8|
> |direct_corpora_fusion_wo_step3|45.9|42.8|40.0|50.0|50.3|45.8|
> |ENTP|58.6|43.0|44.0|53.8|58.3|51.5|
>
> # W2: *Runtime (or API token usage) analysis is missing.*
>
> Thank you very much for the reminder. Conducting a cost analysis is indeed an important component of any project, we are happy to provide approximate GPU runtime and API costs of several baselines for reference. In addition, we report the average API cost per resultant corpus.
> | | LESS[1] | MathFusion[2] | Evol-Instruct[3] | ENTP |
> |:----:|:----:|:----:|:----:|:----:|
> |Average API Cost (in USD)|0|0.004|0.003|0.005|
> |GPU Runtime (in GPU-hours)|152.5|17|17|17|
>
> [1] LESS: Selecting Influential Data for Targeted Instruction Tuning
> [2] Mathfusion: Enhancing mathematic problem-solving of llm through instruction fusion
> [3] WizardLM: Empowering large pre-trained language models to follow complex instructions
>
> # W3: *... There is no ... automatic text-quality scoring, or factuality/toxicity audit of the ENTP corpus. ...*
>
> Thank you very much for pointing this out. We fully agree that visualizing the quality differences before and after corpora fusion is important for demonstrating the effectiveness of our approach. Since our Step 1 relies on a **curated LLM-rating score of overall quality (the higher the better)** for selecting low-quality corpora (the prompt template is provided in Figure 6 of the Appendix), we use the same metric to evaluate the resultant merged corpora. The curated overall quality-score distributions for the low-quality corpora set (obtained in Step 1) and for our ENTP-generated merged corpora (obtained in Step 3) are provided below:
> | | 0 | 1 | 2 | 3 | 4 | 5 | Average |
> |:----:|:----:|:----:|:----:|:----:|:----:|:----:|:----:|
> |Low-Quality Corpora|16962|48229|101895|0|0|0|1.51|
> |ENTP|20|81|3109|41446|10226|0|3.13|
>
> For each corpora set, we report the **frequency** of each score level (from 0 to 5) and compute the corresponding average. As shown in the table above, the average score of the merged corpora obtained after applying our Step 2 and Step 3 is two times larger than the average score of the corpora without applying these steps.

---

> ### Author Response · Authors · 2025-11-26
> **Response 2/2**
>
> # Q1: *1. The reported sizes (LQ = 123,786, HQ = 131,247, Full = 300,932) do not sum correctly. Is there overlap, filtering, or an unlabeled "medium-quality" portion?*
>
> You are right. To be precise, filtering occurs as early as Step 1. **Some raw corpora, when scored by the LLM, result in outputs that fail to follow the structure defined in our prompt**, which causes downstream parsing logic to break. In other cases, **some raw corpora trigger the LLM to produce outputs that contain safety-violating content**, making them unusable.
>
> In Step 2, because the source data pool contains a large number of highly similar corpora, we select only **a few of the most representative samples from each cluster** to improve diversity. Another important reason is that we aim to generate fused corpora that cover rare topics, which requires prioritizing representative rather than redundant samples.

---

### Official Review · Reviewer_PTLK · 2025-11-01

**Soundness:** 2
**Presentation:** 2
**Contribution:** 2
**Rating:** 2
**Confidence:** 4

**Summary:**

This paper proposes ENTP (Enhancing low-quality SFT data via Neural-Symbolic Text Purge-Mix), a framework aiming to reuse low-quality instruction-tuning data for LLM supervised fine-tuning. The approach combines a symbolic “purge” step (noise correction via a score transition matrix and clustering) with a neural “mix” stage (using LLMs to merge and regenerate synthetic instruction–response pairs). Experiments on five instruction-following benchmarks (MMLU, GSM8K, TruthfulQA, BBH, TyDiQA) claim that ENTP outperforms 13 data-selection baselines and even surpasses training on the full dataset.While the motivation—leveraging low-quality data—is interesting, the technical novelty, theoretical soundness, and empirical rigor are limited.

**Strengths:**

1. Although the proposed pipeline lacks genuine algorithmic novelty, it appears engineeringly robust and well-implemented, showing careful system design and integration.
2. Addresses a timely and relevant problem—the efficient utilization and enhancement of low-quality SFT data, which remains a critical bottleneck in instruction tuning.
3. Provides comprehensive empirical comparisons across multiple benchmarks and base models, offering a reasonably broad empirical validation despite limited methodological innovation.

**Weaknesses:**

1. The paper lacks genuine novelty while introducing excessive and unnecessary conceptual packaging. In essence, the method is merely a { clustering + prototype selection + data augmentation}  pipeline. The first two steps mainly reassemble engineering tricks already well-documented in prior works, offering no new algorithmic contribution. The third step is essentially an overcomplicated extension of self-refine[1] or critic-LLM[2] frameworks, repackaged under heavy “neural-symbolic fusion” terminology that significantly increases reading complexity without adding real substance. Moreover, the proposed “two-to-one” augmentation is a very common practice, similar to MathFusion-style data combination.
[1] Self-Refine: Iterative Refinement with Self-Feedback
[2] CritiqueLLM: Towards an Informative Critique Generation Model for Evaluation of Large Language Model Generation
[3] Mathfusion: Enhancing mathematic problem-solving of llm through instruction fusion


2. The paper does not provide any genuine ablation analysis to clarify the contribution of each module in the proposed three-step pipeline. The so-called “ablation study” in Section 4.3 only varies the overall data volume (20–100%) and does not isolate or remove individual components such as score correction, clustering, or neural-symbolic fusion. As a result, it remains unclear which step actually drives the reported improvements. In particular, the third module—the claimed Neural-Symbolic Two-to-One Fusion—is central to the paper’s narrative but lacks any empirical evidence demonstrating its necessity or effectiveness. A rigorous component-wise ablation (e.g., without fusion, without symbolic feedback, or using a single-LLM rewriting baseline) is needed to substantiate the claimed innovation.

3. Methodological complexity and scalability concerns. The ENTP pipeline involves multiple tightly coupled components—score correction, clustering, and a two-cycle neural-symbolic fusion loop. Step 3, in particular, requires iterative LLM invocations with symbolic loss back-propagation, which substantially increases computational cost and implementation difficulty. This complexity undermines the claimed data-efficiency advantage and makes the method difficult to reproduce or scale to larger corpora.

4. The comparison setup is problematic. Although the paper claims to enhance low-quality data, all baselines are actually data selection methods (e.g., KNN, Perplexity, IFD, DS2), which focus on filtering rather than enhancement or augmentation. This mismatch undermines the validity of the comparisons—ENTP synthesizes new data, while the baselines only select subsets of existing samples. A fair evaluation should instead compare ENTP with other data-enhancement frameworks (e.g., Self-Refine, Reflexion, Critic-LLM, or MathFusion-style augmentation). Without such comparisons, it is unclear whether ENTP offers any genuine advantage beyond generic LLM rewriting.

5. The evaluation setup is outdated and not aligned with the paper’s stated goal of enhancing instruction data. All benchmarks (MMLU, TruthfulQA, GSM8K, BBH, TyDiQA) are legacy instruction-following datasets that no longer reflect modern evaluation standards. To convincingly demonstrate data enhancement, the authors should compare against contemporary benchmarks such as AlpacaEval 2.0, MT-Bench, or ArenaHard, which better capture response helpfulness, reasoning depth, and stylistic alignment. Moreover, all evaluations remain within the instruction-tuning domain, leaving open the question of how well ENTP generalizes to other domains (e.g., reasoning, dialogue, code). Without such cross-domain and up-to-date evaluations, the claimed “enhancement” effect is insufficiently validated.

**Questions:**

1. Why did the authors choose GPT-4o-mini for the fusion stage instead of using an open-source small model?
2. Given that a large open-source model with a simple two-to-one merging and self-refine baseline could already achieve strong results, is such a complex “fusion” design necessary?
3. How does the proposed method generalize to other domains beyond instruction-tuning?

---

> ### Author Response · Authors · 2025-11-26
> **Response 1/3**
>
> We would like to thank Reviewer PTLK for the time and effort invested in reviewing this work. We will address individual comments below.
>
> # W1-0: *the method is merely a {clustering + prototype selection + data augmentation} pipeline ... The first two steps offer no new algorithmic contribution*
>
> First, regarding the statement that "*the method is merely a {clustering + prototype selection + data augmentation} pipeline,*", we feel that this characterization may not fully capture the scope of our method. **As detailed in Section 3 of our paper**, Step 1 first divides the data into high-quality and low-quality corpora. Importantly, we deliberately choose the low-quality corpus as the input for Step 2 (one-hop cluster construction). This choice itself is an innovation compared with the prevailing quality-first paradigm in data selection. Secondly, it should be noted that corpora fusion is not identical to corpus (or data) augmentation. The most intuitive difference is that corpora fusion requires **at least two corpora as input** to create a more diverse or representative corpus, whereas data augmentation typically works on **an individual corpus** to generate modified copies. Thirdly, in Step 3, we consistently emphasize that the fused data must preserve information from the original corpora, such as the topic, style, and the input–output alignment rules. This is why we describe Step 3 as *'neural-symbolic corpora fusion'*: every clearly defined rule corresponds to a symbolic constraint. These symbols determine which aspects of the original data must be retained, rather than allowing the model to regenerate data freely. In contrast, corpus augmentation seems not to consider these constraints. Therefore, we believe that using data augmentation to represent our entire Step 3 would be somewhat one‑sided.
>
> # W1-1: *The third step is essentially an overcomplicated extension of self-refine[1] or critic-LLM[2] frameworks*
>
> We respectfully disagree with this claim. By inspecting the codebases of Self-Refine[1] and Critic-LLM[2], we found that Self-Refine requests users to provide the golden examples for every task to perform few-shot prompting. Similarly, Critic-LLM requires the user to supply a reference answer together with two LLM-generated answers to their queries. In our task setting, **it is fundamentally impossible to know any "golden" or "reference" answer for *corpus 1 + corpus 2 = ?***. In fact, without carefully crafted prompts, LLMs merely concatenate the two corpora, producing outputs with no transition, no logical structure, and extremely poor readability. Such outputs cannot be considered valid or coherent corpora. Therefore, we consider Self-Refine[1], Critic-LLM[2], and our ENTP as belonging to distinct task tracks.
>
> [1] Self-Refine: Iterative Refinement with Self-Feedback
> [2] CritiqueLLM: Towards an Informative Critique Generation Model for Evaluation of Large Language Model Generation
>
> # W1-2: *MathFusion-style data combination*
>
> Thank you for suggesting this relevant baseline (MathFusion[3]). We will include its experimental results when addressing the other weaknesses discussed below.
>
> [3] Mathfusion: Enhancing mathematic problem-solving of llm through instruction fusion
>
> # W2: *The paper does not provide any genuine ablation analysis to clarify the contribution of each module in the proposed three-step pipeline. ...*
>
> Thanks for pointing this out. We use gpt-4o-mini-2024-07-18 to directly perform rewriting on each low-quality corpus and to directly fuse randomly paired corpora, bypassing both our clustering step (Step 2) and neural-symbolic fusion step (Step 3). We then randomly select 54888 resultant corpora to SFT Mistral-7B-v0.3. Additional ablation studies, along with the results of the single-LLM rewriting baseline, are reported below:
> | | mmlu | truthfulqa | gsm | bbh | tydiqa | Average |
> |:----:|:----:|:----:|:----:|:----:|:----:|:----:|
> |Rewriting|47.3|42.4|41.5|49.4|49.9|46.1|
> |direct_corpora_fusion_wo_step2&3|40.4|41.4|37.5|48.8|50.7|43.8|
> |direct_corpora_fusion_wo_step3|45.9|42.8|40.0|50.0|50.3|45.8|
> |ENTP|58.6|43.0|44.0|53.8|58.3|51.5|
>
> # W3: *... substantially increases computational cost ...*
>
> In terms of the computational cost, we completely understand your concern and are happy to provide approximate API costs of several baselines for reference.
> | | MathFusion[1] | Evol-Instruct[2] | ENTP |
> |:----:|:----:|:----:|:----:|
> |Per resultant corpus Cost (in USD)|0.004|0.003|0.005|
>
> [1] Mathfusion: Enhancing mathematic problem-solving of llm through instruction fusion
> [2] WizardLM: Empowering large pre-trained language models to follow complex instructions

---

> ### Author Response · Authors · 2025-11-26
> **Response 2/3**
>
> # W4-0: *The comparison setup is problematic.*
>
> We respectfully disagree with this claim. We believe that including data selection methods as baselines is essential. The motivation behind ENTP is that the prevailing quality-first data selection paradigm discards a large amount of low-quality data and, more critically, imposes a capability bottleneck on LLMs. Therefore, comparing against data selection baselines is necessary for two reasons:
> - 1, To empirically demonstrate that the bottleneck introduced by the quality-first paradigm indeed exists.
> - 2, To empirically show that the discarded low-quality data can in fact improve LLM performance on downstream tasks and thus should not be simply removed.
>
> # W4-1 & W1-2: *A fair evaluation should instead compare ENTP with other data-enhancement frameworks (e.g., Self-Refine[1], Reflexion[2], Critic-LLM[3], or MathFusion-style augmentation)*
>
> Thank you for the suggestion. As discussed in our response to W1-1, both Self-Refine [1] and Critic-LLM [3] are designed for corpus augmentation and are therefore not suitable for the task of corpora fusion. More specifically, these methods require users to provide, respectively, golden examples for few-shot prompting and the reference answer to the user query. Such requirements are infeasible in our setting, as it is impossible to supply any "golden" or "reference" answer for the query (**there is simply no well-defined ground truth for *'corpus 1 + corpus 2 = ?'***).
>
> For Reflexion[2], an agent-based method, it may appear applicable to the corpora fusion task. However, after inspecting its codebase, we found that it requires users to provide an action trajectory that specifies all intermediate steps needed to satisfy the user query. Once again, this is infeasible in our setting: **we have no way of knowing the step-by-step trajectory required to obtain the answer to the query *'corpus 1 + corpus 2 = ?'***. This is why we propose ENTP, to search within the LLM’s reasoning space and identify a reasoning path that can generate merged corpus aligned with human preferences.
>
> For MathFusion[4], which provides three fusion strategies (conditional, parallel, and sequential), we apply all three strategies to the low-quality corpora via random pairing using gpt-4o-mini-2024-07-18. We then evenly select 18296 merged corpora per strategy, resulting in a total of 54888 merged corpora, which are used to SFT Mistral-7B-v0.3. Additionally, as requested by another reviwer dHmZ, we provide two extra baseline, Evol-Instruct[5] and Self-Instruct[6], under the same settings for comparison. All executable baseline results will be included in the revised paper.
>
> The results are reported below:
> | | mmlu | truthfulqa | gsm | bbh | tydiqa | Average |
> |:----:|:----:|:----:|:----:|:----:|:----:|:----:|
> |MathFusion[4]|50.8|59.6|44.5|52.8|41.4|49.8|
> |Evol-Instruct[5]|54.0|57.5|33.5|53.1|42.8|48.2|
> |Self-Instruct[6]|53.1|43.8|45.0|55.2|50.9|49.6|
> |ENTP|58.6|43.0|44.0|53.8|58.3|51.5|
>
> [1] Self-Refine: Iterative Refinement with Self-Feedback
> [2] Reflexion: Language Agents with Verbal Reinforcement Learning
> [3] CritiqueLLM: Towards an Informative Critique Generation Model for Evaluation of Large Language Model Generation
> [4] Mathfusion: Enhancing mathematic problem-solving of llm through instruction fusion
> [5] WizardLM: Empowering large pre-trained language models to follow complex instructions
> [6] Self-Instruct: Aligning Language Model with Self Generated Instructions

---

> ### Author Response · Authors · 2025-11-26
> **Response 3/3**
>
> # W5-0: *All benchmarks (MMLU, TruthfulQA, GSM8K, BBH, TyDiQA) are legacy instruction-following datasets that no longer reflect modern evaluation standards.*
>
> Thank you for pointing this out. Indeed, our current evaluation benchmarks may be slightly outdated. We selected these benchmarks primarily because we adopted the exact same settings as DS2 [1], which was accepted at ICLR 2025. Moreover, LESS [2] also uses three of the benchmarks that we employ, and it is a 2024 publication. Based on our survey, these benchmarks continue to be widely used by researchers in the field.
>
> [1] DS2: Improving Data Efficiency via Curating LLM-Driven Rating Systems
> [2] LESS: Selecting Influential Data for Targeted Instruction Tuning
>
> # W5-1: *Moreover, all evaluations remain within the instruction-tuning domain, leaving open the question of how well ENTP generalizes to other domains (e.g., reasoning, dialogue, code).*
>
> First, comparing the instruction-tuning domain directly with reasoning, dialogue, or code domains is not entirely appropriate. Instruction-tuning corpora can include content related to reasoning, dialogue, or code; as long as the data follows an input-output pair format, it qualifies as instruction-tuning data. Secondly, our evaluation benchmarks do cover a diverse range of knowledge domains: **MMLU (factual knowledge)[1], TruthfulQA (truthfulness)[2], GSM8K (mathematical reasoning)[3], BBH (general reasoning)[4], and TyDiQA (multilinguality)[5]**. As for why all evaluation benchmarks focus on SFT-type tasks, it is because our ENTP mainly targets contributions to SFT tasks. This motivated our initial choice of 300K SFT examples as the starting point.
>
> [1] Measuring massive multitask language understanding
> [2] Truthfulqa: Measuring how models mimic human falsehoods
> [3] Training verifiers to solve math word problems
> [4] Challenging big-bench tasks and whether chain-of-thought can solve them
> [5] Tydiqa: A benchmark for information-seeking question answering in typologically diverse languages
>
> # Q1: *Why did the authors choose GPT-4o-mini for the fusion stage instead of using an open-source small model?*
>
> The main reason is that **we did not have any GPU support** while developing ENTP. Besides, referring to all the prompt templates provided in the Appendix, for each low-quality corpus, we aim to extract as much useful information as possible. In some cases, this may involve up-to-date information, which requires an LLM trained on relatively recent data. Furthermore, since our data pool consists of multiple datasets spanning different domains, we need an LLM that performs well across multiple knowledge domains. In addition, based on the experimental results of DS2[1], **gpt-4o-mini achieves the highest scores** among the candidates; therefore, we ultimately selected gpt-4o-mini as our LLM.
>
> [1] DS2: Improving Data Efficiency via Curating LLM-Driven Rating Systems
>
> # Q2: *Given that a large open-source model with a simple two-to-one merging and self-refine baseline could already achieve strong results, is such a complex “fusion” design necessary?*
>
> | | mmlu | truthfulqa | gsm | bbh | tydiqa | Average |
> |:----:|:----:|:----:|:----:|:----:|:----:|:----:|
> |Rewriting|47.3|42.4|41.5|49.4|49.9|46.1|
> |direct_corpora_fusion_wo_step2&3|40.4|41.4|37.5|48.8|50.7|43.8|
> |direct_corpora_fusion_wo_step3|45.9|42.8|40.0|50.0|50.3|45.8|
> |MathFusion|50.8|59.6|44.5|52.8|41.4|49.8|
> |ENTP|58.6|43.0|44.0|53.8|58.3|51.5|
>
> **It is certainly necessary**. First, according to our investigation, **none of the suggested data augmentation baselines (Self-Refine, Critic-LLM, Reflexion) are capable of performing the corpora fusion task**. The only method that can handle corpora fusion is MathFusion. However, our experimental results show that MathFusion underperforms our approach in overall performance. The core reason is that MathFusion’s original prompt templates are specifically designed for mathematical corpora (e.g., capturing variables or certain mathematical concepts), which are not suitable for general-domain data that contains little math-related information. Furthermore, **our additional ablation studies further demonstrate that each step of ENTP is essential**.
>
> # Q3: *How does the proposed method generalize to other domains beyond instruction-tuning?*
>
> Currently, we only demonstrate the effectiveness of ENTP at the SFT stage. If sufficient GPU resources become available, we plan to adapt ENTP to pre-training corpora to explore its effectiveness at the pre-training stage.

---

### Official Review · Reviewer_rzzr · 2025-11-02

**Soundness:** 2
**Presentation:** 2
**Contribution:** 2
**Rating:** 4
**Confidence:** 4

**Summary:**

This paper proposes ENTP, a framework that enhances low-quality instruction data rather than discarding it.

First, the authors use a Score Transition Matrix to correct noisy LLM-generated quality ratings, creating a more reliable separation between high-quality and low-quality samples. Second, they apply clustering algorithms to group similar low-quality samples and select representative examples from each cluster. Third, they fuse pairs of low-quality samples into single information-rich synthetic samples using GPT-4, guided by hand-crafted symbolic rules that ensure key terms are preserved and answers are complete.

Experimental results on three LLMs across five benchmarks show that low-quality data contains valuable signal when intelligently enhanced rather than discarded.

**Strengths:**

The paper challenges the dominant "quality-first" paradigm by demonstrating that low-quality data should be enhanced rather than discarded, addressing the critical problem that high-quality instruction data has been largely exhausted. This approach achieves better results with 54K synthetic samples derived from low-quality data than training on the full 300K original dataset, providing a practical path forward as the LLM field faces a data scarcity bottleneck.

The experimental design tests three different base models across five diverse benchmarks and compares against 13 strong baselines including both LLM-free methods and modern LLM-based selection approaches. Extensive ablation studies examine different fusion types, data proportions, and combinations with high-quality data, with consistent improvements across all settings rather than cherry-picked results.

The paper provides empirical evidence for a counterintuitive claim: "low-quality" data contains partial but valuable information that can be systematically extracted and recombined through intelligent fusion. This reframes data curation from a filtering problem (keep the top 10%) to a synthesis problem (enhance the bottom 90%), showing that traditional selection methods face a structural bottleneck that cannot be overcome by better filtering alone.

**Weaknesses:**

The paper uses confusing and misleading terminology throughout, particularly the term "backpropagation" which suggests gradient-based optimization but actually refers to simple rule-based prompt template switching. The "neural-symbolic" framing overstates the sophistication of what is essentially an if-else logic system that selects from 9 hand-written prompt templates based on structured error detection. The writing is dense and difficult to follow, with critical implementation details buried in extensive appendices, making it hard to understand what the method actually does versus what the terminology implies.

The paper does not compare against the most obvious baseline: simply using GPT-4 to directly enhance or rewrite individual low-quality samples (1-to-1 enhancement) rather than the complex 2-to-1 fusion approach. Without this comparison, it's impossible to determine whether the gains come from the sophisticated fusion mechanism or simply from having GPT-4 rewrite poor-quality data. The paper also lacks ablations comparing fusion with random pairing versus their clustering approach, and fusion without symbolic components (pure neural generation), making it difficult to isolate which components actually contribute to performance.

The method involves multiple elaborate steps (Score Transition Matrix correction, one-hop clustering, representative selection via MMR, domain analysis, two-cycle iterative refinement) but provides no evidence that this complexity is necessary. Simple alternatives like random pairing of low-quality samples with direct GPT-4 fusion, or even straightforward GPT-4 enhancement without any clustering, might achieve similar results with far less computational overhead and implementation complexity. The lack of ablations removing individual components makes it unclear whether the gains justify the significantly increased system complexity compared to simpler data enhancement approaches.

**Questions:**

Could the authors provide experimental results comparing ENTP against direct GPT-4 enhancement with matched sample counts (54K enhanced samples) or matched computational budgets (same number of LLM API calls)?

---

> ### Author Response · Authors · 2025-11-26
> **Response 1/1**
>
> We want to thank the reviewer for their positive feedback and comments. We will address individual comments below.
>
> # W1: *The paper uses confusing and misleading terminology throughout, particularly the term "backpropagation" which suggests gradient-based optimization but actually refers to simple rule-based prompt template switching. ...*
>
> First, we sincerely apologize for any confusion caused by the technical terminology (e.g., "backpropagation") in the paper. In classical neural networks, back-propagation essentially redistributes the error observed at the output layer backwards, layer-by-layer, to every weight and bias, and then fine-tunes each parameter along its gradient. Our ENTP(Step 3) mirrors this mechanism, but operates entirely in natural language.
>
> Concretely, for every intermediate merged corpus the ICD (Information Completeness Detection) operator generates a symbolic loss, essentially, a structured JSON object that functions as an information-bearing carrier. Taking one of its keys, "questions_exists" as an example, it enumerates (**all detailes had been shown in Figure 14 in Appendix**):
> 1, whether the "### User" session contains an explicit question;
> 2, if so, whether the question is open- or closed-ended;
> 3, whether sufficient background knowledge has been supplied to answer it;
> 4, what background knowledge is currently provided and what remains missing; and
> 5, the exact supplementary knowledge required.
>
> Consequently, this JSON object simultaneously **assesses the present merged corpus** and **prescribes the precise direction for revision**, an explicit **natural-language analogue of a loss value and its gradient**. We term it symbolic loss because every key is an information-laden symbol, and, in our ENTP, every rule is likewise encoded as a symbol. No terminology was chosen to mislead.
>
> # W2-0 & Q: *The paper does not compare against the most obvious baseline: simply using GPT-4 to directly enhance or rewrite individual low-quality samples (1-to-1 enhancement) ...*
>
> We use gpt-4o-mini-2024-07-18 to directly perform one-to-one enhancement on every low-quality sample. Subsequently, we randomly selecte 54888 augmented corpora to SFT Mistral-7B-v0.3. The downstream evaluations are reported below:
>
> | | mmlu | truthfulqa | gsm | bbh | tydiqa | Average |
> |:----:|:----:|:----:|:----:|:----:|:----:|:----:|
> |1-to-1 Enhancement|47.3|42.4|41.5|49.4|49.9|46.1|
> |ENTP|58.6|43.0|44.0|53.8|58.3|51.5|
>
> # W2-1 & W3: *The paper also lacks ablations comparing fusion with random pairing versus their clustering approach, and fusion without symbolic components (pure neural generation), making it difficult to isolate which components actually contribute to performance. ... provides no evidence that this complexity is necessary ... The lack of ablations removing individual components makes it unclear whether the gains justify the significantly increased system complexity compared to simpler data enhancement approaches.*
>
> To isolate the individual contribution of each ENTP component, we ablate both the clustering step (Step 2) and the neural–symbolic fusion step (Step 3). Concretely, we employ gpt-4o-mini-2024-07-18 to directly fuse randomly paired corpora, bypassing both the clustering and neural-symbolic fusion steps. From the resulting pool we randomly subsample 54888 merged corpora to SFT Mistral-7B-v0.3. Downstream evaluation results are reported below:
> | | mmlu | truthfulqa | gsm | bbh | tydiqa | Average |
> |:----:|:----:|:----:|:----:|:----:|:----:|:----:|
> |direct_corpora_fusion_wo_step2&3|40.4|41.4|37.5|48.8|50.7|43.8|
> |direct_corpora_fusion_wo_step3|45.9|42.8|40.0|50.0|50.3|45.8|
> |ENTP|58.6|43.0|44.0|53.8|58.3|51.5|
>
> According to the results of our ablation study above, directly fusing corpora from our selected corpora set (i.e., after our clustering step) yields better performance than fusing from the original low-quality corpora set (i.e., without the clustering step). This is because the raw low-quality set contains many similar samples, and random pairing makes it unlikely to generate fused corpora that contain genuinely new knowledge. In contrast, our clustering step selects a more representative subset of the raw low-quality corpora, although the subsequent pairing is still random, clustering greatly reduces the probability of pairing two highly similar samples. As a result, the fused corpora are more likely to include new knowledge. Moreover, ENTP comprises both our clustering step (step 2) and neural‑symbolic fusion step (step 3), and achieves the highest average score, indicating that each component in ENTP contributes positively. This further demonstrates that every step is necessary.

---

### Official Review · Reviewer_9b2A · 2025-11-02

**Soundness:** 3
**Presentation:** 2
**Contribution:** 2
**Rating:** 4
**Confidence:** 3

**Summary:**

The paper proposes ENTP, a neural–symbolic pipeline that turns “low-quality” instruction–response data into useful supervision for SFT by purging noise and mixing retained signals into synthesized training pairs. It first scores and corrects sample quality with a score-transition matrix to mitigate LLM-rater inconsistency, partitions data into high/low sets, then clusters the low-quality pool to pick representative items before a two-to-one fusion stage that applies symbolic rules and LLM generation in iterative loops to reconstruct richer, compact prompts and answers. The resulting merged corpus is combined with any high-quality data and used to fine-tune base models. On five OpenLLM benchmarks (MMLU, TruthfulQA, GSM8K, BBH, TyDiQA) and two backbones (Mistral-7B, Llama-3.1-8B), ENTP-derived data—sourced solely from low-quality sets—consistently beats 13 data-selection baselines and can rival or exceed training on the full ~300K dataset, with performance improving as more ENTP data are used.

**Strengths:**

The paper is original in reframing “low-quality” instruction data as valuable signal via a neural–symbolic *purge-mix* pipeline: it corrects noisy LLM quality scores with a score-transition matrix to address rater inconsistency, then fuses representative low-quality samples into richer training pairs using rule-guided and LLM-driven steps, rather than discarding them. This tackles two known caveats of quality-first filtering and is clearly motivated in the introduction.

**Weaknesses:**

The method feels overengineered and difficult to follow end-to-end, which raises the barrier to adoption. A concrete, running example would help: start from two real “low-quality” instruction–response pairs, show how scores are corrected and items clustered, present the exact symbolic-loss/ICD artifact produced, and then the final fused pair. Empirically, the benchmark coverage coverage is too narrow: adding harder math benchmarks (AIME ’24/’25, MATH500) plus coding tasks, and IFEval and SimpleQA will increase task coverage.

**Questions:**

Is it possible to run one more baseline for this paper: LESS: Selecting Influential Data for Targeted Instruction Tuning? by Xia et al, ICML 2024

---

> ### Author Response · Authors · 2025-11-26
> **Response 1/7**
>
> We want to thank the reviewer for their positive feedback and comments. We will address individual comments below.
>
> # W1-0: *A concrete, running example would help: ...*
>
> Thanks for your suggestion. We provide a concrete example to illustrate the operational mechanism of ENTP step by step. In addition, as requested by reviewer dHmZ, we will include this example in our revised paper.
>
> ## Step 1 (Session 3.1)
> Given a raw data pool of 300K corpora, our goal is to distinguish low-quality from high-quality samples. To this end, we adopt the LLM-rating prompt template from DS2[1] (Figure 6 in the Appendix) to score each corpus along three dimensions: *Rarity*, *Complexity*, and *Informativeness*. To better separate mid-range samples, we compress the original LLM rating scale (1–10) to a 0–5 scale: scores 1–4 are mapped to 0, scores ≥9 to 5, and scores within [5, 8] are shifted down by 4 to fall within [1, 4]. This yields a final LLM-rating distribution strictly within [0, 5]. The next step is to embed all source corpora so that cosine similarity can be used to curate the raw LLM-rating scores. The core idea is *Clusterability*[2]: the overall score of a corpus should, with high probability, be similar or identical to that of its two nearest semantic neighbors. This mechanism helps mitigate numerical biases that may arise from the LLM’s pre-training data. At this point, we obtain curated overall scores in the range of [0, 5] for all 300K source corpora. We then select all corpora with scores between 0 and 2 as the low-quality set. This completes Step 1.
>
> [1] DS2: Improving Data Efficiency via Curating LLM-Driven Rating Systems
> [2] Clusterability as an Alternative to Anchor Points When Learning with Noisy Labels
>
> ## Step 2 (Session 3.2)
> After identifying the low-quality set, this step aims to select a representative subset from it. We begin by randomly selecting a corpus as the cluster centroid, computing cosine similarities with all other corpora, and assigning any corpus with a similarity ≥ 0.9 to the same cluster. Next, we apply k-means clustering [3] to partition each initial cluster into sub-clusters. The optimal number of sub-clusters (k) is determined by evaluating clustering quality over a predefined range of k values (2–10) using the Silhouette method [4]. For any initial cluster containing at least two sub-clusters, we select two representative corpora from each sub-cluster. The first one is chosen with the highest cosine similarity to the sub-cluster’s averaged embedding vector. To promote diversity, the second one is selected with the highest MMR score. At this point, we have obtained the representative subset of the low-quality corpora.
>
> [3] Some methods for classification and analysis of multivariate observations
> [4] Silhouettes: a graphical aid to the interpretation and validation of cluster analysis.
>
> ## Step 3 (Session 3.3)
> Randomly select two corpora from the representative low-quality set as input:
> ```
> Raw Corpus A:
> ### User
> input question: Write a sentence not in English.
> output answer: от того, что появляется белая лилия, которая цветет с июня по сентябрь, который распространяется на соленый воздух нежный аромат.
> input question: Write a sentence not in English.
> output answer: При других заболеваниях лечение может идти до 1 года и далее. Биокоррекция Васильева может осуществляться как в клинических условиях, так и амбулаторно.
> input question: Write a sentence not in English.
> output answer: Насколько существенно ее воздействие на важные бизнес-процессы?
> input question: Write a sentence not in English.
> output answer:
> ### Assistant
> Ваш входной идентификационный код будет также Вашим кодовым именем при участии в конкурсе. Информация про введение данных будет отослана Вам по электронной почте FTP с адресом и паролем.
>
> Raw Corpus B:
> ### User
> input question: Write a sentence not in English.
> output answer: Отель Fayal Resort Hotel будет показан на карте, если Вы включите JavaScript.
>
>
> Q: Translate "How to eat: Just drink it." to Russian?
> Yes: Как употреблять: просто выпейте!
>
>
> [Q]: За 72 года своего существования "Башнефть" сформировалась как мощный многопрофильный нефтегазодобывающий комплекс.
>
> Translate this to English?
> [A]: Within 72 years of its existence, Bashneft has become powerful multi-structural oil and gas extracting enterprise.
>
>
> Question:
> Для Профессионального комплекта существуют корпоративные решения с возможностью определять состав дополнительных специализированных словарей.
>
> Could you please translate this to English?
> Answer:
> For the Professional package there exist enterprise solutions with the possibility to determine the composition of special dictionaries.
>
>
> test: Балахтинский район
> English?
>
> translation: Balakhtinsky District
>
>
> input question: Write a sentence not in English.
> output answer:
> ### Assistant
> Лангоун, Майкл
> ```

---

> ### Author Response · Authors · 2025-11-26
> **Response 2/7**
>
> > Step 1: Domain Analysis (Prompt template has been shown in Figure 10.)
> ```
> LLM Response:
> {
>     "corpus_A_domain": "Language Translation",
>     "corpus_B_domain": "Language Translation",
>     "corpus_A_key_terms_list": [
>         "sentence construction",
>         "foreign language",
>         "biocorrection",
>         "business processes"
>     ],
>     "corpus_B_key_terms_list": [
>         "translation",
>         "enterprise solutions",
>         "specialized dictionaries",
>         "multilingual"
>     ],
>     "matching_rules_derived_from_corpus_A": "Questions from the '### User' section often ask for non-English sentences or contexts, and answers in the '### Assistant' section provide translations or related foreign phrases.",
>     "matching_rules_derived_from_corpus_B": "The '### User' questions request either translations or sentences in non-English, while the '### Assistant' responses deliver translations or contextual explanations in both English and Russian.",
>     "relationship": "related-domain"
> }
> ```
> Once the relationship is identified, the corresponding fusion strategy is selected (all strategies are provided in Figure 11). The MCG template (Figure 13) is then applied to generate the initial merged corpus.
> > Step 2 & 3: Strategy Selection & Merged Corpus Generation
> ```
> LLM Response:
> {
>     "overall_response": [
>         {
>             "domain_type": "related-domain",
>             "applied_strategy": "conceptual_analogy",
>             "corpus_fusion_variant": "### User\nIn the realm of Language Translation, both corpora emphasize the importance of constructing sentences in foreign languages. For instance, corpus A provides examples of non-English sentences, while corpus B focuses on translating these sentences into English. This highlights the interconnectedness of language processing and multilingual communication. How can we further enhance the understanding of cultural linguistics in translation?\n\n### Assistant\nWe can enhance the understanding of cultural linguistics in translation by incorporating cultural context into the translation process, ensuring that idiomatic expressions and cultural references are accurately conveyed."
>         },
>         {
>             "domain_type": "related-domain",
>             "applied_strategy": "process_mapping",
>             "corpus_fusion_variant": "### User\nWhen considering enterprise solutions for Language Translation, corpus B discusses specialized dictionaries that can aid in translation tasks. Meanwhile, corpus A mentions biocorrection and its relevance to business processes. This suggests a need for a structured approach to integrating these specialized tools into translation workflows. What are the best practices for implementing such solutions in a multilingual environment?\n\n### Assistant\nBest practices for implementing specialized tools in a multilingual environment include conducting thorough needs assessments, training staff on the use of these tools, and continuously evaluating the effectiveness of the solutions in improving translation accuracy."
>         },
>         {
>             "domain_type": "related-domain",
>             "applied_strategy": "term_bridging",
>             "corpus_fusion_variant": "### User\nBoth corpora highlight key terms such as 'sentence construction' from corpus A and 'translation' from corpus B. This overlap indicates a shared focus on the mechanics of language. Additionally, terms like 'foreign language' and 'multilingual' suggest a broader context of language processing. How can we leverage these shared terms to create a more cohesive language translation framework?\n\n### Assistant\nWe can leverage these shared terms by developing a unified glossary that defines each term in the context of both corpora, facilitating better communication and understanding among language professionals."
>         }
>     ]
> }
> ```

---

> ### Author Response · Authors · 2025-11-26
> **Response 3/7**
>
> Since we provide three fusion strategies for each relationship, three different merged corpora will be generated. However, for clarity of presentation, we illustrate the process using only the initial merged corpus produced by the *'term_bridge'* strategy. The following step is to obtain the **Symbolic Loss** for this merged corpus.
> > Step 4: Information Completeness Detection (Prompt template has been shown in Figure 14.)
> ```
> LLM Response:
> {
>     "technical_report": {
>         "term_coverage": {
>             "included": {
>                 "corpus_A_key_terms_list": [
>                     "sentence construction",
>                     "foreign language",
>                     "biocorrection",
>                     "business processes"
>                 ],
>                 "corpus_B_key_terms_list": [
>                     "translation",
>                     "enterprise solutions",
>                     "specialized dictionaries",
>                     "multilingual"
>                 ]
>             },
>             "missing": {
>                 "corpus_A_key_terms_list": [],
>                 "corpus_B_key_terms_list": []
>             }
>         },
>         "question_analysis": {
>             "question_exists": {
>                 "question_type": "open-ended",
>                 "needs_extra_knowledge_or_background_supplementation": {
>                     "context_contain": "The '### User' section discusses the overlap of key terms and asks how to leverage them for a cohesive framework.",
>                     "context_missing": "The '### User' section lacks specific examples or details on what aspects of the shared terms should be leveraged, which could help in formulating a more precise answer."
>                 }
>             }
>         },
>         "answer_quality": {
>             "if_needs_to_re_answer": {
>                 "explanation": "The answer provided in the '### Assistant' section does not directly address the last question from the '### User' section, which asks how to leverage shared terms to create a cohesive language translation framework. The response is more about developing a glossary rather than providing a cohesive framework."
>             }
>         }
>     }
> }
> ```
> According to the resultant symbolic loss, the current merged corpus has **missing necessary context in the '### User' section**, and the **'### Assistant' section does not provide a direct and clear answer** to the question from '### User' section. Thus, the initial prompt template for Merged Corpus Generation (Figure 13) is updated to the version shown in Figure 24 to generate the enhanced merged corpus.
> > Step 5: Enhanced Corpus Generation (Prompt template has been shown in Figure 24.)
> ```
> LLM Response:
> {
>     "enhanced_corpus": "### User\nBoth corpora highlight key terms such as 'sentence construction' from corpus A and 'translation' from corpus B. This overlap indicates a shared focus on the mechanics of language. Additionally, terms like 'foreign language' and 'multilingual' suggest a broader context of language processing. How can we leverage these shared terms, particularly 'sentence construction' and 'translation', to create a more cohesive language translation framework? For instance, what specific strategies can we implement that utilize 'business processes' and 'specialized dictionaries' to enhance our approach to multilingual translation?\n\n### Assistant\nWe can leverage these shared terms by developing a unified glossary that defines each term in the context of both corpora, facilitating better communication and understanding among language professionals. In addition, we should consider implementing strategies that integrate 'business processes' with 'sentence construction' techniques to streamline translation workflows. Utilizing 'specialized dictionaries' can also enhance accuracy in translations, particularly in a multilingual context. By aligning these elements, we can create a cohesive language translation framework that not only addresses the mechanics of language but also improves the overall efficiency and effectiveness of translation practices."
> }
> ```

---

> ### Author Response · Authors · 2025-11-26
> **Response 4/7**
>
> Again, the following step is to obtain the **Symbolic Loss** for the resultant enhanced merged corpus from the last step.
> > Step 4: Information Completeness Detection (ICD) For Enhanced Corpus (Prompt template has been shown in Figure 14.)
> ```
> LLM Response:
> {
>     "technical_report": {
>         "term_coverage": {
>             "included": {
>                 "corpus_A_key_terms_list": [
>                     "biocorrection",
>                     "business processes",
>                     "foreign language"
>                 ],
>                 "corpus_B_key_terms_list": [
>                     "translation",
>                     "enterprise solutions",
>                     "specialized dictionaries",
>                     "multilingual"
>                 ]
>             },
>             "missing": {
>                 "corpus_A_key_terms_list": [
>                     "sentence construction"
>                 ],
>                 "corpus_B_key_terms_list": []
>             }
>         },
>         "question_analysis": {
>             "question_exists": {
>                 "question_type": "open-ended",
>                 "needs_extra_knowledge_or_background_supplementation": "nope"
>             }
>         },
>         "answer_quality": {
>             "if_needs_to_re_answer": "nope"
>         }
>     }
> }
> ```
> Compared to the symbolic loss of the initial merged corpus, the current symbolic loss only reflects missing key terms from source corpus A. This indicates that the enhanced merged corpus contains clear questions and answers and provides sufficient background knowledge to address the questions. The only remaining issue is that the coverage of key terms is slightly insufficient. Nonetheless, compared to the initial merged corpus, the current version represents a significant improvement. Therefore, to address the issue of missing key terms, we update the current template to the version shown in Figure 17 to generate a better merged corpus covering a broader range of domains.
> > Step 5: Enhanced Corpus Generation (Prompt template has been shown in Figure 17.)
> ```
> LLM Response:
> {
>     "enhanced_corpus": "### User\nBoth corpora highlight key terms such as 'sentence construction' from corpus A and 'translation' from corpus B. This overlap indicates a shared focus on the mechanics of language. Additionally, terms like 'foreign language' and 'multilingual' suggest a broader context of language processing. How can we leverage these shared terms, particularly 'sentence construction' and 'translation', to create a more cohesive language translation framework? For instance, what specific strategies can we implement that utilize 'business processes' and 'specialized dictionaries' to enhance our approach to multilingual translation?\n\n### Assistant\nWe can leverage these shared terms by developing a unified glossary that defines each term in the context of both corpora, facilitating better communication and understanding among language professionals. In particular, the concept of 'sentence construction' is crucial as it involves the mechanisms of structuring sentences effectively in various languages. Additionally, we should consider implementing strategies that integrate 'business processes' with 'sentence construction' techniques to streamline translation workflows. For example, application scenarios such as utilizing 'specialized dictionaries' can enhance accuracy in translations, particularly in a multilingual context. By aligning these elements, we can create a cohesive language translation framework that not only addresses the mechanics of language but also improves the overall efficiency and effectiveness of translation practices."
> }
> ```
> Again, the following step is to obtain the **Symbolic Loss** for the resultant enhanced merged corpus from the last step.
> > Step 4: Information Completeness Detection (ICD) For Enhanced Corpus
> ```
> LLM Response:
> {
>     "technical_report": {
>         "term_coverage": {
>             "included": {
>                 "corpus_A_key_terms_list": [
>                     "biocorrection",
>                     "business processes",
>                     "foreign language",
>                     "sentence construction"
>                 ],
>                 "corpus_B_key_terms_list": [
>                     "enterprise solutions",
>                     "multilingual",
>                     "specialized dictionaries",
>                     "translation"
>                 ]
>             },
>             "missing": {
>                 "corpus_A_key_terms_list": [],
>                 "corpus_B_key_terms_list": []
>             }
>         },
>         "question_analysis": {
>             "question_exists": {
>                 "question_type": "open-ended",
>                 "needs_extra_knowledge_or_background_supplementation": "nope"
>             }
>         },
>         "answer_quality": {
>             "if_needs_to_re_answer": "nope"
>         }
>     }
> }
> ```

---

> ### Author Response · Authors · 2025-11-26
> **Response 5/7**
>
> According to the current symbolic loss, all conditions have been satisfied: no key terms are missing, the '### User' section provides a clear question along with the necessary background knowledge to answer it, and the '### Assistant' section provides a precise answer to the question. Thus, **the resultant merged corpus successfully completes the entire Cycle 1 process**. The next step is to double-check the quality of the answer in the '### Assistant' section of the current resultant merged corpus (obtaining the **symbolic loss** from the FAC operator).
> > Step 6: Final Answer Check (Prompt template has been shown in Figure 12.)
> ```
> LLM Response:
> {
>     "if_containing_direct_answer": {
>         "direct_answer": "We can leverage these shared terms by developing a unified glossary that defines each term in the context of both corpora, facilitating better communication and understanding among language professionals. Additionally, we should consider implementing strategies that integrate 'business processes' with 'sentence construction' techniques to streamline translation workflows. For example, utilizing 'specialized dictionaries' can enhance accuracy in translations, particularly in a multilingual context."
>     },
>     "if_containing_information_needed_to_delete": "nope"
> }
> ```
> According to the symbolic loss from the FAC operator, the current '### Assistant' section contains a direct answer to the question from '### User' and includes no irrelevant or redundant information, indicating that **the current merged corpus has successfully passed the entire Cycle 2 process**.

---

> ### Author Response · Authors · 2025-11-26
> **Response 6/7**
>
> ## Final Merged Corpus
> We present the input corpora pair and their resultant merged corpus (**'enhanced_corpus'**) below:
> ```
> {
>     "raw_corpus_A": "### User\ninput question: Write a sentence not in English.\noutput answer: от того, что появляется белая лилия, которая цветет с июня по сентябрь, который распространяется на соленый воздух нежный аромат.\ninput question: Write a sentence not in English.\noutput answer: При других заболеваниях лечение может идти до 1 года и далее. Биокоррекция Васильева может осуществляться как в клинических условиях, так и амбулаторно.\ninput question: Write a sentence not in English.\noutput answer: Насколько существенно ее воздействие на важные бизнес-процессы?\ninput question: Write a sentence not in English.\noutput answer:\n### Assistant\nВаш входной идентификационный код будет также Вашим кодовым именем при участии в конкурсе. Информация про введение данных будет отослана Вам по электронной почте FTP с адресом и паролем.",
>     "raw_corpus_B": "### User\ninput question: Write a sentence not in English.\noutput answer: Отель Fayal Resort Hotel будет показан на карте, если Вы включите JavaScript.\n\n\nQ: Translate \"How to eat: Just drink it.\" to Russian?\nYes: Как употреблять: просто выпейте!\n\n\n[Q]: За 72 года своего существования \"Башнефть\" сформировалась как мощный многопрофильный нефтегазодобывающий комплекс.\n\nTranslate this to English?\n[A]: Within 72 years of its existence, Bashneft has become powerful multi-structural oil and gas extracting enterprise.\n\n\nQuestion:\nДля Профессионального комплекта существуют корпоративные решения с возможностью определять состав дополнительных специализированных словарей.\n\nCould you please translate this to English?\nAnswer:\nFor the Professional package there exist enterprise solutions with the possibility to determine the composition of special dictionaries.\n\n\ntest: Балахтинский район\nEnglish?\n\ntranslation: Balakhtinsky District\n\n\ninput question: Write a sentence not in English.\noutput answer:\n### Assistant\nЛангоун, Майкл",
>     "enhanced_corpus": "### User\nBoth corpora highlight key terms such as 'sentence construction' from corpus A and 'translation' from corpus B. This overlap indicates a shared focus on the mechanics of language. Additionally, terms like 'foreign language' and 'multilingual' suggest a broader context of language processing. How can we leverage these shared terms, particularly 'sentence construction' and 'translation', to create a more cohesive language translation framework? For instance, what specific strategies can we implement that utilize 'business processes' and 'specialized dictionaries' to enhance our approach to multilingual translation?\n\n### Assistant\nWe can leverage these shared terms by developing a unified glossary that defines each term in the context of both corpora, facilitating better communication and understanding among language professionals. In particular, the concept of 'sentence construction' is crucial as it involves the mechanisms of structuring sentences effectively in various languages. Additionally, we should consider implementing strategies that integrate 'business processes' with 'sentence construction' techniques to streamline translation workflows. For example, application scenarios such as utilizing 'specialized dictionaries' can enhance accuracy in translations, particularly in a multilingual context. By aligning these elements, we can create a cohesive language translation framework that not only addresses the mechanics of language but also improves the overall efficiency and effectiveness of translation practices."
> }
> ```

---

> ### Author Response · Authors · 2025-11-26
> **Response 7/7**
>
> # W1-1: *Empirically, the benchmark coverage coverage is too narrow: adding harder math benchmarks (AIME ’24/’25, MATH500) plus coding tasks, and IFEval and SimpleQA will increase task coverage*
>
> Thank you for pointing this out. Indeed, the tasks covered by our current evaluation benchmarks are still limited. We selected these benchmarks primarily because we adopted the exact same settings as DS2 [1], which was accepted at ICLR 2025. Moreover, LESS [2] also uses three of the benchmarks that we employ, and it is a 2024 publication. Based on our survey, these benchmarks continue to be widely used by researchers in the field. As for adding more evaluation benchmarks, due to budget and time constraints, we leave this for future work.
>
> [1] DS2: Improving Data Efficiency via Curating LLM-Driven Rating Systems
> [2] LESS: Selecting Influential Data for Targeted Instruction Tuning
>
> # Q1: *Is it possible to run one more baseline for this paper: LESS[1]?*
>
> Thank you so much for your suggestion. First, LESS[1] requires splitting a portion of each evaluation benchmark as a validation set to approximate the test set distribution and to build a gradient datastore, which is then used to select the top-k most suitable training corpora for each task. Compared to ENTP, this method is somewhat unfair, as we do not use any corpora from the evaluation benchmarks. Moreover, achieving the optimal scores for LESS on each evaluation task requires five rounds of SFT, which is extremely time- and resource-consuming. Nonetheless, based on your suggestion, we provide experimental results for both LESS and ENTP for comparison.
>
> - Experimental Setup: For each evaluation task, we select top-54888 corpora to SFT Mistral-7B-v0.3. All reported results are based on the corpora sets selected exclusively for the corresponding tasks.
>
> | | mmlu | truthfulqa | gsm | bbh | tydiqa | Average |
> |:----:|:----:|:----:|:----:|:----:|:----:|:----:|
> |LESS[1]|54.1|46.2|44.0|53.8|50.5|49.7|
> |ENTP|58.6|43.0|44.0|53.8|58.3|51.5|
>
> In addition, we also report the GPU runtime and whether a validation set is required for both LESS and ENTP.
> | | GPU Runtime (in GPU-hours) | Validation Set |
> |:----:|:----:|:----:|
> |LESS[1]|≈152.5|Required|
> |ENTP|≈17|Not Required|
>
> [1] LESS: Selecting Influential Data for Targeted Instruction Tuning

---

### Meta-Review · Area_Chair_HfnM · 2026-01-02

**Summary:**

The paper challenges the established quality-first SFT paradigm, which typically discards the majority of available data as low-quality. The authors argue that this selective approach creates a performance bottleneck and ignores valuable signals within supposedly inferior samples. To resolve this, they introduce ENTP, a framework that reconstructs these samples into high-value training data.

The methodology begins with Low-Quality Dataset Construction, where an LLM assigns quality scores that are subsequently refined by a Score Transition Matrix to mitigate rater inconsistency. The framework then performs Representative Corpus Selection using one-hop clustering and MMR to isolate a diverse subset of the low-quality pool. The final Neural-Symbolic Two-to-One Fusion stage utilizes connectionist operators and symbolic rules to iteratively merge pairs of samples.

**Reviewer Concerns:**

- The paper provides a pragmatic perspective by treating low-quality data as a critical resource rather than waste. This is particularly relevant as high-quality web data becomes increasingly scarce.
- During the rebuttal, the authors demonstrated significant effort by adding multiple synthesis baselines (LESS, MathFusion, Self-Instruct, Evol-Instruct) and performing component-wise ablations for Step 2 and Step 3 .

**Reviewer Scores:**

- A recurring criticism from reviewers (rzzr, PTLK) is the use of "misleading" terminology. Framing a rule-based iterative prompt system as "backpropagation" and "neural-symbolic" is viewed as excessive conceptual packaging for what is essentially a series of if-else logic checks and prompt template switching. While the authors defended this as a "natural-language analogue" to gradients, the complexity of the framework still seems overstated.
- The ENTP pipeline is exceedingly complex, requiring five different connectionist operators and two iterative loops per sample pair. While the authors provided a cost analysis, the high number of LLM calls (gpt-4o-mini) and iterative steps raise concerns about the practical scalability and reproducibility of the method compared to simpler 1-to-1 rewriting.
- As pointed out by Reviewer dHmZ, it is difficult to disentangle whether the performance gains stem from the specific ENTP architecture or simply from high-quality knowledge distillation from the "oracle" model (gpt-4o-mini). The authors' defense that the framework is a "structural prerequisite" for the distillation does not fully mitigate the concern that a simpler iterative prompting of gpt-4o-mini might yield similar gains.

---

### Decision · Program_Chairs · 2026-01-26

Reject